# GTPase Rab11b and effector Rab11-FIP2 promote NLRP3 stability during inflammasome priming

Caroline S Gravastrand[1], Maria Yurchenko[1], Stine Kristensen[1], Astrid Skjesol[1], Carmen Chen [1], Sindre Ullmann[1], Zunaira Iqbal [1], Karoline Ruud Dahlen[1], Kashif Rasheed [1], Unni Nonstad[1], Liv Ryan[1], Terje Espevik[1,2] & Harald Husebye [1✉]

## Abstract

**Membrane trafficking through the trans-Golgi network has been shown to guide activation of the NLRP3 inflammasome. Rab11 GTPases and their effector Rab11-FIP2 regulate endosomal trafficking and retrograde transport. Here, we demonstrate that Rab11b and Rab11-FIP2 contribute to NLRP3 and pro-IL-1β stabilization during the inflammasome priming phase, which is followed by inflammasome activation. We show Rab11-FIP2 to promote TAK1 phosphorylation and TAK1-mediated activation of IKKβ, a process controlling NLRP3 translocation to the trans-Golgi network. Human NLRP3 and Rab11-FIP2 bind each other via their phosphatidylinositol-4 phosphate (PI4P)-binding domains KMKK and N-terminal C2 domain, respectively. We also provide evidence indicating that Rab11-FIP2 stabilizes NLRP3 on early endosomes, which is important for ASC speck formation. These findings provide insights into the mechanisms controlling stability and intracellular trafficking of NLRP3 in human macrophages.**

**Keywords** NLRP3 Stability; Inflammasome Activation; Rab11-FIP2; LPS Priming; Early Endosome
**Subject Categories** Autophagy & Cell Death; Immunology; Membranes & Trafficking

## Introduction

The NOD-, LRR-, and pyrin domain-containing protein 3 (NLRP3) inflammasome is a multimeric protein complex of the innate immune system that activates and controls inflammatory responses in nucleated cells (Broz and Dixit, 2016; Kelley et al, 2019; Lamkanfi and Dixit, 2012; Rathinam et al, 2012). This complex induces pyroptotic cell death and is crucial to the microbial host defence. Canonical activation of the NLRP3 inflammasome is a two-step process which initiates the processing and release of interleukin 1β (IL-1β) and interleukin 18 (IL-18) that do not carry a signal peptide. First, pattern associated molecular patterns (PAMPs) unique to groups of microbes are recognized by pattern recognition receptors (PRR) to induce inflammatory signaling stimulating production of NLRP3 and pro-IL-1β in a process referred to as priming. Lipopolysaccharide (LPS) derived from Gram-negative bacteria is frequently used to prime the NLRP3 inflammasome. The second signal is provided by a wide array of exogenous and endogenous stimuli, such as microbial toxins or infections causing cellular damage. This leads to mitochondrial dysfunction and production of reactive oxygen species, which may cause lysosomal membrane permeability, lysosomal dysfunction and autophagic failure (Zhang et al, 2016). The resulting membrane damage causes ion fluxes that trigger the NLRP3 inflammasome. Nigericin is a potent NLRP3 inflammasome activator that trigger $K^+$ efflux (Munoz-Planillo et al, 2013). NLRP3 oligomerization drives inflammasome activation by binding the apoptosis-associated speck-like protein containing a C-terminal caspase recruitment domain (ASC), to allow ASC speck formation (Feske et al, 2015; Lamkanfi and Dixit, 2014; Martin et al, 2014; Stehlik et al, 2003). The ASC speck serves as a platform for caspase-1-mediated cleavage of pro-IL-1β to mature IL-1β, but is dispensable for caspase-1-mediated cleavage of gasdermin D (GSDMD) and pyroptosis induction (Dick et al, 2016; He et al, 2015).

An important step in NLRP3 activation is translocation to the correct subcellular compartment for the assembly of a functional inflammasome complex. In macrophages, NLRP3 has been reported to localize to mitochondria-associated endoplasmic reticulum membranes (MAMs) (Subramanian et al, 2013; Zhou et al, 2011), the microtubule organizing center (Li et al, 2017) and endosomes (Zhang et al, 2023). Others have reported cytosolic activation complexes (Wang et al, 2013). Recently, NLRP3 was shown to translocate to the *trans*-Golgi network (TGN) and onto dispersed TGN (dTGN) structures containing phosphatidylinositol 4-phosphate (PI4P) following treatment with NLRP3 inflammasome inducers such as nigericin (Chen and Chen, 2018; Zhang et al, 2023). It was proposed that the dTGN could serve as a scaffold for NLRP3 aggregation and ASC speck formation through ionic bonding between a conserved polybasic region in NLRP3 and the negatively charged PI4P. Now, the NLRP3-positive dTGN have been shown to be Rab5- and EEA1-positive and thus early endosomes (Chen and Chen, 2018; Zhang et al, 2023). It was further shown that the accumulation of PI4P and the TGN resident

[1]Centre of Molecular Inflammation Research, Department of Clinical Molecular Medicine, Norwegian University of Science and Technology, Trondheim, Norway. [2]Clinic of Laboratory Medicine, St. Olavs Hospital, Trondheim, Norway. ✉E-mail: harald.husebye@ntnu.no

proteins was a result of a stop in retrograde trafficking caused by nigericin(Zhang et al, 2023).

Rab11a and Rab11b are small GTPases which serve as master regulators of membrane trafficking and have redundant and overlapping functions (Joseph et al, 2023; Lindsay and McCaffrey, 2004). They are also important for compartmentalization of early endosomes for efficient transport to the TGN (Wilcke et al, 2000). Active GTP-bound Rab11 recruits effector proteins such as the Rab11-family interacting proteins (FIPs), like FIP2, to mediate function (Schafer et al, 2014). We have shown that Rab11 and FIP2 control endosomal LPS signaling (Husebye et al, 2010; Klein et al, 2015; Skjesol et al, 2019; Yurchenko et al, 2018). We and others have shown that Rab11 and FIP2 locate to early endosomes (Klein et al, 2015; Lindsay and McCaffrey, 2004; Skjesol et al, 2019; Zerial and McBride, 2001). Like NLRP3, FIP2 binds PI species including PI4P and locates to the Rab11- and TLR4-positive perinuclear endocytic recycling compartment (ERC) (Husebye et al, 2006; Lindsay and McCaffrey, 2004; Skjesol et al, 2019). In this study, we identified FIP2 as an NLRP3 binding partner that controls intracellular trafficking of NLRP3, promoting inflammasome assembly and activation. FIP2 and its binding partner Rab11b were found to modulate NLRP3 stability during the priming step and thereby regulate its caspase-1-mediated cleavage of pro-IL-1β and GSDMD, and pyroptotic cell death in human macrophages. Together, our data demonstrate that Rab11b and FIP2 control stability, trafficking and membrane docking of NLRP3 at the PI4P-positive early endosome, key events in inflammasome assembly and activation.

# Results

## FIP2 silencing strongly reduces canonical NLRP3 activation

To investigate a role of FIP2 in NLRP3 activation and pyroptosis, THP-1-derived macrophages and human primary macrophages were treated with FIP2 siRNA before LPS priming and nigericin treatment. Cell death was measured by lactate dehydrogenase (LDH) release and IL-1β release by ELISA. In addition, caspase-1-mediated cleavage of pro-IL-1β into IL-1β, pro-capse-1 into caspase-1 p20 and GSDMD into the p31 pore forming fragment were measured in cell lysates by immunoblotting. We also blotted for FIP2 to verify the efficiency of FIP2 silencing or level of Flag-FIP2 co-expression. THP-1-derived macrophages silenced for FIP2 showed a 50% reduction in cell death (Fig. 1A). Moreover, IL-1β release was reduced by more than 70% as quantified by ELISA (Fig. 1B). Immunoblotting demonstrated that FIP2 silencing markedly reduced caspase-1 autocleavage giving caspase-1 p20, caspase-1-mediated cleavage of pro-IL-1β to IL-1β p17, and GSDMD to the p31 pore forming form (Fig.1C). Also, the FIP2-silenced cells largely showed a reduction in pro-IL-1β, while pro-caspase-1 and full-length GSDMD were unchanged. Next, we made THP-1 cells stably expressing lentiviral Flag-FIP2 (pLVX-FIP2) or Flag-Empty (pLVX-Empty). Following LPS priming and nigericin treatment, the THP-1-derived-macrophages expressing Flag-FIP2 showed a more than 100% increase in cell death and IL-1β secretion, when compared to those expressing Flag-Empty (Fig. 1D,E).

Accordingly, a marked increase in IL-1β p17 and GSDMD p31 was observed the Flag-FIP2 expressing cells that could a result from higher caspase-1 activity, as suggested the observed increase in caspase-1 p20 in these cells (Fig. 1F). Also, the elevated pro-IL-1β in the Flag-FIP2 expressing cells could result for FIP2 having a role in LPS priming pro-IL-1β. The THP-1-derived macrophages expressing the vector control showed impaired inflammasome activation, this is probably because the genomic insertion of Lentiviral vector DNA partly could preactivate the transduced cells. The encoded Flag-peptide could also play a role.

When comparing cell death in FIP2 silenced THP-1-derived macrophages and those treated with the NLRP3 inhibitor MCC950, NLRP3 inhibition reduced cell death by 85% while FIP2 silencing by ~50% (Figs. 1A and EV1A). Moreover, NLRP3 inhibition reduced IL-1β release by almost 90% and FIP2 silencing by 75% (Figs. 1B and EV1B). Next, we investigated if FIP2 could have a similar role in primary human macrophages. The effect of FIP2 silencing was less pronounced in primary macrophages but remained statistically significant (Fig. 1G). All donors showed a more than 70% reduction in IL-1β release as measured by ELISA (Fig. 1H). Again, immunoblotting of IL-1β confirmed the results obtained by ELISA. As in the FIP2-silenced THP-1-derived macrophages, there was a marked reduction in GSDMD p31 (Fig. 1I). The FIP2 immunoblot confirmed efficient silencing of FIP2, with no change in pro-caspase-1 and full-length GSDMD.

Given the role of FIP2 in NLRP3 inflammasome activation, we investigated the involvement of the other type I FIPs, Rab11FIP1 (FIP1) and Rab11FIP5 (FIP5). Primary macrophages were silenced for FIP1, FIP2 or FIP5 before measuring cell death. Only the FIP2-silenced cells showed a significant reduction in cell death (Fig. EV1C). Successful silencing of FIP2 and FIP1 was verified by immunoblotting (Fig. EV1D). We could not detect FIP5 by immunoblotting using two different FIP5 antibodies. Instead, the efficiency of FIP5 silencing was determined by RT-qPCR (Fig. 1E). We next monitored IL-1β p17, pro-IL-1β, caspase-1 p20 and pro-caspase-1 protein in supernatants, and we found that the FIP2-silenced cells showed a statistically significant reduction in IL-1β p17 while the FIP1- and FIP5-silenced cells showed no change. Also, while pro-IL-1β was significantly reduced by FIP2 silencing, FIP1 silencing gave significantly elevated pro-IL-1β and FIP5 silencing gave no change (Fig. EV1F–H). Caspase-1 p20 levels were not significantly changed by any of the siRNA treatments but highly variable upon FIP5 silencing (Fig. EV1I). The pro-caspase-1 protein levels were not changed in either FIP2- nor FIP5-silenced macrophages but significantly increased. Successful silencing of FIP2 and FIP1 was verified by immunoblotting (Fig. EV1D). We could not detect FIP5 by immunoblotting using two different FIP5 antibodies. Instead, the efficiency of FIP5 silencing was determined by upon FIP1 silencing. These data demonstrate that FIP2 is the only type I FIP that controls both cell death and IL-1β release following NLRP3 activation.

## FIP2 controls NLRP3 and pro-IL-1β protein stability during LPS priming

Having established a role of FIP2 in NLRP3 inflammasome activation, we next investigated if this could be a result of FIP2 controlling NLRP3 and pro-IL-1β mRNA induction during LPS priming. To our surprise, no significant reductions in neither

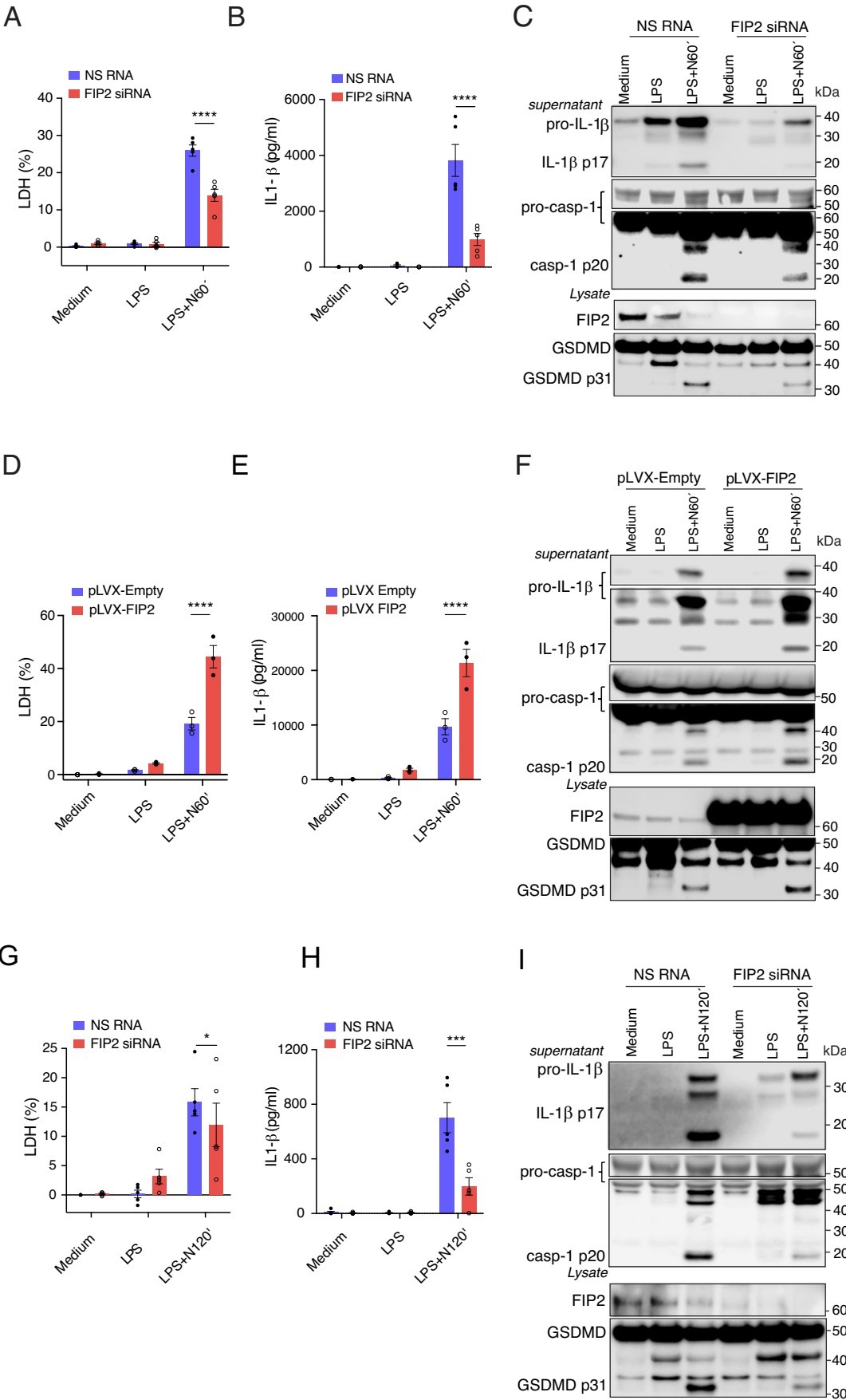

**Figure 1. FIP2 silencing blocks pyroptosis.**

(A) Quantification of LDH release in FIP2-silenced THP-1-derived macrophages, mean between technical replicates of $n = 5$ independent experiments, $P < 0.0001$ (NS RNA vs FIP2 siRNA) after LPS and nigericin. (B) Quantification of IL-1β release by ELISA in FIP2-silenced THP-1-derived macrophages, mean between two technical replicates in $n = 5$ independent experiments, $P < 0.0001$ (NS RNA vs FIP2 siRNA) after LPS and nigericin. (C) Immunoblot of pro-caspase-1 (including a light exposure of the blot), and caspase-1 p20, pro-IL-1β and IL-1β p17 in supernatants, and FIP2 and GSDMD in lysates of FIP2 silenced THP-1-derived macrophages, representative of more than $n = 5$ independent experiments. (D) Quantification of LDH release in FIP2-Flag-expressing THP-1-derived macrophages, the mean between two technical replicates of $n = 3$ independent experiment, $P < 0.0001$ (Flag-Empty- vs Flag-FIP2- expressing cells) after LPS and nigericin. (E) Quantification of IL-1β release by ELISA in FIP2-Flag expressing THP-1-derived macrophages, the mean between technical two replicates of $n = 3$ independent experiments, $P < 0.0001$ (Flag-Empty- vs Flag-FIP2- expressing cells) after LPS and nigericin. (F) Immunoblot of pro-IL-1β, IL-1β p17, pro-caspase-1 (including a light exposure of the blot) and caspase-1 p20 in supernatants, and FIP2 and GSDMD in lysates of Empty-Flag or FIP2-Flag expressing THP-1-derived macrophages, representative of $n = 3$ independent experiments. (G) Quantification of LDH-release in FIP2-silenced human primary macrophages, the mean between two technical replicates in $n = 5$ independent experiments, $P = 0.0266$ (NS RNA vs FIP2 siRNA) after LPS and nigericin. (H) Quantification of IL-1β release by ELISA in FIP2-silenced human primary macrophages, the mean between technical two replicates in $n = 5$ independent experiments, $p = 0.0007$ (NS RNA vs FIP2 siRNA) after LPS and nigericin. (I) Immunoblot of pro-IL-1β, IL-1β p17, pro-caspase-1 (including a light exposure of the blot) and caspase-1 p20 in supernatants, and FIP2 and GSDMD from lysates of in FIP2-silenced human primary human macrophages, representative of more than $n = 5$ independent experiments. The cells were treated with NS RNA (non-silencing RNA) or FIP2 siRNA, before being left untreated or primed with 100 ng/mL LPS for 2 h and treated with 10 μM nigericin as indicated. Data information: In (A, B, D, E, G, H), the results are presented as mean $+/-$ s.e.m. and shown as black bars (two-way ANOVA Tukey's multiple comparisons test with adjusted $P$ values). N Nigericin. Source data are available online for this figure.

NLRP3 nor pro-IL-1β mRNAs were observed in the FIP2-silenced THP-1-derived macrophages (Fig. 2E). In contrast, significant reductions were observed in NLRP3 and pro-IL-1β proteins (Fig. 2A,B). The THP-1- derived macrophages expressing Flag-FIP2 showed an increase in NLRP3 and pro-IL-1β protein by 35% and 100% during LPS priming, respectively (Fig. 2C,D). Here also LPS primed NLRP3- and pro-IL-1β mRNAs were increased (Fig. 2F). Thus, Flag-FIP2 co-expression clearly stabilized both NLRP3- and pro-IL-1β protein during LPS priming. Next, human macrophages were FIP2-silenced to confirm FIP2's role in NLRP3 and pro-IL-1β protein stability during LPS priming. Also here, FIP2 silencing reduced NLRP3- and pro-IL-1β protein (Fig. 2G,H). However, only the reduction in pro-IL-1β protein was found statistically significant. The same was found for pro-IL-1β mRNA (Fig. 2I). Although NLRP3 protein levels were reduced by 50%, FIP2 silencing did not affect NLRP3 mRNA levels.

To verify the specificity of the FIP2 antibody used for immunoblotting, we used THP-1 control cells expressing non-targeting RNA guide (Skjesol et al, 2019), and two cell lines made deficient in FIP2 expression by FIP2-specific guide RNAs. The cells were lysed and immunoblotted with the FIP2 antibody used for western blotting. No FIP2 signal could be detected in the FIP2 knockout (KO) cell lines (Appendix Fig. S1A). Next, we investigated the specificity of the NLRP3 antibody used. Wild-type THP-1 and NLRP3 KO cells were left unstimulated or stimulated with LPS for 2 h, before lysis and western blotting using the NLRP3 antibody for detection (Appendix Fig. S1B). In the unstimulated wild-type THP-1-derived macrophages, a unique NLRP3 band was markedly enhanced upon LPS stimulation. This was not seen in the NLRP3 KO cells, verifying the specificity of the NLRP3 antibody used. We here demonstrate that FIP2 controls NLRP3 and pro-IL-1β protein stability during LPS priming, without significantly affecting NLRP3 and pro-IL-1β mRNA induction in THP-1-derived macrophages.

## Rab11b controls NLRP3-stimulated cell death and IL-1β secretion by regulating NLRP3 and pro-IL-1β protein levels

Having established the involvement of FIP2 in stabilizing NLRP3 and pro-IL-1β during inflammasome priming, we next investigated

the role of the Rab11 GTPases Rab11a and Rab11b. Like the FIP2-silenced cells, the Rab11b-silenced THP-1-derived macrophages showed a more than 50% reduction in cell death (Fig. 3A), and IL-1β release measured by ELISA by showed a reduction close to 50% while the Rab11a-silenced cells showed no significant reduction (Fig. 3B). When investigating the amount of IL-1β p17 by western blotting, the Rab11a-silenced cells showed markedly increased IL-1β p17 while in the Rab11b-silenced cells it was hardly detectable (Fig. 3C). As seen with IL-1β p17, caspase-1 p20 was also increased in Rab11a-silenced cells and hardly detectable in the Rab11b-silenced cells. In line with these results, Rab11a silencing also gave elevated GSDMD p31, while Rab11b silencing gave a marked reduction. Similar results were observed in primary human macrophages where Rab11a silencing resulted in higher IL-1β p17 (Fig. 3D). However, both caspase-1 p20 and GSDMD p31 were moderately reduced. In the Rab11b-silenced cells, IL-1β p17 and caspase-1 p20 as well as GSDMD p31 were both reduced (Fig. 3D). Neither the pro-caspase-1 nor GSDMD full length protein were affected by Rab11a or Rab11b silencing in THP-1 cells or primary human macrophages (Fig. 3C,D). We cannot explain the difference in caspase-1 activity giving high IL-1β p20, and low GSDMD p31 and caspase-1 autocleavage in the Rab11a silenced cells (Fig. 3D).

The efficiency of the Rab11a- and Rab11b silencing in THP-1-derived macrophages was verified by RT-qPCR and showed an equal 95% silencing efficiency (Fig. EV2A). Primary human macrophages displayed a silencing efficiency of 90% for Rab11a and 92% for Rab11b, respectively (Fig. EV2B). Successful silencing of Rab11a in THP-1-derived macrophages and primary human macrophages was also verified by western blotting (Fig. EV2C,D). Although we used four different antibodies for Rab11b, we could not verify Rab11b silencing by western blotting. Next, we investigated if Rab11a and Rab11b could control LPS-primed NLRP3 and pro-IL-1β protein and mRNA stability. Interestingly, Rab11b- but not Rab11a-silencing gave significantly decreased amounts of NLRP3- and pro-IL-1β protein (Fig. 3E–G). As observed for FIP2 silencing, Rab11b silencing did not significantly reduce neither LPS-primed NLRP3- nor pro-IL-1β mRNAs (Fig. 3H,I). Together, these results demonstrate that Rab11b silencing mimics the effect of FIP2 silencing on NLRP3 and pro-IL-1β protein stability during LPS priming. It is therefore likely that Rab11b and with FIP2 regulate NLRP3-stimulated cell death and

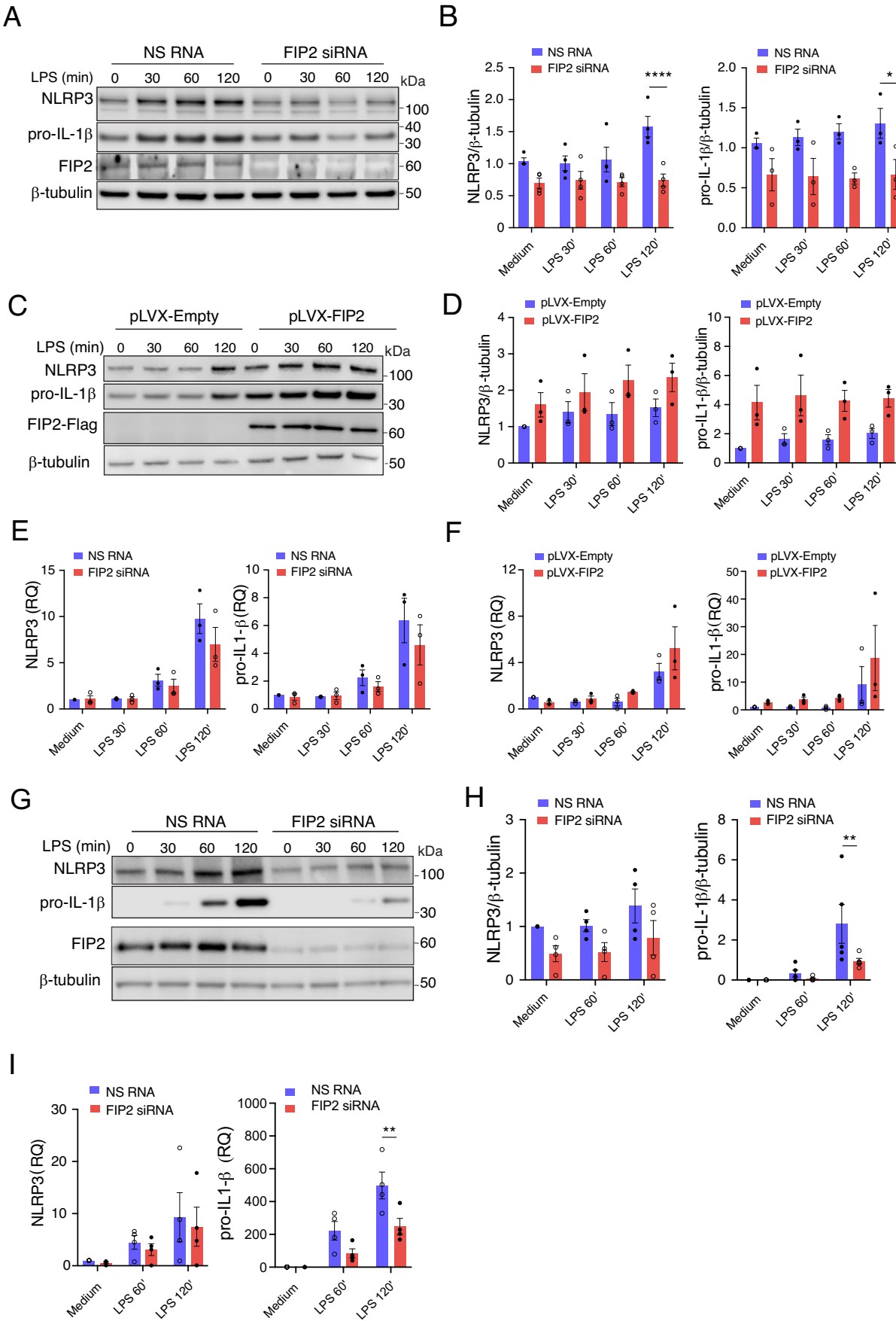

**Figure 2. FIP2 controls NLRP3 and pro-IL-1β stability.**

(A) Immunoblot of NLRP3, pro-IL1-β, FIP2 and β-tubulin in NS RNA (non-silencing RNA) treated or FIP2 siRNA-treated THP-1-derived macrophages stimulated as indicated. (B) Quantification of NLRP3 and pro-IL-1β protein in NS RNA and FIP2 siRNA treated cells, in $n = 4$ (B, left) and $n = 3$ (B, right) independent experiments, $P < 0.0001$ (NS RNA vs FIP2 siRNA, left panel) and $P = 0.0040$ (NS RNA vs FIP2 siRNA, right panel) after 120 min of LPS. (C) Immunoblot of NLRP3 and pro-IL-1β in pLVX-Empty and Flag-FIP2 expressing THP-1-derived macrophages stimulated as indicated. (D) Quantification of NLRP3 and pro-IL-1β protein in NS RNA) and FIP2 siRNA-treated cells, $n = 3$ independent experiments. (E) NLRP3 and pro-IL-1β mRNA in the cells from (B). (F) NLRP3 and pro-IL-1β mRNA in the cells from (D). (G) Quantification of ASC in NS RNA and FIP2 siRNA-treated cells stimulated as indicated, $n = 3$ independent experiments. (H) Quantification of ASC in pLVX-Empty and Flag-FIP2 expressing THP-1-derived macrophages stimulated as indicated, $n = 3$ independent experiments, $P = 0.0104$ (NS RNA vs FIP2 siRNA) after 120 min of LPS. (I) NLRP3 and pro-IL-1β mRNA in the cells from (H), $P = 0.0040$ (NS RNA vs FIP2 siRNA) after 120 min of LPS. Data information: In (B, D, E, F, H, I), data are presented as mean +/− s.e.m. and shown as black bars (two-way ANOVA Tukey's multiple comparisons test with adjusted $P$ values). N Nigericin. Source data are available online for this figure.

IL-1β secretion by stabilization of NLRP3 during the priming process.

## FIP2 co-localizes with ASC specks and is needed for its formation

As FIP2 silencing could reduce NLRP3 inflammasome activation, we next investigated if FIP2 silencing could also reduce ASC speck formation. Human primary macrophages were treated with LPS and nigericin as indicated and co-stained with antibodies towards ASC and FIP2, or ASC and NLRP3. A clear co-localization of both FIP2 and NLRP3 with ASC was seen at the ASC speck (Fig. 4A,B). To quantify FIP2 and NLRP3 at the ASC speck, the spot function of IMARIS 8.2 was used to define the ASC speck in Z-stacks from fixed cells immunostained for ASC and FIP2 or NLRP3. The number of ASC specks formed were markedly reduced by FIP2 silencing (Fig. 4C), as was the amount of FIP2 on the ASC-specks where FIP2 voxel intensities were reduced to background (Fig. 4D). In contrast, the ASC-specks showed a relatively constant level of NLRP3 (Fig. 4E). Also, in the THP-1-derived macrophages both FIP2 and NLRP3 co-localized with ASC on the ASC-speck (Fig. EV3A,B). Also, here the FIP2-silenced cells showed a statistically significant reduction in ASC-specks (Fig. EV3C). Interestingly, when formed the ASC-specks showed a relatively constant level of NLRP3 and ASC (Fig. EV3D; Appendix Fig. S2A). To verify the specificity of the ASC antibody used for detection of specks, ASC-deficient THP-1-derived macrophages were used. As expected, no ASC specks were formed and no IL-1β was secreted (Fig. EV3E; Appendix Fig. S2B).

In wild-type THP-1-derived macrophages, the mean intensities of ASC and FIP2 proteins measured in the ASC specks were similar at both time points of nigericin treatment, as was the mean FIP2 intensity (Fig. EV3E,F). To verify the NLRP3 involvement in ASC speck formation we treated LPS primed wild type and NLRP3 KO THP-1 cells with nigericin and quantified ASC specks and the intensities of ASC and NLRP3 as indicated (Fig. EV3G–I). Together these results demonstrate that FIP2 is needed for ASC speck formation and that FIP2 like NLRP3 locate to the speck.

## NLRP3 binds FIP2 via its KMKK motif

Since FIP2 had a strong effect on NLRP3 stability and subsequent inflammasome activation, we next investigated if FIP2 could bind to NLRP3 and ASC. HEK293T cells were co-transfected with Flag-NLRP3 and EGFP-FIP2, with or without ECFP-Rab11. Anti-Flag-agarose-pulldowns showed that Flag-NLRP3 co-precipitated EGFP-FIP2, both in the absence and presence of Rab11 (Fig. 5A).

However, HEK293T cells have endogenous Rab11 that could contribute to interaction in cells not co-expressing ECFP-Rab11. Next, we investigated if Flag-FIP2 could co-precipitate with NLRP3 and ASC. HEK293T cells were co-transfected with Flag-FIP2, ECFP-Rab11, EGFP-NLRP3 and HA-ASC. Indeed, the Flag-pulldowns showed that Flag-FIP2 co-precipitated EGFP-NLRP3 and Rab11, but not HA-ASC (Fig. 5B). It has been shown that murine NLRP3 binds PI4P through its KKKK-motif, and when mutated failed to give inflammasome activation (Chen and Chen, 2018). The murine NLRP3 KKKK-motif resembles the human NLRP3 KMKK-motif that is located at amino acids position 131 to 134. We next made a series of Flag-NLRP3 deletion mutants to locate the FIP2 binding site in the human NLRP3. The full length NLRP3 and the NLRP3 mutants, NLRP3$_{132-1036}$ and NLRP3$_{152-1036}$ were co-expressed with EGFP-FIP2 in HEK293T cells. The pulldown of the Flag-NLRP3 variants showed that NLRP3$_{132-1036}$, lacking the first lysine in the KMKK motif showed no binding to EGF-FIP2 (Fig. 5C). The other mutant, NLRP3$_{152-1036}$ showed approximately 15% of the binding observed for the wild-type NLRP3 when normalized to the Flag intensity of the respective pulldown The Flag-NLRP3 variants and their FIP2-binding are summarized in Fig. 5D. To verify the involvement of NLRP3 KMKK-motif in FIP2 binding, we next made a NLRP3 mutant where the KMKK-motif was mutated to AAAA, NLRP3$_{AAAA}$. When comparing the Flag-NLRP3 and Flag NLRP3$_{AAAA}$ pulldowns by immunoblotting, NLRP3$_{AAAA}$ showed almost no binding to FIP2 compared to the NLRP3 full-length (Fig. 5E). We also included FIP2I481E, a mutant defective in Rab11 binding (Jagoe et al, 2006). Indeed, the pulldowns of FIP2I481E showed that Flag-NLRP3 did not bind FIP2I481E, suggesting that also Rab11 contributes to FIP2's binding to NLRP3 (Fig. 5E). In Fig. 5F, the main motifs of Flag-NLRP3 and EGFP-FIP2 with putative NLRP3 and FIP2 binding are schematically summarized. To investigate what part of FIP2 that is responsible for NLRP3 binding, we next did pulldowns with full-length FIP2 and the FIP2 deletion mutants, Flag-FIP2$_{1-192}$ and Flag-FIP2$_{193-512}$ (Fig. 5F). Interestingly, while the N-terminal FIP2$_{1-192}$ variant containing the C2-domain showed strong binding, the C-terminal FIP2$_{193-512}$ deletion mutant showed no binding (Fig. 5G). Schematics of the Flag-FIP2 variants used and their capability of binding EGFP-NLRP3 are shown in Fig. 5H. Together these results show that the KMKK motif in NLRP3 is the FIP2 binding site and that the NLRP3 binding motif in FIP2 is in the N-terminal FIP2 C2-domain. Furthermore, our data suggest that NLRP3 can be found in complex with FIP2 and Rab11, and that binding of Rab11 to FIP2 is required for interaction of FIP2 with NLRP3.

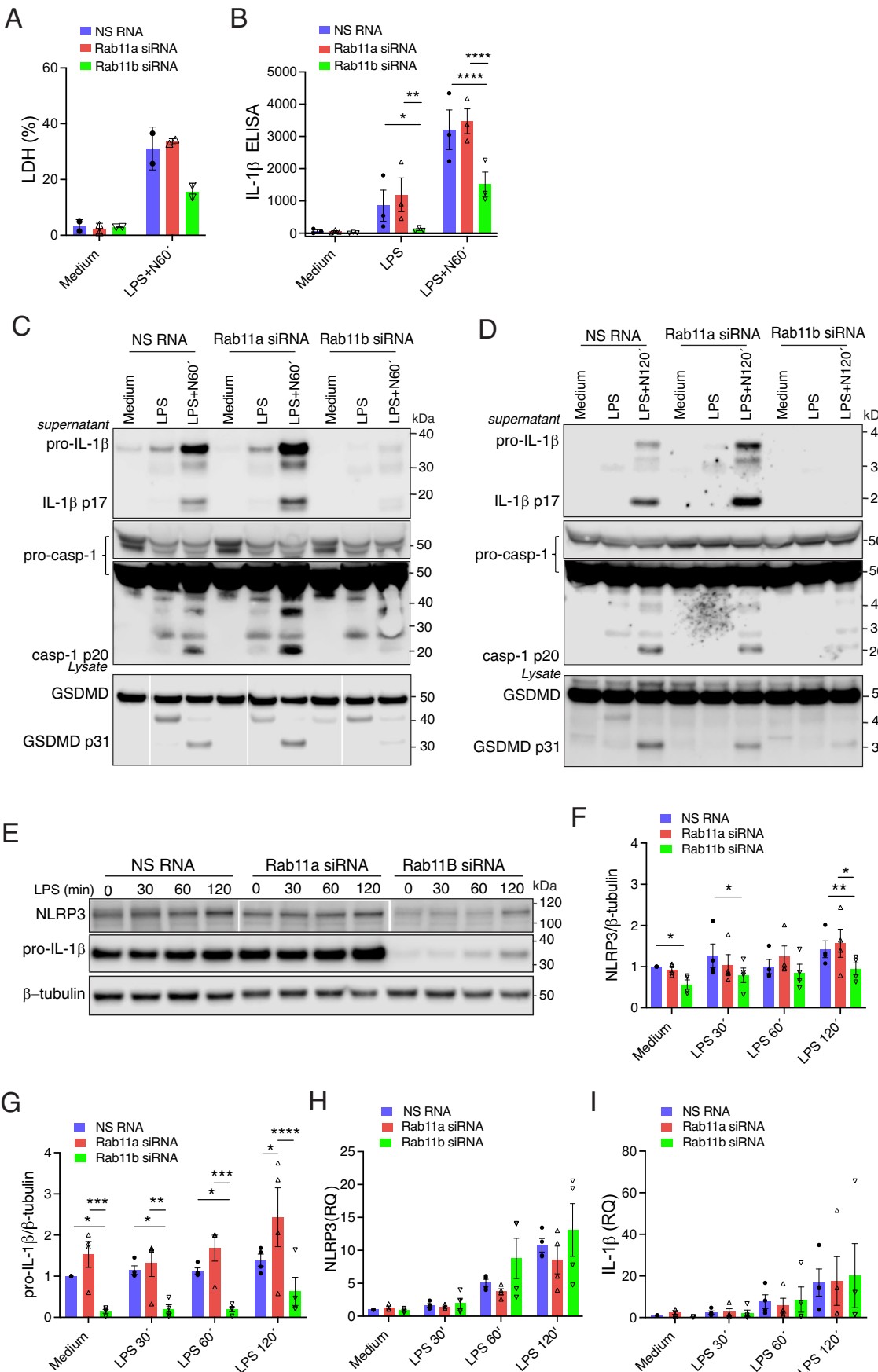

◄ **Figure 3.   Rab11b controls both cell death and IL-1β secretion.**

(A) Quantification of LDH release in Rab11a or Rab11b-silenced THP-1-derived macrophages stimulated as indicated, the mean between two technical replicates in $n = 2$ independent experiments. (B) Quantification of IL-1β release by ELISA in Rab11a- or Rab11b-silenced THP-1-derived macrophages in supernatants, mean between two technical replicates in $n = 3$ independent experiments, $P = 0.0326$ (NS RNA vs Rab11b siRNA) and $P = 0.0030$ (Rab11a siRNA vs Rab11b siRNA) after 120 min LPS; $P < 0.0001$ (NS RNA and Rab11b siRNA) and (Rab11a siRNA vs Rab11b siRNA) after LPS and nigericin. (C) Immunoblot of pro-IL-1β, IL-1β p17, pro-caspase-1 (including light exposure of the blot), caspase-1 p20 in supernatants from Rab11a- and Rab11b-silenced THP-1-derived macrophages; and GSDMD and β-tubulin in lysates from the same experiment, one representative experiment of $n = 3$. (D) Immunoblot of pro-IL-1β, IL-1β p17, pro-caspase-1 (including light exposure of the blot), caspase-1 p20 in supernatants from Rab11a and Rab11b silenced primary human macrophages; and GSDMD in lysates from the same donor, one representative experiment of $n = 3$. (E) Immunoblot of NLRP3, pro-IL1-β, FIP2 and β-tubulin in THP-1-derived macrophages treated with NS RNA, Rab11a siRNA or Rab11b siRNA stimulated as indicated, one representative experiment of $n = 3$. (F) Quantification of NLRP3 protein normalized towards β-tubulin in Rab11a- or Rab11b-silenced THP-1-derived macrophages stimulated as indicated and averaged between technical replicates in $n = 4$ independent experiments. $P = 0.0328$ (NS RNA vs Rab11b siRNA) unstimulated; $P = 0.0200$ (NS RNA vs Rab11b siRNA) after 30 min of LPS; $P = 0.0182$ (NS RNA vs Rab11b siRNA) and $P = 0.0021$ (Rab11a siRNA vs Rab11b siRNA) after 120 min of LPS. (G) Quantification of pro-IL-1β protein normalized towards β-tubulin in Rab11a- or Rab11b-silenced THP-1-derived macrophages stimulated as indicated and averaged between technical replicates in $n = 4$ independent experiments. $P = 0.0417$ (NS RNA vs Rab11b siRNA) and $P = 0.0009$ (Rab11a siRNA vs Rab11b siRNA) in unstimulated cells; $P = 0.0228$ (NS RNA vs Rab11b siRNA) and $P = 0.0065$ (Rab11a siRNA vs Rab11b siRNA) after 30 min of LPS; $P = 0.0244$ (NS RNA vs Rab11b siRNA) and $P = 0.0004$ (Rab11a siRNA vs Rab11b siRNA) after 60 min of LPS and 0.0115 (NS RNA vs Rab11a siRNA) and $P < 0.0001$ (Rab11a siRNA vs Rab11b siRNA) after 120 min of LPS. (H) NLRP3 and pro-IL-1β mRNA in the cells from (F). (I) NLRP3 and pro-IL-1β mRNA in the cells from (G). The cells were treated with NS RNA, Rab11a siRNA or Rab11b siRNA before primed with 100 ng/mL LPS and 5 µM nigericin as indicated. Data information: Data in (A, B, F–I), are presented as mean $+/-$ s.e.m. and shown as black bars (two-way ANOVA Tukey's multiple comparisons test with adjusted $P$ values). N Nigericin. Source data are available online for this figure.

## FIP2 controls NLRP3 translocation to TGN and onto dTGN during inflammasome activation

We next investigated the cellular location of NLRP3 during LPS priming. We have previously shown that TLR4 is enriched in the perinuclear Rab11-positive ERC (Husebye et al, 2010; Klein et al, 2015). Recently, NLRP3 was shown to accumulate in the TGN of murine and human macrophages, and on dTGN structures following nigericin treatment (Andreeva et al, 2021; Chen and Chen, 2018; Nanda et al, 2021; Zhang et al, 2023). As seen in Fig. 6A, there are low amounts of NLRP3 in the TGN of THP-1-derived macrophages before LPS priming and high amounts following LPS priming (Fig. 6B). Quantification of NLRP3 showed that the level of NLRP3 in the TGN was increased by 250% upon LPS-priming in THP-1-derived macrophages (Fig. 6E). FIP2 was also enhanced in the TGN following LPS priming (Appendix Fig. 4G).

Next, we investigated if FIP2 could regulate NLRP3 translocation to the TGN. In the TGN of unstimulated cells the levels of NLRP3 were the same, while in FIP2-silenced THP-1 cells the NLRP3 levels showed a significant reduction of 66%. When comparing the percentage TGN positive for NLRP3, FIP2-silenced cells showed a close to 90% reduction in the perinuclear TGN-ring (Appendix Fig. S3). We next investigated whether treatment of THP-1-derived macrophages with the NLRP3 inflammasome inhibitor, MCC950, could affect NLRP3 in the perinuclear TGN-ring in unstimulated and LPS-primed cells. We found no change in NLRP3 when the levels were compared in unstimulated and LPS-primed cells following MCC950 treatment (Fig. 6F). Higher intensities of FIP2 and NLRP3 in the TGN were also seen following LPS priming of primary human macrophages. In contrast to the THP-1-derived macrophages, the TGN was more tubular and occupied most of the cells in human monocyte-derived macrophages (Appendix Fig. S4A–D).

To investigate if PI4P could have a role in FIP2-mediated recruitment of NLRP3 to the TGN, THP-1-derived macrophages were co-stained for TGN and PI4P using primary antibodies. In resting cells most PI4P was found in the perinuclear TGN-ring (Fig. EV4A). Quantification of PI4P levels in the perinuclear TGN-

ring showed that LPS priming markedly reduced the amounts of PI4P (Fig. EV4A–C). Interestingly, the reduction of PI4P in this compartment was not affected by FIP2 silencing (Fig EV4C). Instead, PI4P appeared as peripheral puncta. These results suggest that FIP2 controls NLRP3 translocation to the perinuclear TGN during LPS priming independent of PI4P.

We next investigated if FIP2 silencing could affect the number of dTGN structures formed during nigericin treatment of LPS-primed THP-1-derived macrophages. Quantification of TGN46-positive dTGN structures in unstimulated and nigericin-treated LPS-primed cells showed that the number of dTGN structures was increased from 0.2 to 4.4 structures per cell, and the increase was statistically significant (Fig. 6G). Interestingly, the FIP2-silenced cells only showed 1.6 dTGN structures per cell, a more than 70% reduction that was statistically significant. Moreover, the dTGN structures that were still present showed a statistically significant 80% reduction in NLRP3 levels (Fig. 6H).

In primary human macrophages, we found FIP2- and NLRP3-positive dTGN structures which varied in size from small puncta to larger structures with a defined limiting membrane (Appendix Fig. S4E,F). When quantified, the concentration of FIP2 was increased in the perinuclear TGN following LPS priming (Appendix Fig. S4B,D,G). We also found increased FIP2 concentrations in the dTGN structures formed after nigericin treatment, that was markedly reduced in FIP2-silenced cells (Appendix Fig. S4H,I). These observations support the specificity of the FIP2 antibody used. Together, these results show that the concentration of FIP2 is markedly increased in the perinuclear TGN-ring during LPS priming and on the dTGN structures formed following nigericin treatment, like what was observed for NLRP3. This is consistent with a role of FIP2 in NLRP3 recruitment to both these compartments, which could also contribute to NLRP3 stability.

## NLRP3 and FIP2 are located on endosomes positive for PI4P and Rab5 during inflammasome activation

Given the role of FIP2 in controlling the number of dTGN and the level of NLRP3 on them we next investigated if FIP2 could control the increase in PI4P on early endosomes, reported during NLRP3

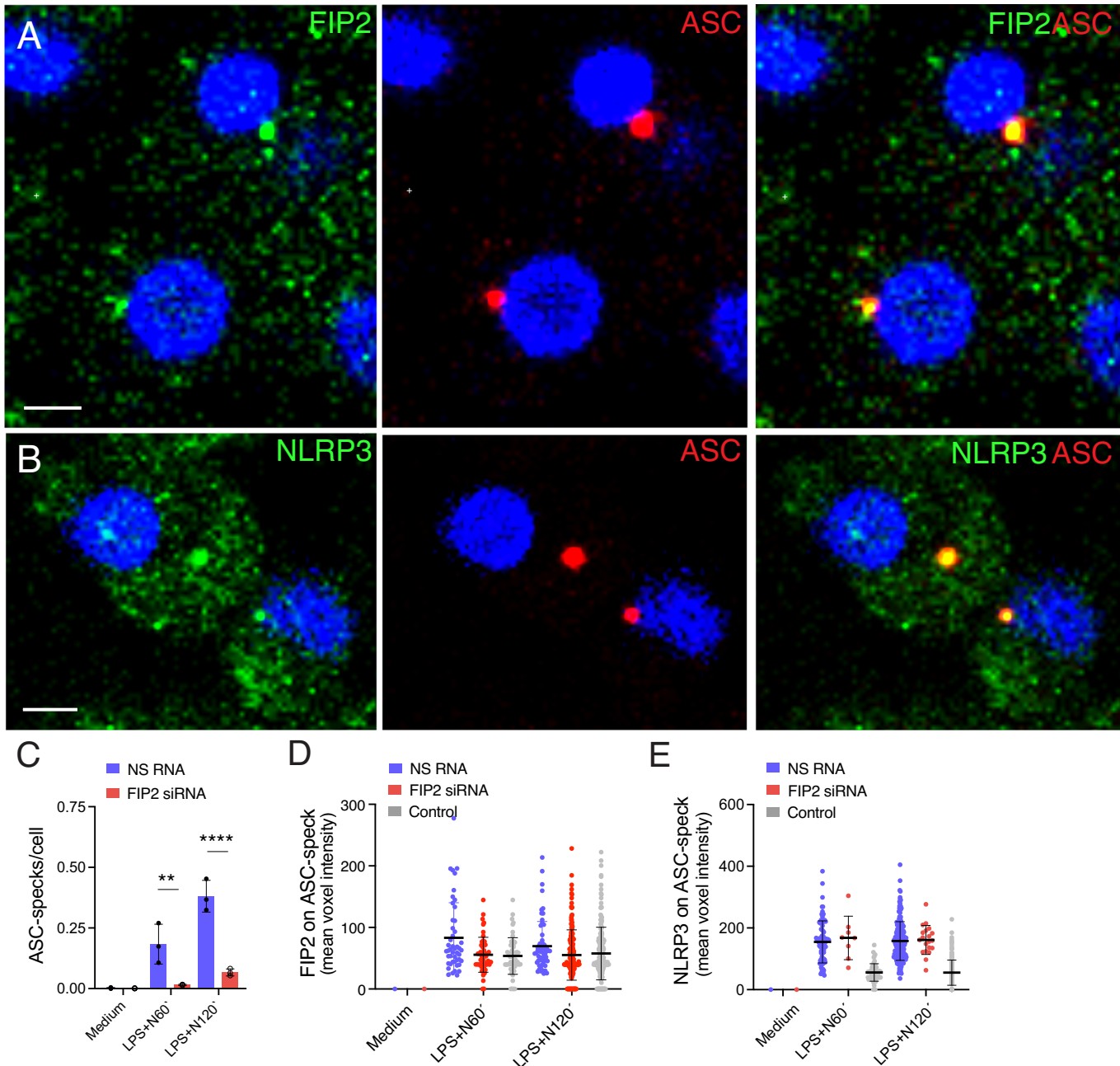

**Figure 4. FIP2 co-localizes with ASC in the ASC speck.**

(**A**) Confocal image showing NLRP3 (green) on ASC speck (red). (**B**) Confocal image showing NLRP3 (green) on ASC speck (red). The THP-1-derived macrophages were treated with NS RNA or FIP2 siRNA before being LPS-primed for 2 h before treatment with 5 µM nigericin for 30 min (**A, B**). (**C**) Quantification of ASC speck formation in FIP2-silenced primary human macrophages, $n = 3$ biological replicates monitoring 206–570 nucleated cells per replicate, $P = 0.0043$ (NS RNA vs FIP2 siRNA) and $P < 0.0001$ (NS RNA vs FIP2 siRNA) after LPS and 60 or 120 min of nigericin. (**D**) Quantification of FIP2 on ASC specks, in one biological replicate monitoring 169–434 nucleated cells of (**C**). (**E**) Quantification of NLRP3 on ASC specks, in one biological replicate of (**C**). The THP-1-derived macrophages and primary macrophages were treated with NS RNA or FIP2 siRNA as indicated. The cells were primed with 100 ng/mL LPS for 2 h and treated with 5 µM nigericin for 1 h or 2 h. ASC specks were identified using the IMARIS 8.2 imaging software on 3-D confocal imaging data. Data information: In (**C–E**), data are presented as mean +/− s.d. and shown as black bars (two-way ANOVA Tukey's multiple comparisons test with adjusted P values). N nigericin. Scale bar = 5 µm. Source data are available online for this figure.

inflammasome activation (Zhang et al, 2023). After nigericin treatment of LPS primed cells, PI4P was found on NLRP3-positive structures, varying in size from small puncta to endosomes with a visual limiting membrane (Fig. 7A). We identified these as enlarged

Rab5 positive early endosomes containing NLRP3 in defined microdomains on the endosomal limiting membrane (Fig. 7B). In line with these observations, we also observed enlarged Rab5 endosomes with PI4P microdomains on the limiting membrane

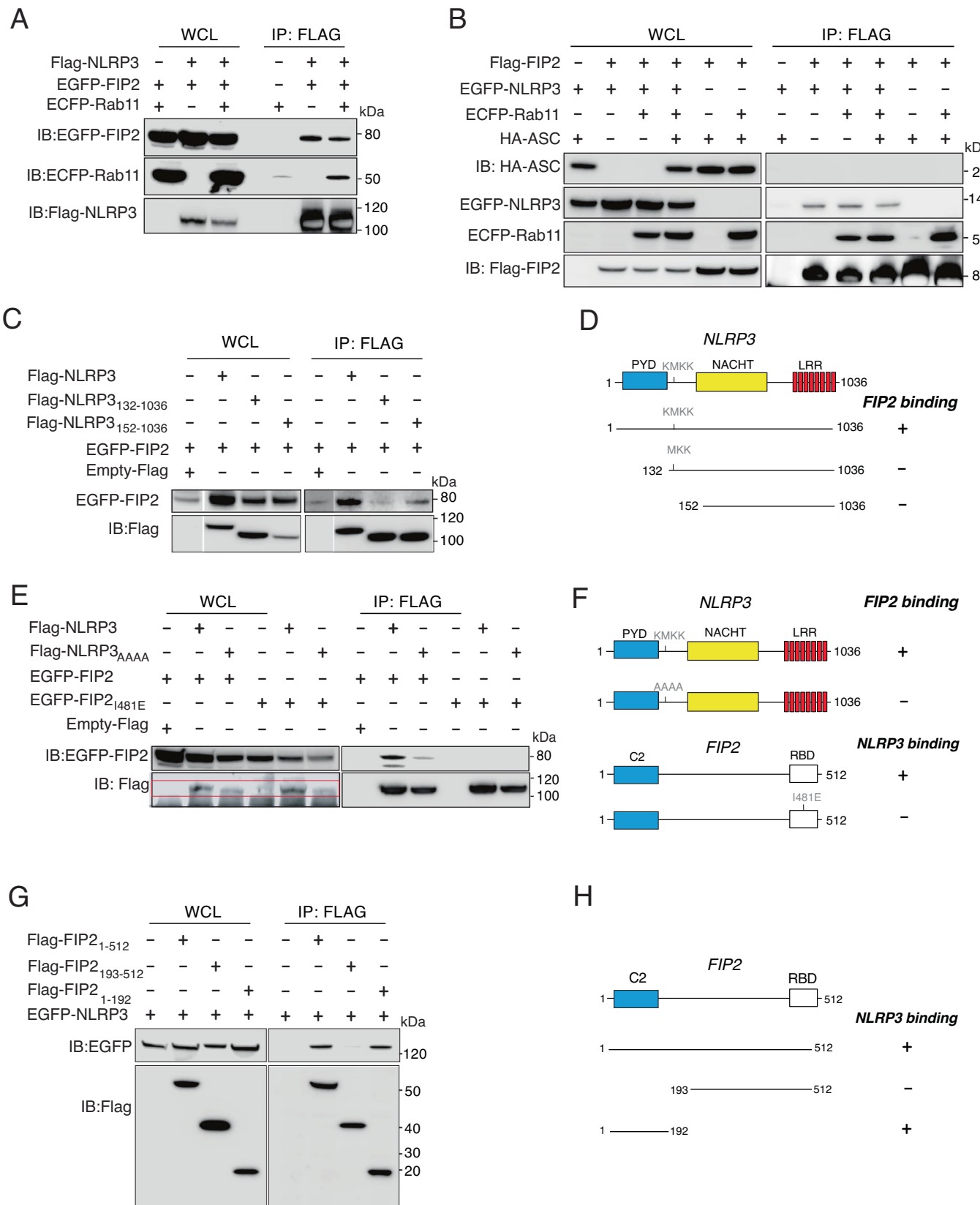

**Figure 5. NLRP3 binds to FIP2 via its KMKK motif.**

(A) Immunoblot of Flag-NLRP3 pulldown in lysates from HEK293T cells co-transfected with Flag-NLRP3, EGFP-FIP2, and ECFP-Rab11. (B) Immunoblot of Flag-FIP2 pulldowns in lysates from HEK293T cells co-transfected with EGFP-NLRP3, ECFP-Rab11, and HA-ASC. (C) Immunoblot of Flag-NLRP3, Flag-NLRP3$_{132-1036}$ and Flag-NLRP3$_{152-1036}$ pulldowns in lysates from HEK293T cells co-transfected with EGFP-FIP2. (D) Schematics mapping the NLRP3 region binding FIP2. (E) Immunoblot of Flag-NLRP3 or Flag-NLRP3$_{AAAA}$ pulldowns from HEK293T cells co-transfected with EGFP-FIP2. (E) Immunoblot of Flag-NLRP3 pulldowns from HEK293T cells co-transfected with EGFP-FIP2 or EGFP-FIP2$_{I481E}$. (F) Schematics of NLRP3 and FIP2 and mutants affecting binding. (G) Immunoblot of Flag-FIP2, Flag-FIP2$_{1-192}$ or Flag-FIP2$_{193-512}$ pulldowns from HEK293T cells co-transfected with EGFP-NLRP3. (H) Schematics mapping the region of FIP2 binding NLRP3. HEK293T cells were transiently transfected with expression plasmids encoding the indicated fusion proteins for 48 h, or 24 (C, E) before immunoprecipitation by anti-Flag affinity agarose. The Empty-Flag-vector were used to ensure equal plasmid DNA amounts. Source data are available online for this figure.

(Fig. 7C). Finally, we found FIP2-positive endosomes with NLRP3 and FIP2 co-localization in microdomains (Fig. 7D). The PI4P-positive puncta are most likely small early endosomes or microdomains on the limiting membrane of enlarged endosomes, as observed for TGN46 on enlarged EEA1 endosomes (Fig. EV5A), PI4P on Rab5 endosomes (Fig. 7C) and for NLRP3 on FIP2 or Rab11b endosomes (Figs. 7D and EV5E). Also, we observed co-localization of NLRP3 and FIP2 in microdomains of enlarged FIP2-positive endosomes in the cells expressing Flag-FIP2 (Fig.7D). In some cases, PI4P also covered larger regions of the limiting membrane (Fig. 7A), as observed for NLRP3 on Rab5-positive endosomes (Fig. 7B). Nigericin treatment of LPS-primed cells increased the number of PI4P-positive endosomes by more than 400% (Fig. 7E), a number that was reduced by more than 70% in FIP2-silenced THP-1-derived macrophages (Fig. 7E, left panel). Also, we observed a 30% reduction in the amount of NLRP3 on the PI4P-positive endosomes (Fig. 7E, right panel). We next investigated if the number of PI4P-positive endosomes was affected by the NLRP3 inhibitor MCC950. If so, it could be indicative of the presence of oligomerized NLRP3 on the PI4P endosomes. Indeed, MCC950 treatment reduced the number of PI4P-positive endosomes by more than 40%, and the reduction was statistically significant (Fig. 7F). In the remaining PI4P-positive endosomes, NLRP3 was reduced by 30%.

To verify the specificity of the NLRP3 antibody used for imaging we next immunostained wild-type and NLRP3 KO THP-1-derived macrophages with the NLRP3-specific antibody used above. In unstimulated wild-type cells, we observed a modest level of NLRP3 staining while in NLRP3 KO cells staining that was markedly weaker (Appendix Fig. S5A,B). Following LPS and nigericin treatment of the wild-type THP-1-derived macrophages resulted in both perinuclear NLRP3 as well as NLRP3-positive endosomal structures that were not observed in the NLRP3 KO cells (Appendix Fig. S5C,D). Instead, weak background staining was observed.

Because NLRP3 activation was also found to be dependent on Rab11b, we used THP-1-derived macrophages co-expressing Flag-Rab11b to investigate if Rab11b was located on Rab5 endosomes. Indeed, Rab11b was frequently found on enlarged Rab5 endosomes following nigericin treatment of LPS-primed cells (Fig. EV5C). The Rab5 endosomes also showed Rab5 and Rab11b co-localization in microdomains on the limiting membrane the enlarged endosomes. The Rab11b-positive endosomes also showed PI4P- and NLRP3-positive microdomains (Figs. EV5D and EV5E). Together, these results indicate that both FIP2 and Rab11 could be involved in the stabilization of NLRP3 on PI4P-positive early endosomes.

## FIP2 controls LPS-stimulated IKKβ activation

IKKβ activation has been reported to be the cue for NLRP3 translocation to the TGN (Nanda et al, 2021; Schmacke et al, 2022). We therefore investigated if FIP2 could affect LPS-stimulated IKKβ activation, suggesting a mechanism whereby FIP2 could drive NLRP3 translocation to the TGN. The TAK1 kinase complex phosphorylates and activates IKKβ (Wang et al, 2001; Zhang et al, 2014). Interestingly, in FIP2-silenced THP-1-derived macrophages the LPS-stimulated phosphorylation of TAK-1 was reduced by 80% at all timepoints of LPS stimulation, while IKKβ activation measured by phosphorylation of IKKβ and IKKα was decreased by 75% (Fig. 8A,B). FIP2 silencing was verified by immunoblotting (Fig. 8A). To confirm the strong effect of TAK1 on LPS-activated IKKβ and IKKα in FIP2-silenced THP-1-derived macrophages, we used the TAK1 inhibitor 5Z-7-Oxozeaenol (5ZO7). TAK1 has been proven instrumental for LPS-stimulated IKKβ activation in NLRP3 inflammasome activation in human macrophages (Nanda et al, 2021; Schmacke et al, 2022). As seen in Fig. 8C,D, TAK1 inhibition has a similar effect on IKKβ activation as observed in the FIP2-silenced cells. Together our results show that TAK1 inhibition blocks NLRP3 inflammasome activation (Fig. 8E–G), resembling our findings in FIP2-silenced THP-1-derived macrophages.

## Discussion

Recruitment of NLRP3 to endosomal membranes is required for proper inflammasome activation. In this work, we demonstrate that FIP2 and its binding partner, Rab11b, are critical regulators of the intracellular location of NLRP3 during LPS priming and inflammasome activation. This conclusion is based on our observations that suppression of FIP2 or Rab11b, but not Rab11a, strongly inhibited caspase-1-mediated cleavage of pro-IL-1β and GSDMD. Furthermore, FIP2 controlled NLRP3 stabilization, and NLRP3 localization to the perinuclear TGN during LPS priming and endosomal membranes guiding ASC speck formation. FIP2 was found to be instrumental for LPS-stimulated IKKβ activation via TAK1, a process recruiting NLRP3 to the perinuclear TGN(Nanda et al, 2021; Schmacke et al, 2022). Interestingly, FIP2 binds NLRP3 via its N-terminal C2-domain, which also binds phosphoinositide species including PI4P (Lindsay and McCaffrey, 2004). The FIP2 binding site in NLRP3 was identified as the PI4P-binding KMKK motif that when mutated abolished binding. Together these results may explain the potent regulatory effect of FIP2 on NLRP3 recruitment to TGN and stabilization on endosomal membranes during inflammasome activation.

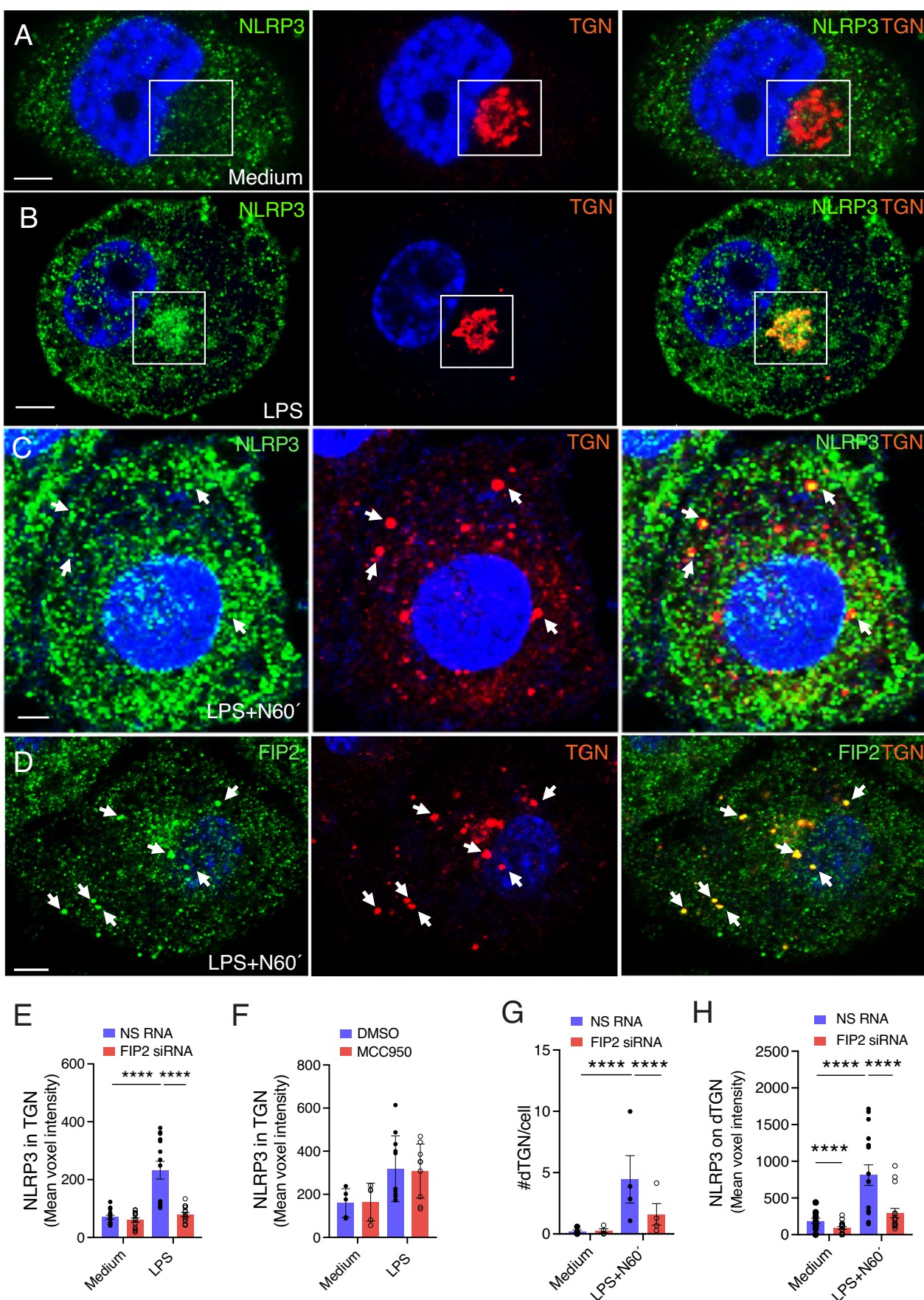

**Figure 6. FIP2 recruits NLRP3 to the TGN during LPS priming and onto endosomal structures after nigericin.**

(A) Confocal image showing NLRP3 (green) and TGN46 (red) in resting THP-1-derived macrophages. (B) Confocal image showing NLRP3 (green) and TGN46 (red) in LPS-primed THP-1-derived macrophages. (C) Confocal images showing NLRP3 in TGN46-positive endosomal structures in THP-1-derived macrophages following nigericin treatment of LPS-primed cells as indicated. (D) Confocal images showing FIP2 in TGN46-positive endosomal structures in THP-1-derived macrophages upon nigericin treatment of LPS-primed cells as indicated. (E) Quantification of NLRP3 in the TGN-ring structure in unstimulated and LPS-primed THP-1-derived macrophages, $n = 4$ independent experiments monitoring in total 339–455 cells pr condition. $P < 0.0001$ before and after LPS stimulation of NS RNA treated cells, $P < 0.0001$ (NS RNA vs FIP2 siRNA) after 120 min of LPS. (F) Quantification of NLRP3 in the TGN46-positive-ring-structure in unstimulated and LPS-primed THP-1-derived macrophages pretreated with DMSO or MCC950 15 min, 55-155 cells monitored in total, $n = 2$ independent experiments. MCC950 were added 30 min prior LPS priming. (G) Quantification number of dTGN-structures in unstimulated and nigericin-treated LPS-primed THP-1-derived macrophages, $n = 4$ independent experiments monitoring in total 339–455 cells pr condition. $P < 0.0001$ before and after LPS and nigericin treatment of NS RNA-treated cells, $P < 0.0001$ (NS RNA vs FIP2 siRNA) after LPS and nigericin treatment. (H) Quantification of NLRP3 in the dTGN structure in unstimulated and nigericin-treated LPS-primed THP-1-derived macrophages, $n = 4$ independent experiments monitoring in total 309–455 cells pr condition. $P < 0.0001$ (NS RNA vs FIP2 siRNA) untreated, $P < 0.0001$ before and after LPS and nigericin treatment of NS RNA-treated cells, $P < 0.0001$ (NS RNA vs FIP2 siRNA) after LPS and nigericin. The TGN46-positive perinuclear ring-structures were identified using the IMARIS 8.2 imaging software on 3-D confocal imaging raw data. MCC950 were added 30 min prior LPS priming (F). The TGN46-positive dTGN structures were identified using the spot function of the IMARIS 8.2 imaging software on 3-D confocal imaging raw data. The THP-1 macrophages were left untreated and LPS-primed for 2 h before being treated with 5 µM nigericin for 1 h. Data information: In (E, G, H), data are presented as mean $+/-$ s.e.m. and in (F) as mean $+/-$ s.d. and shown as black bars (two-way ANOVA, Tukey's multiple comparisons test with adj. $P$ values). N nigericin. Scale bar $= 5$ µm. Source data are available online for this figure.

NLRP3 is by far the most studied inflammasome and is regulated by a diversity of factors (Swanson et al, 2019). Here, we found that Rab11b and FIP2 were essential regulators of the canonical NLRP3 inflammasome activation in human macrophages. Interestingly, Rab11b but not Rab11a, controlled NLRP3 inflammasome stability and subsequent activation, describing a new role of Rab11b and FIP2. We have previously shown that Rab11a controls phagocytosis of *E. coli* and subsequent production of the type I interferon IFN-β (Husebye et al, 2010). Thus, Rab11a and Rab11b may have differential roles in regulating innate immune responses in human macrophages.

The conserved KKKK motif in the polybasic region in NLRP3 in mice was recently shown to bind the negatively charged PI4P, enriched on early endosomes following inflammasome activation (Chen and Chen, 2018). The motif serves as a PI4P-binding domain for NLRP3 required for inflammasome activation in mouse macrophages (Chen and Chen, 2018). In human NLRP3, the second lysine in the KKKK motif is changed to methionine, giving KMKK. Surprisingly, we found that the deletion mutant missing the first lysine in the KMKK-motif was deficient in FIP2 binding. Furthermore, when the KMKK motif was substituted with AAAA, FIP2 binding was lost. Thus, NLRP3 binding to FIP2 clearly depends on the KMKK motif. Also, Rab11 binding to FIP2 seemed to guide NLRP3 binding as a FIP2 mutant carrying a single substitution in its Rab11 binding motif could not bind NLRP3.

The translocation of NLRP3 to the TGN has been reported to be dependent on IKKβ activation, and IKKβ is required for the assembly and activation of the NLRP3 inflammasome (Nanda et al, 2021; Schmacke et al, 2022). IKKβ binds NLRP3 and drives NEK7-independent priming in human macrophages by recruiting NLRP3 to PI4P (Asare et al, 2022; Schmacke et al, 2022). We found that both NLRP3 and FIP2 levels were increased in perinuclear TGN following LPS priming. However, FIP2-mediated translocation of NLRP3 must be independent of PI4P, as TGN resident PI4P was markedly reduced during LPS priming and unchanged by FIP2 silencing. Interestingly, we found that already during the priming stage PI4P was going endosomal.

TAK1 is an important regulator of NLRP3 inflammasome activation (Okada et al, 2014), and TAK1 phosphorylates IKKβ to activate it (Zhang et al, 2014). We confirmed this by using the TAK1 inhibitor 5Z-7-oxozeaenol (5ZO) that completely abolished

TAK1 and IKKβ/IKKα (Fig. 8). In this study, we showed that FIP2 silencing restricted LPS stimulated TAK1 phosphorylation. Furthermore, TAK1 was found instrumental in IKKβ activation and subsequent NLRP3 inflammasome activation as previously reported for human macrophages (Nanda et al, 2021). Thus, we have demonstrated that FIP2 controls the translocation of NLRP3 to the TGN and onto endosomal membranes by a mechanism involving TAK1-mediated phosphorylation of IKKβ, as summarized in the synopsis.

Our data suggest that the NLRP3-FIP2 interaction is important for the translocation of NLRP3 to the TGN and likely NLRP3 stabilization in this compartment. This is supported by our observation where FIP2-silenced THP-1-derived macrophages show similar NLRP3 protein amounts as the NS RNA treated in unstimulated cells. In contrast, when the FIP2-silenced the LPS-primed cells show significantly less NLRP3 protein and therefore FIP2 must have a stabilizing effect largely during LPS priming. Furthermore, FIP2 could also prevent NLRP3 docking on PI4P-positive TGN membranes during priming by blocking PI4P binding. Taken together our data support an important role of FIP2 in TGN NLRP3 protein stability during priming and inflammasome activation.

NLRP3 activators such as nigericin cause K+ efflux and disrupt endosome-TGN retrograde transport resulting in accumulation of *trans*-Golgi markers and PI4P on endosomes (Chen and Chen, 2018; Lee et al, 2023; Zhang et al, 2023). The NLRP3-positive dTGN has previously been shown to be Rab5 and EEA1-positive early endosomes (Zhang et al, 2023), an observation that we confirmed here. Inhibition of retrograde transport has also been shown to cause enlargement of early endosomes in addition to cause endosomal accumulation of accumulation of TGN resident proteins (Huotari and Helenius, 2011). Our results show that FIP2 silencing markedly reduced ASC-speck formation as well as NLRP3 and PI4P levels on early endosomes, pointing to an important role of FIP2. Also, nigericin treatment of LPS primed cells resulted in considerable enlargement of NLRP3- and PI4P-positive endosomes. Interestingly, these enlarged endosomes showed microdomains where both PI4P and NLRP3, as well as FIP2 and NLRP3 showed co-localization. The presence of FIP2 on NLRP3-positive microdomains could point to a mechanism where FIP2 also controls PI4P and NLRP3 location on the endosomal hub

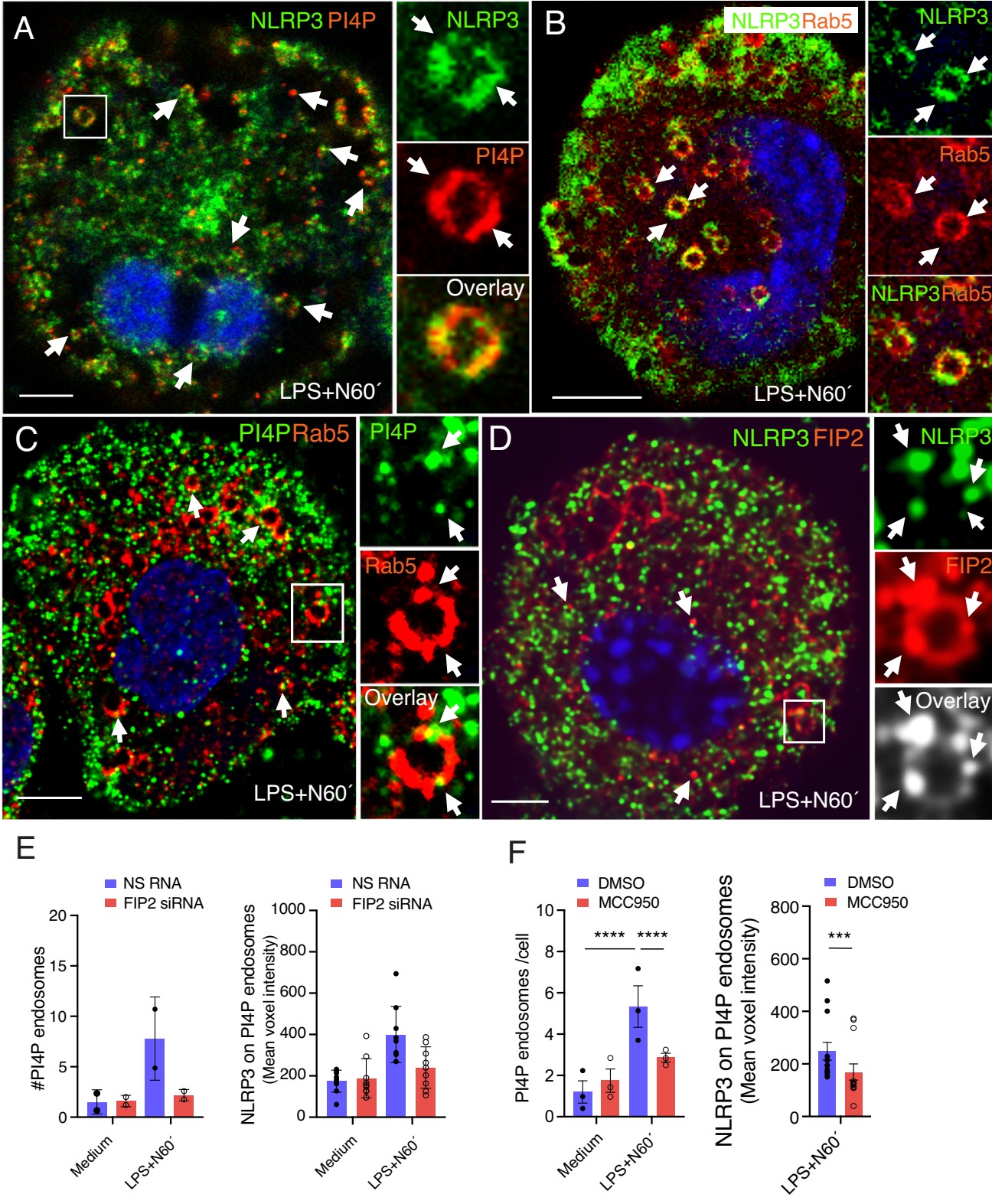

**Figure 7. NLRP3 shifts from the TGN and onto enlarged early endosomes following nigericin treatment.**

(A) Confocal image showing PI4P in the TGN-ring of resting cells. (B) Confocal image showing that PI4P is depleted from the TGN46-positive ring and appear on puncta endosomal structures following LPS priming. (C) Confocal image showing NLRP3 (green) on an enlarged PI4P-positive endosome (red). (D) Confocal image showing PI4P (green) on an enlarged Rab5-coated endosome (red). (E) Quantification of PI4P-positive endosomes and their amount of NLRP3 before and after nigericin treatment of LPS-primed cells, monitoring in total 207–331 cells in $n = 2$ independent experiments. (F) Left panel: quantification of PI4P-positive endosomes and their amount of NLRP3 before and after nigericin of LPS-primed cells, $n = 3$ independent experiments monitoring in total 207–331 cells. $P < 0.0001$ between unstimulated and LPS- and nigericin-treated cells pretreated with DMSO, $P < 0.0001$ between LPS- and nigericin-treated cells pretreated with DMSO or MCC950. Right panel: $P = 0.0001$ between LPS- and nigericin-treated cells pretreated with DMSO or MCC950. The THP-1 cells were treated with FIP2 siRNA or 10 µM MCC950 as indicated before being primed with 100 ng/mL LPS for 2 h and treated with 5 µM Nigericin. MCC950 were added 30 min prior to LPS priming. Data information: In (E), data are presented as mean +/− s.d. and in (F) presented as s.e.m. and shown as black bars—left panel: (two-way ANOVA, Tukey's multiple comparisons test with adj. $P$ values)—right panel: (one sample and Wilcoxon test) right panel. N nigericin. Scale bar = 5 µm. Source data are available online for this figure.

where ASC-speck formation to take place. In addition to cells expressing Flag-FIP2, Flag-Rab11b expressing cells frequently showed enlarged endosomes positive for Rab5, NLRP3 and PI4P. We found that the THP-1-derived macrophages expressing Flag-Rab11b also showed increased NLRP3 activation measured as caspase-1-mediated cleavage of GSDMD. Concomitantly, increased GSDMD cleavage was observed in inflammasome-activated cells. This also supports that NLRP3 inflammasome activation occurs from enlarged endosomes.

The presence of Type II PI4-Kinases has been reported in the Rab11-positive ERC. Rab11 defines the perinuclear ERC that includes the perinuclear TGN, and Rab11 regulates the compartmentalization of early endosomes important for efficient retrograde transport from the endosome to TGN (Wilcke et al, 2000). It is known that PI4-Kinases like PI4KIIα generate endosomal PI4P that regulates receptor sorting at early endosomes(Henmi et al, 2016). Early endosome-to-TGN transport is organized by the retromer complex (Wassmer et al, 2009). At the early endosome, FIP2 have been reported to interact with EHD3, a PI4P binding protein, in such a way that is critical for retrograde transport from the early endosome (Naslavsky et al, 2006). Retromer function also depends on actin-polymerization (Hao et al, 2013). We have previously shown that FIP2 stabilizes Rac1 and CdC42(Skjesol et al, 2019), which is instrumental for actin-cytoskeleton dynamics (Dong et al, 2016). The EHD proteins also regulate early endosome to the lysosome transport. Deficient retrograde trafficking from the early endosome in the FIP2-silenced cells could be a result of TGN46 being missorted for lysosomal degradation. This has been reported for the low-density lipoprotein receptor LDL receptor, mannose-6-phosphate receptor and for TGN46s orthologue TGN38 in Rab11-deficient cells (Hu et al, 2022; Zulkefli et al, 2019). We have not addressed this but active PI4-Kinases like PI4KIIα, generating endosomal PI4P, also accumulate due to the lack of retrograde trafficking (Henmi et al, 2016). We found FIP2 to affect PI4P levels both in the TGN of unstimulated and on early endosomes after inflammasome activation. Therefore, it is likely that FIP2 could also regulate of PI4-Kinase activity in the TGN of resting cells and on the early endosome in the priming events leading to inflammasome activation.

Our data strongly suggests that FIP2 also can control anterograde vesicle trafficking, bringing TGN46 and NLRP3 from the perinuclear TGN to the early sorting endosome. Defective translocation of NLRP3 to the TGN and reduced anterograde transport in FIP2-silenced macrophages could explain the lower abundancy of NLRP3 on the TGN46-positive early endosomes. As FIP2 is also likely to control retrograde trafficking of these proteins

back to the perinuclear TGN our data support that FIP2 together with Rab11b control recycling of NLRP3 between perinuclear TGN and early endosome. Moreover, given the importance of the KMKK in FIP2 binding and the role of FIP2 in NLRP3's stability and location, FIP2 may bind to this domain to guide vesicle transportation and at the same time protect NLRP3 from PI4P binding in TGN. This to ensure NLRP3 to reach the early endosomal compartment before binding PI4P to initiate inflammasome activation. Thus, our data strongly support that vesicle trafficking and tethering of Rab11b-positive FIP2/NLRP3 on the early endosomes is an important for inflammasome assembly and activation as summarized in the synopsis.

In summary, we have uncovered a novel role for FIP2 in controlling NLRP3 inflammasome activation in a Rab11b-specific manner. Our data clearly support a model where nigericin triggers a halt in retrograde trafficking, causing PI4P and NLRP3 to accumulate on early sorting endosomes. We also demonstrated that FIP2 controls TAK1-mediated activation of IKKβ important for NLRP3 translocation to the TGN. To do this, FIP2 binds NLRP3 via the positively charged KMKK motif that reported to control NLRP3 binding to PI4P at the early endosome. These data shows that coordination of membrane traffic by Rab11b and its effector molecule FIP2 and should play an essential role in antimicrobial defense and sterile inflammatory diseases involving the NLRP3 inflammasome.

# Methods

**Reagents and tools table**

| Reagent/resource | Reference or source | Identifier or catalog number |
|---|---|---|
| **Experimental models** | | |
| THP-1 cells | ATTC | ATCC®, TIB-202™ |
| ASCKO THP-1 | Invivogen | thp-dasc |
| FIP2KO THP-1 cells | This study | FIP2 KO3 and FIP2 KO9 |
| NLRP3KO THP-1 cells | Gift from Claire Bryant (Gram et al, 2021) | NLRP3KO |
| PBMCs isolated from human blood (buffy coats) | The blood Bank at St. Olav´s University Hospital | Primary human macrophages |
| 293 T cells | ATTC | CRL-11268™ |

| Reagent/resource | Reference or source | Identifier or catalog number |
|---|---|---|
| **Recombinant DNA** | | |
| pClhsASC-HA | Gift from Kate Fitzgerald (Hornung et al, 2009) | Addgene, 41553 |
| pEGFP C2-NLRP3 | Gift from Christian Stehlik (Khare et al, 2012) | Addgene, 73955 |
| pLVX-EF1α-IRES-Puro-Vector | Takara Bio | 631988 |
| psPAX2 | Gift from Dieter Trono | Addgene, 12260 |
| pMD2.G | Gift from Dieter Trono | Addgene, 12259 |
| LentiCRISPRv2 plasmid | Feng Zhang lab | Addgene, 52961 |
| pEGFP-Rab11FIP2 | Gift from Mary McCaffrey (Lindsay and McCaffrey, 2002) | EGFP-FIP2 |
| pEGFP-Rab11a | Husebye et al, 2010 | |
| Rab11b | ATG:Biosyntetics | |
| pCMV-DYKDDDDK-N vector | Takara Bio | 635688 |
| pLVX-FIP2 | This study | Flag-FIP2 |
| pLVX-Rab11a | This study | Flag-Rab11a |
| pLVX-Rab11b | This study | Flag-Rab11b |
| LentiCRISPRv2 plasmid | Gift from Feng Zhang | Addgene, 52961 |
| Gibson Assembly kit | Thermo Fisher Scientific | A46624 |
| Plasmid sequencing | Eurofins | |
| **Antibodies** | | |
| **Primary antibodies** | | |
| Mouse anti-Caspase-1 (p20) | Adipogen Life Sciences | AG-20B-0048-C100 |
| Rabbit-anti ASC (AL177) | Adipogen Life Sciences | AG-25B0006F-C100 |
| Mouse anti IL-1 beta/IL-1F2 | R&D Systems | MAB201 |
| Rabbit anti-Rab11FIP2 | Abcam | ab180504 |
| Rabbit anti-Rab11FIP2 | Abcam | ab174313 |
| Rabbit anti-β-tubulin | Abcam | ab6046 |
| Goat-anti-NLRP3 (ab4207) | Abcam | Ab4207 |
| Rabbit anti-TGN38 | Abcam | ab50595 |
| Rabbit anti-GFP | Takara | 632592 |
| Mouse anti-GFP | Takara | 632592 |
| Mouse anti-FLAG M2 | Takara | 632381 |
| Goat-anti FIP2 (S-17) | Santo Cruz Biotechnology | sc-163274 |
| Rabbit anti-Gasdermin D | Cell Signalling Technology | #96458 |
| Rabbit anti-Gasdermin D | Cell Signalling Technology | (#97558 |
| Rabbit anti-phospho-TAK1 | Cell Signalling Technology | #4508 |
| Rabbit anti-phospho-IKKα/IKKβ (16A6) | Cell Signalling Technology | #2697 |

| Reagent/resource | Reference or source | Identifier or catalog number |
|---|---|---|
| Rabbit anti-IKKα | Cell Signalling Technology | #2682 |
| Rabbit anti-IKKβ (D30C6) | Cell Signalling Technology | #8943 |
| Rabbit anti-Rab5 (C8B1) | Cell Signalling Technology | #3547 |
| Rabbit anti-TGN46 | Thermo Fisher Scientific | PA5-23068 |
| Mouse anti-Rab5a | Thermo Fisher Scientific | 66339-1-IG |
| Rabbit anti-Rab5 | Thermo Fisher Scientific | PA5-29022 |
| Rabbit anti-EEA1 | Thermo Fisher Scientific | PA5-17228 |
| Anti-PI4P IgM | Echelon Biosciences | Z-P004 |
| **Secondary antibodies** | | |
| Chicken anti-rabbit IgG Alexa Fluor 488 | Thermo Fisher Scientific | A-21443 |
| Chicken anti-rabbit IgG Alexa Fluor 647 | Thermo Fisher Scientific | A-21443 |
| Donkey anti-goat IgG Alexa Fluor 488 | Thermo Fisher Scientific | A-32814 |
| Donkey anti-goat IgG Alexa Fluor Plus 647 | Thermo Fisher Scientific | A-32849 |
| Donkey anti-mouse IgG Alexa Fluor Plus 488 | Thermo Fisher Scientific | A-32766 |
| Donkey anti-mouse IgG Alexa Fluor Plus 647 | Thermo Fisher Scientific | A32787 |
| IgG Alexa Flour 488 | Thermo Fisher Scientific | A-11059 |
| Donkey anti-mouse IgG Alexa Fluor Plus 488 | Thermo Fisher Scientific | A-32814 |
| Rabbit anti-mouse IgG Alexa Fluor 488 | Thermo Fisher Scientific | A-11059 |
| Rabbit anti-mouse IgG Alexa Fluor 647 | Thermo Fisher Scientific | A-21239 |
| Goat anti-mouse IgM | Thermo Fisher Scientific | A-21042 |
| Rabbit anti-mouse IgM FITC | Thermo Fisher Scientific | A-31557 |
| Human TNF-alpha DuoSet ELISA | R&D Systems | DY210-05 |
| Human IL-1 beta/IL-1F2 DuoSet ELISA | R&D Systems | DY201-05 |
| **Oligonucleotides and other sequence-based reagents** | | |
| AllStars Negative Control | Qiagen | SI03650318 |
| Hs_RAB11FIP1_12 | Qiagen | SI02778972 |
| Hs_RAB11FIP2_5 | Qiagen | SI04305672 |
| Hs_RAB11FIP5_5 | Qiagen | SI03246782 |
| Hs_RAB11A_5 | Qiagen | SI04305672 |
| Hs_RAB11B_5 | Qiagen | SI02662695 |
| Fwd primer | 5′-CACCGTGTCCGA GCAAGCCCAAAA-3′ | FIP2 Guide 1 |
| Rev primer | 5′-AAACCTTTTGG GCTTGCTCGGACA-3′ | FIP2 Guide 1 |
| Fwd primer | 5′-CACCGGGACCTG TGCATAACTATA-3′ | FIP2 Guide 2 |
| Rev primer | 5′-AAACTATAGTT ATGCACAGGTCCC-3′ | FIP2 Guide 2 |

| Reagent/resource | Reference or source | Identifier or catalog number |
|---|---|---|
| Flag-NLRP3 1-3108 Fwd | 5′-TATCTCGAGGT ATGAAGATGGCAA GCACCC-3′ | NLRP3 full length |
| Flag-NLRP3 1-3108 Rev | 5′-ATCGCGGCCGCTACCAA GAAGGCTCAAAGA-3′ | NLRP3 full length |
| Flag-NLRP3 121-3108 Fwd | 5′-TATCTCGAGGTATG AAGAAAGATT ACCGTAAGAAG-3′ | Flag-NLRP3$_{121-3108}$ |
| Flag-NLRP3 121-3108 Rev | 5′-ATCGCGGCCGCTAC CAAGAAGGCTCAAAGA-3′ | Flag-NLRP3$_{121-3108}$ |
| Flag-NLRP3 394-3108 Fwd | 5′-TATCTCGAGGTGA CAGGAATGCCCGTCTGGG-3′ | Flag-NLRP3$_{394-3108}$ |
| Flag-NLRP3 394-3108 Rev | 5′-ATCGCGGCCGCT ACCAAGAAGGCTCAAAGA-3′ | Flag-NLRP3$_{394-3108}$ |
| Fwd primer 1 | 5′-GCTTATGGCCAT GCAGGCCCGAATGAAGATG GCAAGCACCC-3′ | NLRP3$_{AAAA}$ mutant |
| Rev primer 1 | 5′-GACCTTTCGAGAATCT CTATTTGTGCAGCGGCGGC AGATTAC-3′ | NLRP3$_{AAAA}$ mutant |
| Fwd primer 2 | 5′-CTATTTGTGCAGCGGC GGCAGATTACCGTAAGA AGTACAGAAAG-3′ | NLRP3$_{AAAA}$ mutant |
| Rev primer 2 | 5′-GGAGATCTCGGTCG ACCGCTACCAAGAAGG CTCAAAGAC-3′ | NLRP3$_{AAAA}$ mutant |
| **Chemicals, enzymes and other reagents** | | |
| K12 LPS | Invivogen | tlrl-eklps |
| nigericin | Invivogen | tlrl-nig |
| 5Z-7-oxozeaenol | Sigma-Aldrich | O9890-1MG |
| L-glutamine | Sigma-Aldrich | G7513 |
| 2-mercaptoethanol | Thermo Fisher Scientific | 31350-010 |
| Phorbol 12-myristate 13-acetate (PMA) | Sigma-Aldrich | P813 |
| Phusion High-Fidelity DNA Polymerase | Thermo Fisher Scientific | F530S |
| Lymphoprep™ | Axis-Shield | AXS-1114547 |
| Pooled A+ serum | St. Olav's University Hospital | A+ serum |
| Recombinant human M-CSF/CSF1 | R&D Systems | 216-MC-025 |
| Lipofectamine™RNAiMax | Thermo Fisher Scientific | 13778075 |
| Lipofectamine™ 3000 | Thermo Fisher Scientific | L3000001 |
| GeneJuice® Transfection Reagent | Merck | 70967-5 |
| PhosSTOP phosphatase inhibitor cocktail | Roche | 4906845001 |
| QIAzol | Qiagen | 79306 |
| RNeady Mini kit | Qiagen | 74106 |
| DNase set | Qiagen | 79256 |
| Endofree plasmid Maxi kit | Qiagen | 12362 |
| High-Capacity RNA-to-cDNA™ Kit | Thermo Fisher Scientific | 4387406 |

| Reagent/resource | Reference or source | Identifier or catalog number |
|---|---|---|
| PerfeCTa qPCR FastMix | Quanta Biosciences | 733-1398 |
| LDH Cytotoxicity Assay Kit | Cayman Chemical | 601170 |
| Anti-Flag (M2) agarose | Sigma- Aldrich | A2220 |
| NuPAGE LDS sample buffer | Thermo Fisher Scientific | NP0007 |
| RPMI-1640 medium | Sigma | R8758 |
| DMEM | BioWhittaker® | 12-604 F |
| Heat-inactivated fetal bovine serum (FBS) | Thermo Fisher Scientific | 10270106 |
| Hank's Balanced Salt solution | Sigma-Aldrich | H9269 |
| EcoRI-HF | New England Biolabs | R3101S |
| XhoI | New England Biolabs | R0146L |
| NotI-HF | New England Biolabs | R3189L |
| CytoFix™ | BD BioSciences | 100412100 |
| Acetone | Sigma-Aldrich | 34850-2.5L-M |
| Methanol | Sigma-Aldrich | 34860-2.5L-R |
| **Other** | | |
| Lenti-X™ Concentrator | Takara | 631232 |
| Nanodrop | Thermo Fisher Scientific | |
| StepOnePlus™ Real-Time PCR cycler | Thermo Fisher Scientific | |
| TCS SP8 Confocal Microscope | Leica Microsystems | |
| Nikon Crest v3 Spinning Disk Confocal Microscope | Nikon Instruments | |
| The Bitplane-IMARIS 8.4.2 | Oxford instruments | Imaris software |
| GraphPad-PRISM | GraphPad | Prism 8.2.1-10.6.1 |
| Odyssey Fc Imager | LI-COR | |
| Image Studios version 5.2 | LICORBio | |
| 24-well glass bottom plates | MatTek | P24G-1.5-13-F |
| µ-Plate 96 Well Square Glass Bottom | Ibidi | 89627 |

## Reagents

Ultrapure K12 LPS (tlrl-eklps) and nigericin (tlrl-nig) were from Invivogen, 5Z-7-oxozeaenol (Sigma-Aldrich, O9890-1MG). The following primary antibodies were used mouse anti-caspase-1 (p20) (Bally-1) (AG-20B-0048-C100) and rabbit-anti ASC (AL177) (AG-25B0006F-C100) were from AdipoGen Life Sciences. Mouse anti IL-1 beta/IL-1F2 (MAB201) and goat anti-mouse IL-1β (AF-401-NA) from R&D Systems. Rabbit anti-Rab11FIP2 [EPR12294-85] (ab180504), rabbit anti-Rab11FIP2 (ab174313), rabbit anti-β-tubulin (ab6046) and goat-anti-NLRP3 (ab4207), rabbit anti-TGN38 (ab50595) were from Abcam. Rabbit anti-GFP (632592)

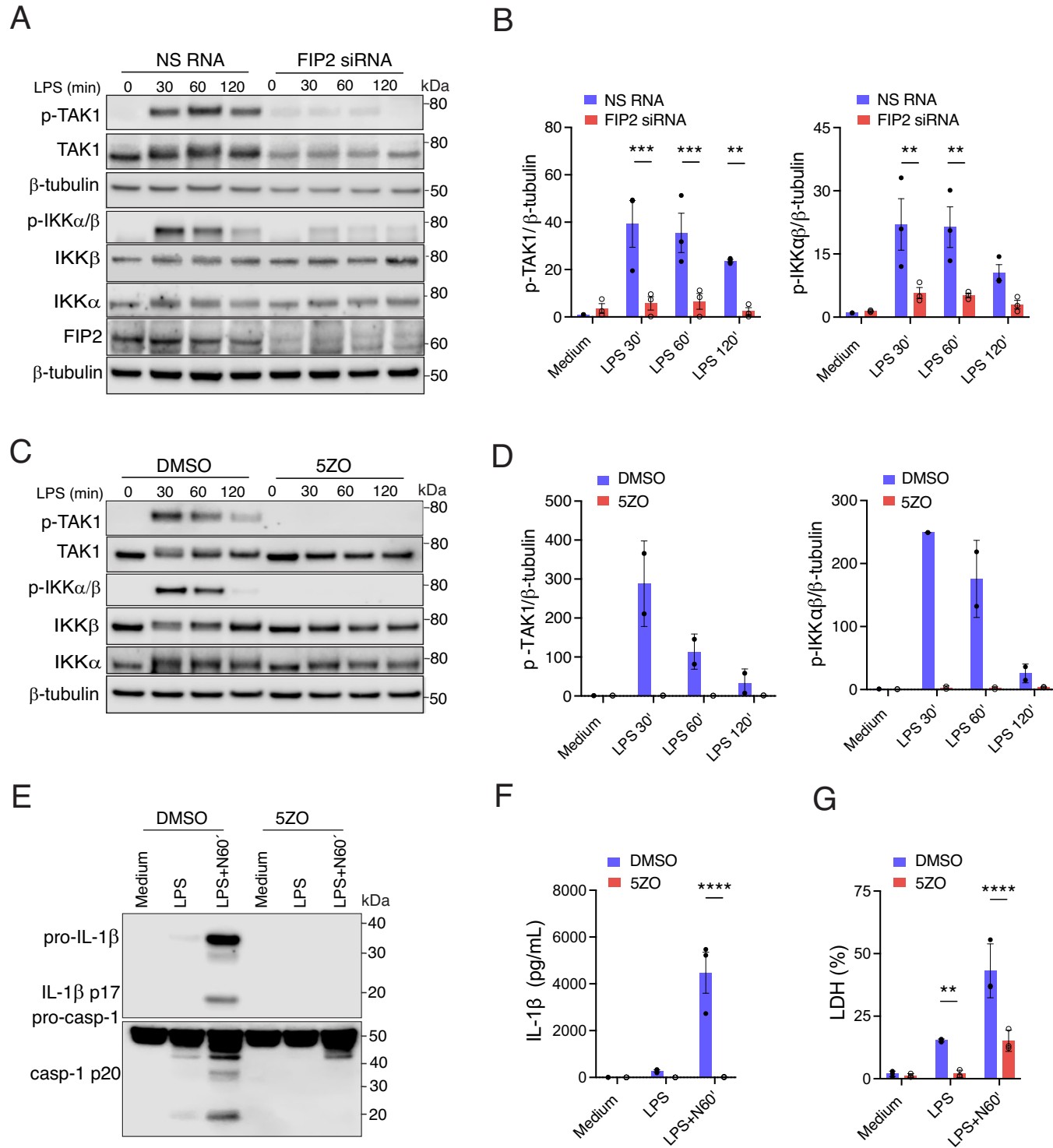

and mouse anti-GFP a (632381) were from Takara Bio. Mouse anti-FLAG M2 was from Sigma-Aldrich. Goat-anti FIP2 (S-17) (sc-163274) and mouse anti-TGN38/46 (sc-166594) were from Santa Cruz Biotechnology. Rabbit anti-Gasdermin D (#96458), rabbit anti-Gasdermin D (#97558), rabbit anti-phospho-TAK1 (#4508), rabbit anti-phospho-IKKα/IKKβ (16A6) (#2697), rabbit anti-IKKα (#2682) and rabbit anti-IKKβ (D30C6) (#8943), rabbit anti-Rab5

(C8B1) (#3547) were from Cell Signaling Technology. Rabbit anti-TGN46 (PA5-23068), mouse anti-Rab5a (66339-1-IG), rabbit anti-Rab5 (PA5-29022) and rabbit anti-EEA1 (PA5-17228) were from Thermo Fisher Scientific. Purified Anti-PI4P IgM (Z-P004) was from Echelon Biosciences. Secondary antibodies used for imaging: Chicken anti-rabbit IgG Alexa Fluor 488 (A-21443), chicken anti-rabbit IgG Alexa Fluor 647 (A-21443), donkey anti-goat IgG

◄ **Figure 8. FIP2 control LPS-stimulated IKKβ activation through TAK1.**

(A) Immunoblots showing LPS-stimulated TAK1 phosphorylation, and IKKβ/IKKβ phosphorylation together with total levels of TAK1, IKKβ, IKKα, FIP2 and β-tubulin. NS RNA or FIP2 siRNAs-treated THP-1-derived macrophages were primed with 100 ng/mL LPS as indicated. (B) Quantification of TAK1 phosphorylation (left panel) and IKKβ phosphorylation (right panel) normalized versus β-tubulin from $n = 3$ independent experiments. $P = 0.0011$ (NS RNA vs FIP2 siRNA) LPS-stimulated for 30 and 60 min. (C) Immunoblots showing LPS-stimulated TAK1 phosphorylation, and IKKβ/IKKβ phosphorylation together with total levels of TAK1, IKKβ, IKKα and β-tubulin THP-1-derived macrophages pretreated with DMSO or 3 μM 5Z-7-Oxozeaenol (5ZO) for 30 min before being primed with 100 ng/mL LPS. (D) Quantification of TAK1 phosphorylation (left panel) and IKKβ phosphorylation (right panel) normalized versus β-tubulin from $n = 2$ independent experiments. (E) Immunoblot of pro-caspase-1, caspase-1 p20, pro-IL-1β and IL-1β p17 in supernatants THP-1-derived macrophages pretreated with DMSO or 5ZO for 30 min before LPS-primed and treated with nigericin as indicated. (F) Quantification of IL-1β release by ELISA, in the mean between technical two replicates in $n = 3$ independent experiments. $P < 0.0001$ (DMSO- vs 5ZO-treated) following nigericin treatment of LPS-primed cells. (G) Quantification of LDH release comparing the mean between technical two replicates in $n = 3$ independent experiments. $P = 0.0051$ (DMSO- vs 5ZO-treated) after LPS priming and $P < 0.0001$ (DMSO- vs 5ZO-treated) after LPS priming and nigericin treatment. THP-1-derived macrophages were primed with 100 ng/mL LPS for 2 h before treatment with 5 μM nigericin as indicated. In (B, F, G), data are presented as mean $+/-$ s.e.m. and in (D) mean $+/-$ s.d. and shown as black bars (two-way ANOVA Tukey's multiple comparisons test with adj. $P$ values). Source data are available online for this figure.

(H + L) Alexa Fluor Plus 488 (A-32814), donkey anti-goat IgG (H + L) Alexa Fluor Plus 647 (A-32849), donkey anti-mouse IgG (H + L) Alexa Fluor Plus 488 (A32766), donkey anti-mouse IgG (H + L) Alexa Fluor Plus 647 (A32787), rabbit anti-mouse IgG Alexa Flour 488 (A-11059), rabbit anti-mouse IgG Alexa Flour 647 (A-21239), goat anti-mouse IgM (A-21042) and rabbit anti-mouse IgM FITC (31557) were from Thermo Fisher Scientific.

## Cells and cell lines

THP-1-derived macrophages (monocytic cell line derived from a patient with acute monocytic leukemia ATCC®, TIB-202™), ASC KO THP-1 (Invivogen, thp-dasc), FIP2 KO THP-1-derived macrophages and NLRP3 KO THP-1-derived macrophages (Gram et al, 2021) were cultured in RPMI-1640 medium (Sigma, R8758) supplemented with 10% heat-inactivated fetal bovine serum (FBS) (Gibco, 10270106), 700 μM L-glutamine (Sigma-Aldrich, Merck G7513) and 5 μM 2-mercaptoethanol (Thermo Fisher Scientific, Gibco™ 31350-010) at 37 °C and 5% $CO_2$. Differentiation into macrophage-like THP-1-derived macrophages was induced by 60 ng/mL phorbol 12-myristate 13-acetate (PMA) (Sigma-Aldrich, P8139). Human peripheral blood mononuclear cells (PBMCs) were isolated from buffy coats from the Blood Bank at St. Olav´s University Hospital (Department of Immunology and Transfusion Medicine) by density gradient centrifugation using Lymphoprep™ (Axis-Shield, AXS-1114547). Blood samples were obtained from healthy volunteers that gave signed consent for experimental procedures, approved by Regional Ethical Committee in Central Norway (# S-04114). Monocytes were isolated by adherence culture plates in RPMI-1640 supplemented with 5% pooled A+ serum (Department of Immunology and Transfusion Medicine, St. Olav´s University Hospital) at 37 °C and 5% $CO_2$ for 45 min before washing three times in Hank´s Balanced Salt solution (Sigma-Aldrich, H9269). Monocytes were further differentiated into macrophages for 10 days in RPMI-1640 supplemented with 10% A+ serum (Department of Laboratory Medicine, Unit of Immunology and Transfusion Medicine, St Olavs University Hospital), 700 μM L-glutamine, recombinant human M-CSF/CSF1 (R&D Systems, 216-MC-025) with a medium change on day 3. HEK293T cells (ATTC, CRL-11268™) were cultured in DMEM (BioWhittaker®, 12-604 F) supplemented with 10% FB and 700 μM L-glutamine at 37 °C with 8% $CO_2$. All cell lines were regularly checked for mycoplasma contamination.

## siRNA treatment

THP-1-derived macrophages were transfected with 16 nM siRNA in Lipofectamine™ RNAiMax (Thermo Fisher Scientific, 13778075) 24 h after seeding in media without antibiotics. Fresh media (without PMA) was added after 48 h, and cells were stimulated after 48 h of rest. Primary macrophages were transfected with 20 nM siRNA on day 6 and 8 after seeding, using Lipofectamine™ 3000 (Thermo Fisher Scientific, L3000001). The AllStars Negative Control siRNA (QIAGEN, SI03650318) was used as a non-silencing control and termed NS RNA. Hs_RAB11FIP1_12 (SI02778972), Hs_RAB11FIP2_5 (SI04305672), Hs_RAB11FIP5_5 (SI03246782), Hs_RAB11A_5 FlexiTube siRNA (SI04305672) and Hs_RAB11B_6 FlexiTube siRNA (SI02662695) were all from Qiagen and used to target human FIP1, FIP2, FIP5, Rab11a and Rab11b, respectively.

## Generation of a stable THP-1 cell line co-expressing Flag-FIP2, Flag-Rab11a or Flag-Rab11b

pLVX-Empty, pLVX-FIP2, pLVX-Rab11a and pLVX-Rab11b was made by PCR-cloning into the pLVX-EF1α-IRES-Puro-Vector (Takara, 631988). The EGFP-Rab11FIP2 (Skjesol et al, 2019), ECFP-Rab11a (Husebye et al, 2010) and synthetic Rab11b DNA (ATG: Biosyntetics) were used as templates. THP-1 sublines expressing lentiviral encoded genes were made by transduction with virus pseudo viral particles assembled in HEK293T cells. Briefly, HEK293T cells were co-transfected with packaging plasmids psPAX2 and pMD2.G (kindly provided by the TronoLab (Addgene plasmids 12260 and 12259)), to produce pseudo viral particles expressing the gene of interest. Supernatants from HEK293T cells were collected at 48 h and 72 h, combined and concentrated using Lenti-X™ Concentrator (Takara, 631232). Subsequently, THP-1-derived macrophages were transduced with the viral particles along with 8 μg/mL protamine sulphate. The transduced cells were then selected with puromycin (1 μg/mL) for 2 weeks, and protein expression verified by western blotting.

## Generation of a stable THP-1 deficient in FIP2 expression

To make the THP-1 FIP2 knock out cell lines a LentiCRISPRv2 plasmid was uses for transduction [PMID: 25075903] (gift from Feng Zhang lab - #52961 Addgene, Watertown, MA, USA). The primers were used for inserting the FIP2 RNA guides into

LentiCRISPRv2 is shown in the Reagents table. For production of pseudo viral particles, the same packaging plasmids and methodology were applied as described in previous section.

## Cloning of Flag NLRP3 full-length and NLRP3 deletion mutants

Phusion High-Fidelity DNA Polymerase (Thermo Fisher Scientific) was used for amplification of NLRP3 gene cDNA using the pEGFP-C2-NLRP3 plasmid as template for making the Flag-NLRP3 variants. The pEGFP-C2-NLRP3 was a gift from Christian Stehlik (Addgene plasmid # 73955; http://n2t.net/addgene:73955; RRID:Addgene_73955). PCR-products, or restricted vectors, were purified by QIAquick PCR purification and gel extraction kits (Qiagen, 28704). Endofree plasmid Maxi kit (Qiagen, 12362) was used for endotoxin-free plasmids preparations. The PCR-products were subcloned into pLVX-EF1α-IRES-Puro vector (Takara Bio, 631988). Sequencing of plasmids was done at Eurofins Genomics. The primers used for cloning of Flag-NLRP3 and deletion mutants are shown in the Reagents table.

## Cloning of the NLRP3$_{AAAA}$ mutant

Two PCR-products were prepared using the full-length Flag-NLRP3$_{1-3108}$ plasmid as template.

The primers used for generation of the NLRP3$_{AAAA}$ mutant are shown in the Reagents table. PCR-products were purified from agarose after being separated by agarose gel electrophoresis (QIAquick PCR Purification Kit, 28106) and combined with cleaved vector for combining DNA fragments using a Gibson Assembly kit (Thermo Fisher Scientific, A46624) following manufacturer recommendations for three DNA fragments. The kit was used to combine the EcoRI-HF (New England Biolabs, R3101S) cleaved empty pCMV-DYKDDDDK-N vector (Takara Bio, 635688) with 2 PCR-products to change the NLRP3 KMKK motif to AAAA. The final ligation mix was transformed into competent *E. coli* DH5α. The final NLRP3$_{AAAA}$ plasmid vector was verified by sequencing (Eurofin Genomics).

## LDH and ELISA assays

Cell culture supernatants were collected and centrifuged at $10,000 \times g$ for 1 min followed by LDH release analysis using LDH Cytotoxicity Assay Kit (Cayman Chemical, 601170) following the manufacturer's instructions. On the day of the experiment medium was changed into medium containing 1% FBS or 1% human A$^+$ serum. This to allow better reliability of the LDH assay. Following stimulations with 100 ng/mL LPS and 5 µM Nigericin 100 µL of lysis buffer was added to the cells and incubated at 37 °C for 30 min, debris was pelleted by centrifugation at $10,000 \times g$ for 1 min. Then 50 µL of the reaction mixture was added to the samples and mixed and the samples incubated at room temperature for 20–30 min before adding 50 µL of stop buffer. IL-1β in supernatants from THP-1-derived macrophages or primary human macrophages were detected using the human IL-1 beta/IL-1F2 DuoSet ELISA (R&D Systems DY201-05) or the human TNF-alpha DuoSet ELISA (R&D Systems, DY210-05).

## Immunofluorescence staining and imaging

For immunofluorescence experiments, $2 \times 10^5$ THP-1-derived macrophages/well or $1.0 \times 10^6$ PBMCs were seeded in 24-well glass bottom plates (MatTek, P24G-1.5-13-F) or µ-Plate 96 Well Square Glass Bottom (Ibidi, 89627). Following stimulation, the supernatant was removed, and the cells were fixed either in 1:1 solution of methanol/acetone at $-20$ °C for a minimum of 20 min followed by rehydration in PBS at RT for 1 h, with 4% paraformaldehyde (PFA) or CytoFix™ (BD BioSciences, 100412100), following the immunostaining protocol previously described. The cells were blocked in 10% A$^+$ in PBS for 20 min at RT and immunostained using primary antibodies at 2–5 µg/mL in 1% A$^+$ serum in PBS. After washing three times for 5 min in PBS the samples were incubated with highly cross-adsorbed fluorescently labeled secondary antibodies at a concentration of 2 µg/mL in 1% A$^+$ serum in PBS. The cells were washed three times in PBS before incubated for nuclear staining with Hoechst stain (1 µg/ml) for 5 min and washed two times in PBS before confocal microscopy. Following fixation all solutions were supplemented with saponin to 0.05% as previously described (Husebye et al, 2010).

Confocal images were captured using a Leica TCS SP8 (Leica Microsystems) confocal microscope equipped with an HC plan-apochromat 63×/1.4 CS2 oil-immersion objective using 488 nm, 561 nm, and 633 nm white laser lines and the 405 nm diode laser for detection. Three-dimensional data was obtained from 12-bit depth imaging data and used to build Z-stacks for individual channels. The white laser intensity was set to 70% as recommended by the manufacturer, and channel fluorescence intensity to 30% or less. Images were also captured on a Nikon Crest v3 Spinning Disk Confocal Microscope equipped using a CFI Plan Apo l D 100X Oil NA 1.45 objective for detection. Imaging was preferentially done between lines and as the same scanning speed and laser intensity for a given set of experiments and with 16-bit depth imaging data. The resulting images enhanced by deconvolution using the NIS-Elements Advanced Research Software Ver. 60.10.01. The Bitplane-IMARIS 8.4.2 software and the 3D-spot function with thresholding were used for quantification giving voxel intensities from imaging raw data. For the ASC speck-detection spot size was 3 µm in *xy* diameter and 4 in *z*-depth, for the detection of endosomes the *xy* diameter was set to 2 and *z*-depth to 3. For the 3-D perinuclear ring structures the spot was set to cover most of the ring diameter varying from 8-12 in xy-diameter while 6-8 in z-depth. Cells without an intact nucleus or not being completely displayed in the Z-stacks were excluded from the analysis done using the IMARIS imaging software. Also, immunofluorescence shown in membrane structures or ASC-specks that were not located in intact cells were excluded. In cells stained by the PI4P antibody showing immunostaining without resolvable cell structures were excluded. The data was presented as mean voxel intensities. Using MATLAB, the software could give numerical values that could be analyzed by GraphPad-PRISM 8.2.1. The PRISM software provides a test for normal distribution. For LDH and ELISA quantifications, the sample size most often was too small to satisfy the requirements of normal distribution. The larger data sets presented in Fig. 6 and Fig. 7 some meet the requirements of normal distribution ($P = 0.05$). Either the parameters for the Shapiro-Wilk or Kolmogorov–Smirnov test was meet. In others 3 out of 4 meet the requirements of normal distribution. PRISM files for the

statistics calculated is provided. Statistical significance was calculated by the Two-way ANOVA multiple comparison tests, reporting multiplicity adjusted *P* values (adj. *P* values) and the results displayed as mean +/− s.e.m. if from three or more independent experiments, or +/− s.d. from less experiments than three and not calculating statistical significance. The statistical test when used is stated in the figure text. Also, a statement of sample sizes has been included in the figure legends.

## Real-time quantitative PCR

In total, $6 \times 10^5$ THP-1-derived macrophages/well or $4.0 \times 10^6$ PBMCs were seeded in six-well plates. Total RNA was isolated from cell lysates in QIAzol (Qiagen, 79306) using chloroform extraction with subsequent purifications on RNeady Mini kit (Qiagen, 74106) following the manufacturer´s protocol. DNA was digested using the DNase set (Qiagen, 79256). RNA concentrations were measured using NanoDrop (Thermo Scientific) and cDNA synthesis was performed using High-Capacity RNA-to-cDNA™ Kit (Thermo Fisher Scientific, 4387406). Real time quantitative real-time PCR (RT-qPCR) was done using the PerfeCTa qPCR FastMix (Quanta Biosciences, 733-1398) in a 20 µL reaction volume in duplicates and cycled in a StepOnePlus™ Real-Time PCR cycler (Applied Biosystems, Thermo Fisher Scientific) on technical replicates. The following TaqMan Gene Expression Assays were used; IL-1β (Hs01555410_m1), NLRP3 (Hs00918082_m1), Rab11FIP2 (Hs00208593_m1), Rab11a (Hs00900539_m1), Rab11b (Hs00188448_m1) and TBP (Hs00427620_m1) all from Applied Biosystems, Thermo Fisher Scientific).

## DNA transfection and co-immunoprecipitation

List of plasmids used: hsNLRP3 in pEGFP-C2 was a gift from Christian Stehlik (Addgene plasmid # 73955; http://n2t.net/addgene:73955; RRID:Addgene_73955) and hsASC-HA in pCI was a gift from Kate Fitzgerald (Addgene plasmid # 41553; http://n2t.net/addgene:41553; RRID:Addgene_41553) and hsRab11a in pECFP-C1 (Husebye et al, 2010). HEK293T cells were seeded at 250,000 cells/well in six-well plates and transfected the following day using Genejuice transfection reagent (Merck, 70967-5). 48 h after transfection, cells were in PBS and harvested in ice-cold lysis buffer (150 mM NaCl, 50 mM Tris-HCl pH 8.0, 1 mM EDTA, 1% NP-40) with added 50 mM NaF, 2 mM $Na_3VO_3$ (Sigma-Aldrich) containing PhosSTOP phosphatase inhibitor cocktail (Roche, 4906845001) and cOmplete mini protease inhibitor cocktail (Roche, 11836153001). Lysates were cleared by centrifugation at $21,000 \times g$ at 4 °C for 15 min, and co-IPs were performed using 50 µL anti-Flag (M2) affinity agarose (Sigma-Aldrich, A2220) on rotation at 4 °C for 3 h or ON. Samples were washed four times and eluted at 80 °C for 7 min in 1× NuPAGE LDS sample buffer (Thermo Fisher Scientific, NP0007), and agarose beads were subsequently removed by high-speed centrifugation. DTT was added to a concentration of 25 mM and samples were heated again (80 °C for 7 min) before being subjected to SDS-PAGE and western blotting.

## SDS-PAGE and western blotting

Protein was precipitated by isopropanol precipitation from the organic phase of Qiazol lysates, after RNA samples had been isolated, and according to manufacturer's instructions (Thermo Fisher Scientific). The protein pellet was air dried and dissolved in 4 M urea in 2% SDS and 1× LDS supplemented with 25 mM DTT before heated at 95 °C dissolved in 1× LDS supplemented with 25 mM DTT for 5 min. Samples were loaded and run on 4–12% Bis-Tris NuPage gels and blotted on nitrocellulose membranes using the iBlot2 system (Thermo Scientific, IB21001). Membranes were blocked in 5% BSA for 1 h at RT before incubation with primary antibodies up to 72 h. Membranes were washed in three times in TBS-T (Tris Buffered Saline with 0.1% Tween-X100) before incubated with HRP-conjugated immunoglobulins for 1 h at RT. The following HRP-conjugated immunoglobulins from Agilent Dako Products were used: Swine anti-rabbit immunoglobulins/HRP (P039901-2), goat anti-mouse immunoglobulins/HRP (P044701-2) and rabbit anti-goat immunoglobulins/HRP (P044901-2). Following three washes in TBS-T the blots were developed using SuperSignal West Femto Substrate (Thermo Scientific, 34095). Images were captured with the Li-COR Odyssey system and analyzed by the Odyssey Image Analysis software Ver 5.2.

## Graphics

Synopsis image was created with BioRender.com.

## Data availability

This study includes no data deposited in external repositories.

The source data of this paper are collected in the following database record: biostudies:S-SCDT-10_1038-S44318-026-00755-7.

## Peer review information

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

## Acknowledgements

Confocal imaging was performed at the Cellular and Molecular Imaging Core Facility (CMIC), Norwegian University of Science and Technology (NTNU). CMIC is funded by the Faculty of Medicine and Health Sciences at NTNU and Central Norway Regional Health Authority. The Research Council of Norway through its Centers of Excellence funding scheme, awarded the grant #223255/F50 to TE, that financed: CG, MY, AS, CC, TE and HH. The Research Council of Norway, FRIMEDBIO program, awarded the grant # 275876 to TE, that financed: CG, MY, AS, CC, SU, ZI, CRD, TE and HH. The Research Council of Norway, FRIPRO program, awarded the grant #334787 to TE, that financed: SK, LR, TE and HH.

## Author contributions

**Caroline S Gravastrand**: Conceptualization; Data curation; Software; Formal analysis; Validation; Investigation; Methodology; Writing—original draft. **Maria Yurchenko**: Formal analysis; Validation; Investigation; Visualization; Methodology; Writing—review and editing. **Stine Kristensen**: Conceptualization; Formal analysis; Validation; Investigation; Visualization; Methodology; Writing—review and editing. **Astrid Skjesol**: Data curation; Formal analysis; Validation; Investigation; Visualization; Methodology. **Carmen Chen**: Data curation; Validation; Investigation; Methodology. **Sindre Ullmann**: Software; Formal analysis; Validation; Investigation; Visualization; Methodology. **Zunaira Iqbal**: Formal analysis; Validation; Investigation; Visualization; Methodology. **Karoline Ruud Dahlen**: Validation; Investigation; Visualization; Methodology. **Kashif Rasheed**: Formal analysis; Validation; Methodology. **Unni Nonstad**: Methodology. **Liv Ryan**: Data curation; Formal analysis; Methodology. **Terje Espevik**: Conceptualization; Resources; Data curation; Supervision; Funding acquisition; Validation; Investigation; Writing—original draft; Project administration; Writing—review and editing. **Harald Husebye**: Conceptualization; Data curation; Software; Formal analysis; Supervision; Validation; Investigation; Visualization; Methodology; Writing—original draft; Project administration; Writing—review and editing.

Source data underlying figure panels in this paper may have individual authorship assigned. Where available, figure panel/source data authorship is listed in the following database record: biostudies:S-SCDT-10_1038-S44318-026-00755-7.

## Funding

 Olavs Hospital - Trondheim University Hospital).

## Disclosure and competing interests statement

The authors declare no competing interests.

# Expanded View Figures

**Figure EV1.   FIP2 is the driver of caspase-1-mediated IL-1β cleavage of the class I FIPs.**

(A) Quantification of LDH release in THP-1-derived macrophages treated with DMSO or NLRP3 inhibitor MCC950 before LPS priming and nigericin treatment, $n = 2$ independent experiments. (B) IL-1β ELISA from the cell supernatants in (A). (C) Quantification of LDH release in primary human macrophages treated with NS RNA, FIP1 siRNA, FIP2 siRNA or FIP5 siRNA between technical replicates of macrophages from $n = 3$ human donors, $P = 0.0022$ (NS RNA vs FIP2 siRNA) in LPS-primed and nigericin-treated cells. (D) Immunoblots of FIP2, FIP1 and β-tubulin in NS RNA-, FIP1 siRNA-, FIP2 siRNA- or FIP5 siRNA-treated human macrophages to verify FIP2 and FIP1 silencing. (E) FIP5 mRNA levels measured by RT-qPCR in NS RNA- and FIP5 siRNA-treated human macrophages, between technical replicates from $n = 4$ human donors. $P = 0.0001$ between NS RNA- and FIP2 siRNA-treated macrophages (Welch $t$ test). (F) Immunoblot of pro-IL-1β, IL-1β p17, pro-caspase-1 (including a light exposure of the blot) and caspase-1 p20 in supernatants from human macrophages treated with NS RNA, FIP1 siRNA, FIP2 siRNA or FIP5 siRNA, one donor of at least $n = 3$ human donors. (G) Quantification of IL-1β p17 protein levels in supernatants from NS RNA-, FIP1 siRNA-, FIP2 siRNA- or FIP5 siRNA-treated human macrophages, $n = 4$ human donors. $P = 0.0001$ (NS RNA vs FIP2 siRNA) after LPS and nigericin. (H) Quantification of pro-IL-1β protein levels in supernatant of the macrophages from G, $n = 3$ human donors. $P = 0.0159$ (NS RNA vs FIP1 siRNA) and $P = 0.0218$ (NS RNA vs FIP2 siRNA) after LPS and nigericin. (I) Quantification of caspase-1 p20 protein levels in the macrophage supernatants, $n = 3$ human donors. (J) Quantification of pro-caspase-1 protein levels in the macrophages from (G), $n = 3$ donors. The cells were treated with respective siRNAs before primed with 100 ng/mL LPS for 2 h and treated with 5 µM Nigericin for 2 h as indicated. In data (A, B), data are presented as mean $+/-$ s.d. and in (C, E, G–J) as mean $+/-$ s.e.m. and shown as black bars (two-way ANOVA Tukey's multiple comparisons test with adj. $P$ values). Source data are available online for this figure.

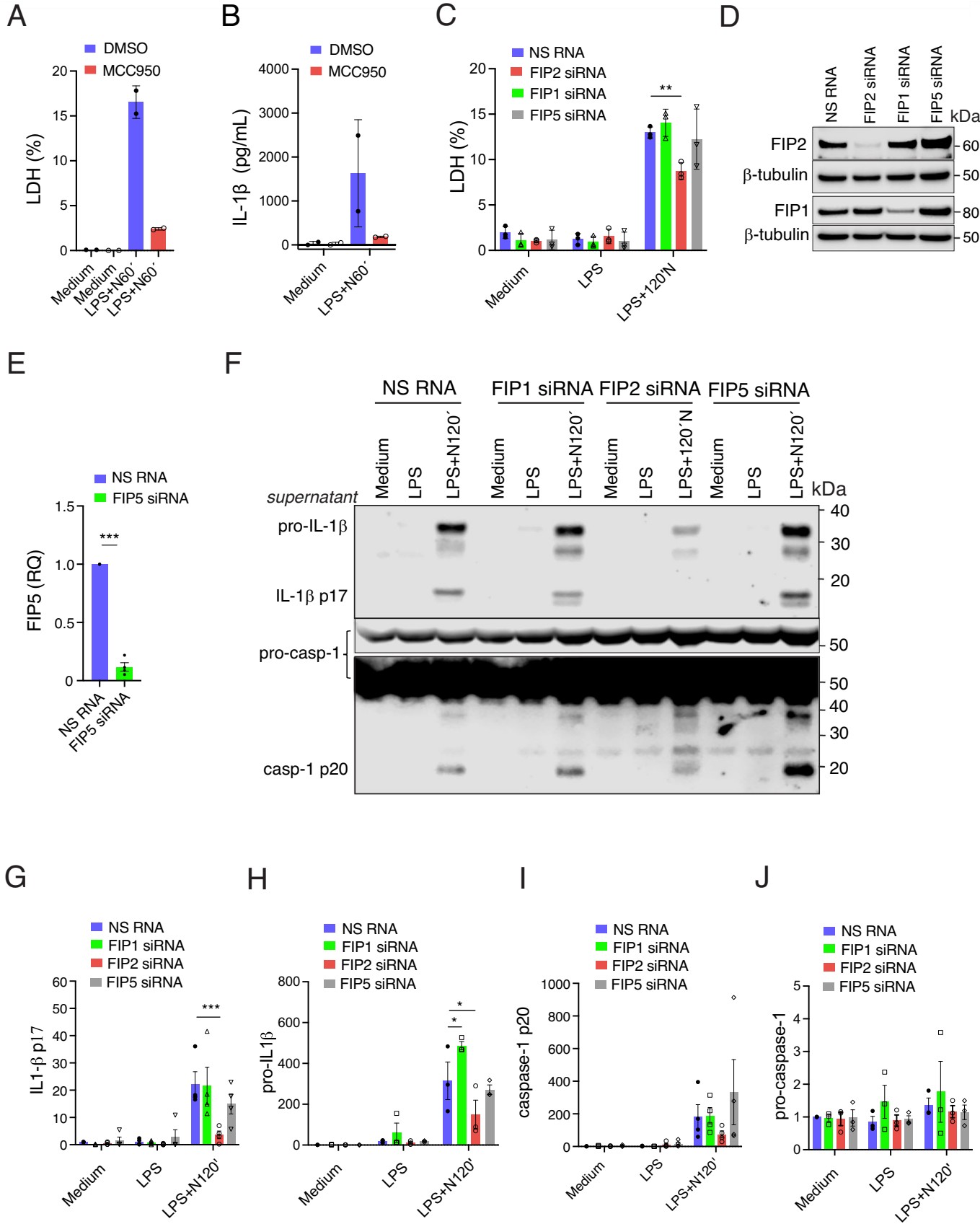

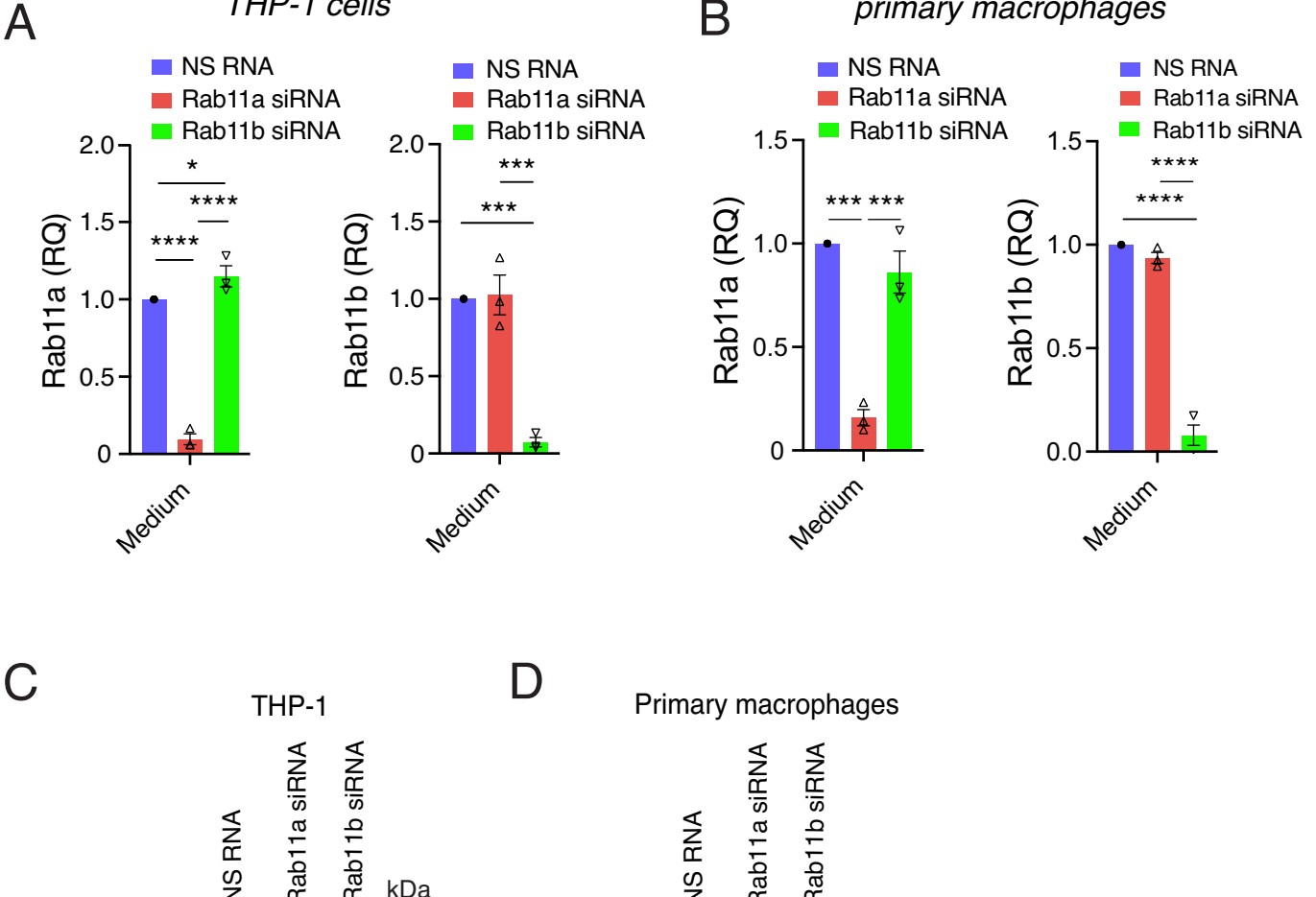

**Figure EV2. Rab11a and Rab11b silencing efficiency.**

(A) Rab11a and Rab11b levels measured by RT-qPCR in THP-1-derived macrophages, $n = 3$ independent experiments. Left panel: $P < 0.0001$ (NS RNA vs Rab11a siRNA), $P = 0.0285$ (NS RNA vs Rab11b siRNA) and $P < 0.0001$ (Rab11a siRNA vs Rab11b siRNA) unstimulated cells. Rigth panel: $P = 0.0004$ (NS RNA vs Rab11b siRNA) and $P = 0.0001$ (Rab11a siRNA vs Rab11b siRNA) unstimulated cells. (B) Rab11a and Rab11b levels measured by RT-qPCR in primary human macrophages, $n = 3$ human donors. Left panel: $P = 0.0002$ (NS RNA vs Rab11b siRNA) and $P = 0.0004$ (Rab11a siRNA vs Rab11b siRNA) unstimulated cells. Right panel: $P < 0.0001$ (NS RNA vs Rab11a siRNA) and $P < 0.0001$ (Rab11a siRNA vs Rab11b siRNA) unstimulated cells. (C) Rab11a levels measured by Rab11a immunoblotting in THP-1-derived macrophages. (D) Rab11a levels measured by Rab11a immunoblotting in primary human macrophages. In data (A, B), data are presented as mean $+/-$ s.e.m. and shown as black bars (one-way ANOVA Holm–Sidak´s multiple comparisons test with adj. $P$ values). Source data are available online for this figure.

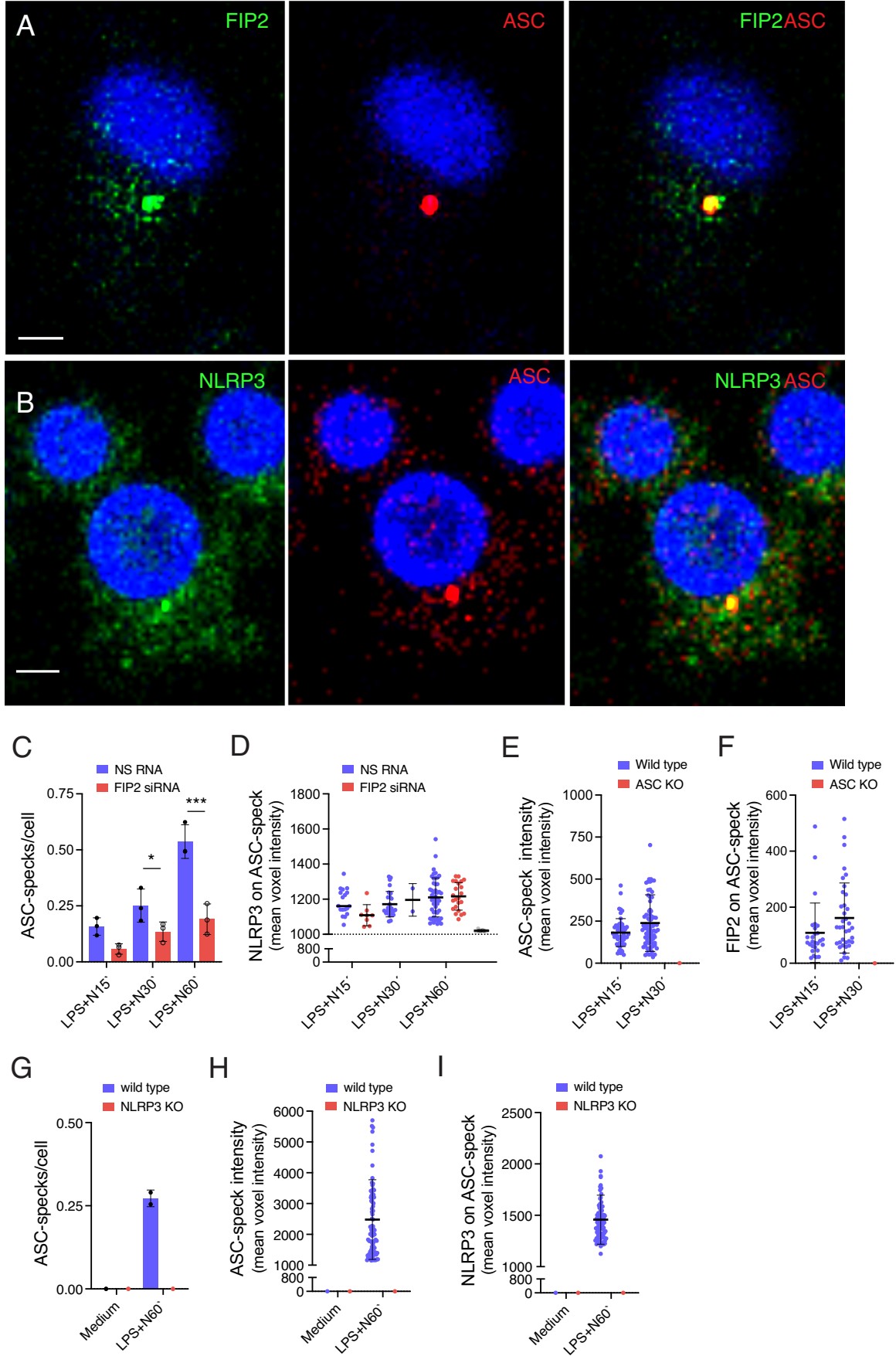

**Figure EV3.   Characterization of ASC speck formation and NLRP3 recruitment.**

(A) Confocal image showing FIP2 (green) on ASC speck (red). (B) Confocal image showing NLRP3 (green) on ASC speck (red). The THP-1-derived macrophages were treated with NS RNA or FIP2 siRNA before they were primed with LPS for 2 h and then treated with nigericin for 30 min (A, B). (C) Quantification of ASC speck formation in LPS-primed cells following 15 min, 30, min and 60 min of nigericin treatment of THP-1-derived macrophages, in $n = 3$ three biological replicates of the same experiment containing 195–239 cells in total. $P = 0.0137$ (NS RNA vs FIP2 siRNA) 30 min nigericin of treatment in LPS-primed cells and $P < 0.0001$ (NS RNA vs FIP2 siRNA) 60 min of nigericin treatment of LPS-primed cells. (D) Quantification NLRP3 intensity on ASC specks in LPS primed cells treated with nigericin as indicated, $n = 1$ biological replicate. (E) Quantification of ASC speck intensity in wild type and ASC-deficient THP-1-derived macrophages. 266–483 cells were monitored per condition. (F) Quantification of FIP2 intensity on ASC specks in the cells of (E). (G) Quantification of NLRP3 intensity on ASC specks in wild type and NLRP3 KO THP-1-derived macrophages stimulated as indicated, $n = 2$ independent experiments monitoring in total 243–413 cells. (H) Quantification of ASC speck intensity in the cells of (G). (I) Quantification of NLRP3 on ASC specks in the cells of (G). ASC specks were identified by the spot detection mode of the IMARIS 8.2 imaging software on 3-D confocal imaging raw data. The cells were treated with respective siRNAs before primed with 100 ng/mL LPS for 2 h and treated with 5 µM Nigericin as indicated. Data in (D, H, I) are presented as 16 bits and therefore have higher values than the data in (E, F) which is presented as 12 bits. In (C), data are presented as mean $+/-$ s.e.m. and shown as black bars (one-way ANOVA Holm–Sidak´s multiple comparisons test with adj. $P$ values). N nigericin. Source data are available online for this figure.

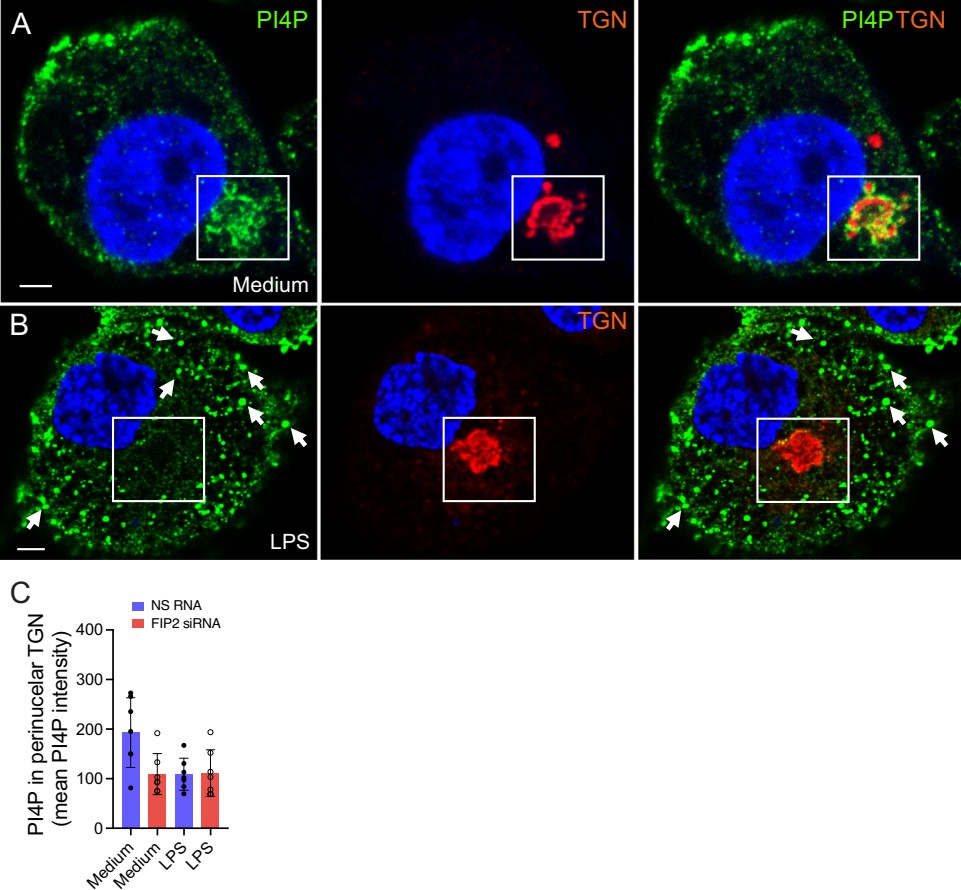

**Figure EV4. There is a marked reduction of PI4P in the TGN-ring ring following LPS priming.**

(A) Confocal images showing PI4P-positive (green) in the perinuclear TGN-ring (red) in unstimulated THP-1-derived macrophages. (B) Confocal images showing PI4P (green) TGN46-positive puncta (red) in LPS-primed THP-1-derived macrophages. (C) Quantification of PI4P in the perinuclear TGN-ring during. The cells were left unstimulated or primed with 100 ng/mL LPS for 2 h. PI4P was quantified in TGN46-positive ring structures were using the IMARIS 8.2 imaging software on 3-D confocal imaging raw data. In (C), data are presented as mean +/− s.d. and shown as black bars. Arrows points at TGN46 puncta positive for PI4P. Scale bar = 5 µm. Source data are available online for this figure.

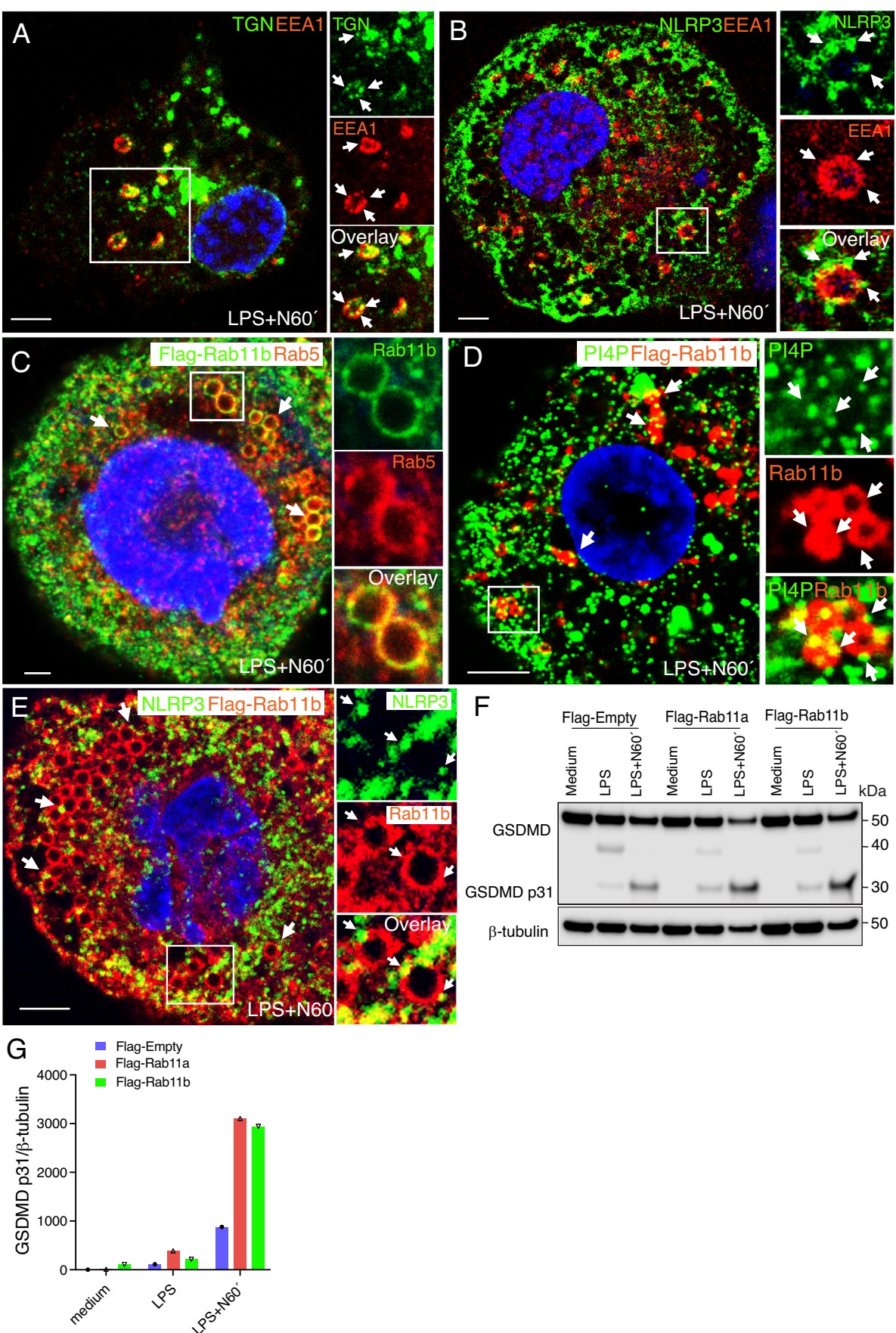

◀ **Figure EV5. PI4P and NLRP3 are locate to microdomains on Rab11b-positive early endosomes.**

(A) Confocal image showing TGN46 (green) on EEA1-positive endosomes (red). (B) Confocal image showing NLRP3 (green) on a EEA1-positive endosome (red). (C) Confocal image showing Rab5 (red) on Rab11b-positive endosomes (green) in Flag-Rab11b expressing THP-1-derived macrophages. (D) Confocal image showing PI4P (green) positive microdomains on Flag-Rab11b-positive endosomes. (E) Confocal image showing NLRP3 (green) positive microdomains on Flag-Rab11b endosomes (red). (F) Immunoblot total GSDMD, caspase-1 cleaved p31 GSDMD fragment and b-tubulin in Flag-Empty, Flag-Rab11a and Flag-Rab11b co-expressing THP-1-derived macrophages. (G) Quantification of the GSDMD p31 fragment in the cells from (F). The cells were primed with 100 ng/mL LPS for 2 h and treated with 5 μM nigericin for 1 h. Scale bar = 5 μm. Source data are available online for this figure.

