## [Peer Review File · The EMBO Journal]

GTPase Rab11b and effector Rab11-FIP2 promote NLRP3 stability during inflammasome priming

Caroline Gravastrand, Maria Yurchenko, Stine Kristensen, Astrid Skjesol, Carmen Chen, Sindre Ullmann, Zunaira Iqbal, Karoline Dahlen, Kashif Rasheed, Unni Nonstad, Liv Ryan, Terje Espevik, and Harald Husebye

Corresponding author(s): Harald Husebye (harald.husebye@ntnu.no)

Review Timeline:

Submission Date:	14th Apr 25
Editorial Decision:	2nd Jun 25
Revision Received:	14th Oct 25
Editorial Decision:	24th Nov 25
Revision Received:	2nd Feb 26
Accepted:	25th Feb 26

Editor: Ioannis Papaioannou

Transaction Report:

Dear Dr. Husebye,

Thank you for submitting your manuscript EMBOJ-2025-121050 for consideration by The EMBO Journal, and for your patience during peer review. Your manuscript has now been seen by two experts in the field, and we have received their detailed and informative reports, which you can find below.

I am pleased to say that, as you will see, both referees indicate interest in your study and the reported findings, and suggest that this work will likely be of broad interest to the innate immunity community. However, both of them identify a number of limitations and mention that additional controls and validation experiments are required for several experiments, further clarification is necessary for some data, and better contextualization of the findings and clarity in the proposed mechanistic model are also needed. We largely agree with the referees and think that these are all important issues that must be addressed for the conclusions of the work to be fully supported and for a broader impact of the work on the field.

Given the referees' supportive comments and positive recommendations, I would like to invite you to submit a revised version of your manuscript taking the referees' suggestions on board, along with a detailed point-by-point response addressing all referees' comments. I should add that it is The EMBO Journal policy to allow only a single round of major revision, and acceptance of your manuscript will therefore depend on the completeness of your responses in this revised version. Please let me know if you have any questions or comments that you would like to discuss with me. If there are any major points you do not agree with or cannot address during your revision, I would encourage you to share them with me as early as possible to discuss how to proceed further in the most efficient way.

We generally allow three months as standard revision time (September 1, 2025). As a matter of policy, competing manuscripts published during this period will not negatively impact our assessment of the conceptual advance presented by your study. However, we request that you contact us as soon as possible upon publication of any related work, to discuss how to proceed. Should you foresee a problem in meeting this three-month deadline, please let us know in advance and we will be able to grant an extension.

Thank you for the opportunity to consider your work for publication in The EMBO Journal. I look forward to your revision.

Best regards,

Ioannis

Instructions for preparing your revised manuscript

1. When you are ready to submit the revision, please upload:

- A Word file of the manuscript text (including legends of main Figures, EV Figures and Tables). Please make sure that changes are highlighted (or "tracked") to be clearly visible.

- Individual production-quality figure files (one file per figure). When assembling your figures, please refer to our figure preparation guidelines in order to ensure proper formatting and readability in print as well as on screen:

If the data shown in a figure are obtained from n {less than or equal to} 2, please use scatter plots showing the individual data points.

- i. the name of the statistical test used to generate error bars and P values
- ii. the number (n) of independent experiments (please specify technical or biological replicates) underlying each data point (discussion of statistical methodology can be reported in the Materials and Methods section, but figure legends should contain a basic description of n , P, and the test applied)
- iii. the nature of the bars and error bars (s.d., s.e.m.).

- A point-by-point response to the referees' comments, with a detailed description of the changes made (as a word file). All referees' concerns must be fully addressed and their suggestions taken on board. When preparing your letter of response to the referees' comments, please bear in mind that this will form part of the Review Process File and will therefore be available online to the community. Please note that you have the possibility to opt out of the transparent process at any stage prior to publication by letting the editorial office know (contact@embojournal.org); if you do opt out, the Review Process File link will point to the following statement: "No Review Process File is available with this article, as the authors have chosen not to make the review process public in this case.". For more details on our Transparent Editorial Process, please visit our website: <https://www.embopress.org/page/journal/14602075/authorguide#transparentprocess>

- Expanded View (EV) files (replacing Supplementary Information) that are collapsible/expandable online. A maximum of 5 EV Figures can be typeset. EV Figures should be cited as "Figure EV1, Figure EV2" etc. in the text, and their respective legends should be included in the manuscript file after the legends of regular figures. See detailed instructions regarding Expanded View files here: <https://www.embopress.org/page/journal/14602075/authorguide#expandedview>

- For the figures that you do NOT wish to display as Expanded View figures, they should be bundled together with their legends in a single PDF file called "Appendix", which should start with a short Table of Contents (including page numbers). Appendix figures should be referred to in the main text as: "Appendix Figure S1, Appendix Figure S2" etc. Please see detailed instructions here: <https://www.embopress.org/page/journal/14602075/authorguide#expandedview>

- A complete author checklist, which you can download from our author guidelines (<https://www.embopress.org/page/journal/14602075/authorguide>). Please note that the checklist will also be part of the Review Process File.

2. Please note that no statistics should be calculated and shown in Figures if $n=2$. Please also note that each p value should be reported as an exact value.

3. Before submitting your revision, primary datasets (and computer code, where appropriate) produced in this study need to be deposited in appropriate public databases (see <https://www.embopress.org/page/journal/14602075/authorguide#dataavailability>). The accession numbers, database, and the specific URLs (links) should be listed in a formal "Data availability" section (placed after Methods), following the example below:

"The RNA-seq datasets produced in this study are available in the following database:
Gene Expression Omnibus GSE46843 (<https://www.ncbi.nlm.nih.gov/geo/query/acc.cgi?acc=GSE46843>)"

*** All links should resolve to a page where the data can be accessed. ***

*** Please remember to provide in the Data availability section of your revised manuscript reviewer passwords if the datasets are not yet public. ***

*** The Data Availability Section is restricted to new primary data that are part of this study. In case you have no data that require deposition in a public database, please state so instead of referring to the database: "Our study includes no data deposited in public repositories." under the heading "Data availability". ***

4. The materials and methods need to be described in the manuscript using our structured methods format, which is now required for all research articles. According to this format, the Methods section includes a single "Reagents and Tools Table" - listing key reagents, experimental models, software and relevant equipment including their sources and relevant identifiers- followed by a "Methods and Protocols" section describing the methods. Please download and fill our Reagents and Tools Table template (.docx), which you can find in our author guide:

<https://www.embopress.org/page/journal/14602075/authorguide#structuredmethods>. When submitting your revised manuscript, please do not include the Reagents and Tools Table in the Methods section of the manuscript but instead upload it as a separate file choosing the file type "Reagent Table".

5. Please check that the title and the abstract of the manuscript are brief, yet explicit, even to non-specialists. The length of the title should not exceed 100 characters, and the abstract should be a single paragraph not exceeding 175 words.

6. Please also note our reference format: <https://www.embopress.org/page/journal/14602075/authorguide#referencesformat>.

8. Please remember: digital image enhancement is acceptable practice, as long as it accurately represents the original data and conforms to community standards. If a figure has been subjected to significant electronic manipulation, this must be noted in the

figure legend or in the "Materials and Methods" section. The editors reserve the right to request original versions of figures and the original images that were used to assemble the figure.

9. Our journal encourages inclusion of data citations in the reference list to directly cite datasets that were obtained from public databases. Data citations in the article text are distinct from normal bibliographical citations and should directly link to the database records from which the data can be accessed. In the main text, data citations are formatted as follows: "Data ref: Smith et al, 2001" or "Data ref: NCBI Sequence Read Archive PRJNA342805, 2017". In the Reference list, data citations must be labeled with "[DATASET]". A data reference must provide the database name, accession number/identifiers, and a resolvable link to the landing page from which the data can be accessed at the end of the reference. Further instructions are available at: <https://www.embopress.org/page/journal/14602075/authorguide#referencesformat>.

10. We request authors to consider both actual and perceived competing interests. Please review our policy (<https://www.embopress.org/page/journal/14602075/authorguide#conflictsofinterest>) and update your competing interests statement if necessary. Please name this section 'Disclosure and competing interests statement' and place it after the Acknowledgements section.

11. Please note that all corresponding authors are required to provide an ORCID ID upon submission of a revised manuscript (<https://orcid.org/>). Please find instructions on how to link your ORCID ID to your account in our manuscript tracking system in our Author guidelines (<https://www.embopress.org/page/journal/14602075/authorguide#authorshipguidelines>).

12. We use CRediT to specify the contributions of each author in the journal submission system. CRediT replaces the author contribution section, which should be removed from the manuscript. Please use the free text box to provide more detailed descriptions. See also guide to authors: <https://www.embopress.org/page/journal/14602075/authorguide#authorshipguidelines>.

14. We would also welcome the submission of cover suggestions or motifs to be used by our Graphics Illustrator in designing a cover.

Referee #1:

The NLRP3 inflammasome is a multiprotein complex that plays a central role in innate immunity, with its activation representing a key step in the inflammatory response. A critical upstream event in NLRP3 activation is its recruitment to vesicular compartments that arise either from disruption of the trans-Golgi network (TGN) or from impaired retrograde trafficking. At the molecular level, this recruitment is proposed to involve a positively charged motif within NLRP3 that mediates its binding to phosphatidylinositol-4-phosphate (PI4P), a lipid enriched on these disrupted membrane compartments.

In this study, Husebye and colleagues report that RAB11-FIP2 (FIP2), a Rab11 effector protein, functions as a regulator of NLRP3 activation. FIP2 is known to coordinate vesicular trafficking, and has previously been linked to the regulation of TLR signaling. The authors demonstrate that knockdown of FIP2 leads to reduced LDH and IL-1 β release following nigericin-induced NLRP3 activation, along with diminished processing of caspase-1 and gasdermin D in both THP-1 cells and monocyte-derived macrophages (MDMs). Notably, knockdown of Rab11b produced a similar phenotype, suggesting that Rab11b and FIP2 cooperate to regulate NLRP3 inflammasome activity.

Imaging studies revealed that FIP2 and NLRP3 co-localize with the ASC speck during activation, and co-immunoprecipitation assays support a direct interaction between NLRP3 and FIP2. Further investigation into the activation mechanism showed that LPS priming promotes the recruitment of NLRP3 to the Golgi/TGN in a FIP2-dependent manner, and that this recruitment persists during inflammasome activation. Co-localization analyses also indicated that both NLRP3 and FIP2 associate with Rab5-positive endosomes, suggesting a role for early endosomes in coordinating inflammasome assembly.

This is an interesting and timely study that identifies a novel role for the FIP2-Rab11b axis in regulating NLRP3 inflammasome activation. Given recent findings highlighting the importance of TGN disruption and retrograde trafficking impairment in NLRP3 activation, this work is likely to be of broad interest to the innate immunity and cell biology communities. However, as NLRP3

activation remains a highly active and debated area of research, it is essential that the authors carefully contextualize their findings within the broader literature, particularly with regard to existing models of inflammasome regulation. Furthermore, to convincingly establish the proposed mechanism and gain acceptance within the field, the study must meet a high standard of rigor. Therefore, I recommend the inclusion of additional controls and validation experiments to strengthen the conclusions.

Please find my detailed comments and suggestions below.

Major points:

1) One major point that remains unclear is whether FIP2 regulates NLRP3 activation at the level of translational priming, post-translational priming, or activation itself. Given FIP2's known roles in TLR signaling, and the observation that FIP2 deficiency reduces pro-IL-1 β induction, it is plausible that the primary effect on NLRP3 activation results from a defect in priming-i.e., Signal 1 of the NLRP3 pathway. However, it is also possible that FIP2 functions at both Signal 1 and Signal 2 levels. This distinction requires further clarification and experimental dissection.

2) While the authors present a substantial amount of data, it remains difficult to reconcile these findings with existing literature. Specifically, it is unclear what model the authors propose regarding NLRP3 activation. Including a schematic summarizing their proposed mechanism would significantly enhance clarity and help situate their findings within the broader field.

3) It is commendable that the study investigates NLRP3 activation using antibodies and knockdowns of the endogenous protein, rather than relying on overexpression of fluorescently tagged constructs. However, the validity of these tools must be rigorously verified. All knockdowns should be confirmed by immunoblotting for the targeted proteins, and the specificity of all antibodies must be validated in knockout or knockdown cell lines. This is especially critical given the known cross-reactivity of many NLRP3 antibodies. Without such validation, there is a risk that some of the observed staining in the figures reflects off-target signals rather than genuine NLRP3.

4) Another point that requires clarification is whether the authors are investigating a disrupted trans-Golgi network (TGN) compartment or stalled endosomes resulting from impaired retrograde trafficking. While TGN46 is used as a marker, it is important to consider whether TGN46 truly reflects an intact TGN structure, or if it is itself affected by retrograde trafficking defects. This distinction is critical for the interpretation of the data. The authors should explain why they chose TGN46 as a marker and clarify which model-the disruption of the TGN versus the accumulation of stalled endosomal compartments-is better supported by their data. This would help the reader understand the mechanistic implications of their findings within the context of retrograde trafficking pathways.

Additional points:

Figure 1:

- Fig. 1A: Immunoblots confirming FIP2 knockdown are missing and should be included to validate the efficiency of the knockdown.
- Fig. 1C: There is a noticeable reduction in pro-IL-1 β levels. How much of the reduction seen in the ELISA can be attributed to this decrease in pro-IL-1 β ? It may be more informative to assess IL-18 secretion by ELISA, as IL-18 does not require priming and could better reflect inflammasome activation independently of Signal 1.
- Fig. 1F: Include blots for pro-IL-1 β and pro-caspase-1 to provide a complete picture of inflammasome component levels.
- Fig. 1G-I: These experiments raise concerns. FIP2 overexpression appears to increase pro-caspase-1 levels, suggesting potential transcriptional regulation. Do the authors propose that FIP2 regulates inflammasome components at the transcriptional level? This possibility should be explored.
- General Comment (Fig. 1): It is essential to demonstrate that NLRP3 and ASC protein levels remain unchanged upon FIP2 knockdown to ensure observed effects are not due to altered expression of these core inflammasome components.

Figure EV1C:

- Confirm FIP2 knockdown by immunoblotting to support the specificity and efficiency of the silencing.

Figure 2:

- Rab11a/b KD: Confirm knockdowns of Rab11a and Rab11b by immunoblotting.
- Fig. 2C: Full-length blots should be shown for pro-IL-1 β , caspase-1, ASC, NLRP3, and GSDMD to allow proper interpretation of protein processing.
- Figs. 2B-C: To address the discrepancy between ELISA data and immunoblots, it would be helpful to assess pro-IL-1 β levels in the supernatant via immunoblotting.
- Fig. 2D: The text states that caspase-1 p20 levels are increased, but the figure appears to show a reduction. Please clarify this inconsistency.

Figure 3:

- Validate the specificity of the NLRP3 and FIP2 immunostaining by using knockout or knockdown cells.
- Include additional quantification, such as the number of NLRP3-positive or FIP2-positive ASC specks, to strengthen the interpretation of speck formation and co-localization.

Figure 4:

- Include a schematic of NLRP3 domains and the constructs tested to help the reader understand the structure-function relationship being investigated.
- Fig. 4D: Whole-cell lysates (WCLs) are missing and should be included to verify equal protein expression and loading.
- A discrepancy exists: Panel A shows that Rab11 is not required for the NLRP3-FIP2 interaction, whereas Panel E suggests it is. This contradiction should be addressed and explained.
- Clarify whether the Rab11 used in pull-down experiments refers to Rab11a, Rab11b, or both.

Figure 5:

- Indicate the percentage of Golgi structures that are NLRP3-positive to better quantify the observed localization.
- Consider whether the NLRP3 signal seen in unprimed cells is background or non-specific. Could the reduced expression of NLRP3 in FIP2-deficient cells explain the observed staining patterns?

Figure 6:

- Is a direct interaction between FIP2 and NLRP3 required for NLRP3 translocation to the TGN, or is this an indirect effect? Additional experiments could help distinguish between these possibilities.

Figure 7:

- Why does FIP2 deficiency or MCC950 treatment reduce PI4P levels on vesicles? This result implies that NLRP3 recruits PI4P, which contradicts some previously published findings. The authors should reconcile this discrepancy.

Figure 8:

- The current data do not establish a clear link to NLRP3 activation. It would strengthen the manuscript to demonstrate that IKK signaling is required for NLRP3 activation in this context, possibly via inhibition or genetic approaches

Referee #2:

In this MS the authors present data to support that Rab11-FIP2 and Rab11b, but not Rab11a, control caspase-1 mediated cleavage of pro-IL-1b and GSDMD, pyroptotic cell death and TAK1/IKK mediated NLRP3 translocation to the trans-Golgi network in human macrophages. They present further data to show that NLRP3 binds to Rab11-FIP2 via its KMKK motif, Rab11-FIP2 interacts with NLRP3 via its N-terminal C2-domain and that PI4P positive endosomes and ASC-speck formation were also controlled by Rab11-FIP2. This study builds on earlier work by the authors and further develops work from others presenting a mechanism to explain how endosomal regulation of PI4P is important for endosome-to-trans-Golgi network trafficking for recruitment of NLRP3 and subsequent inflammasome activation. The data broadly supports the authors conclusions although additional control data is required and further explanation for some of the observations are necessary. Concerns to be addressed:

Fig 1A-F FIP2 knockdown or silencing only reduces inflammasome activity. Knock down probably does not remove all the FIP2 although data is not presented which shows how successful the knock down is and this information needs to be presented in all cell types for all relevant experiments especially as the authors claim this technique is FIP2 silencing (apologies if this reviewer has missed this: so far a FIP2 blot is present in Fig 8, but not elsewhere and it needs to accompany every figure where knock down/silencing/depletion is present). Given the knock down only partially reduces inflammasome activation can the authors explain why this is (presumably due to residual FIP2 expression after knock down, but this needs clarifying)? The words FIP2 knock down, silencing and deletion are used interchangeably and this needs to be consistent supported by FIP2 blots to confirm the claims.

Fig 1G-H: why is the empty vector control here also affecting inflammasome activity?

Fig 1I: why would an elevation in pro-IL-1b expression be expected in the presence of the lentiviral Flag-FIP2 (pLVX-FIP2) or Flag-Empty (pLVX-Empty) vectors as suggested by the authors "in line with these results" when the previously presented results concern changes in inflammasome activity rather than inflammasome priming? Control blots showing increased or no change of FIP2 expression in test vs control transduced cells are required here.

Fig EV1 control blots showing the efficiency of silencing approaches need to be shown here

Fig 2 again control blot along side the qPCRs showing the efficiency of knock down of the Rab proteins is required here. Rab11a seems to enhance casp 1 processing as well as proIL1 processing in the blots in 2C and D?

Fig 3: at this resolution it is hard to say if FIP2 is at the core of the ASC speck. The proteins are certainly co-located, but the text needs to be modified to take this into account.

Referee #1:

The NLRP3 inflammasome is a multiprotein complex that plays a central role in innate immunity, with its activation representing a key step in the inflammatory response. A critical upstream event in NLRP3 activation is its recruitment to vesicular compartments that arise either from disruption of the trans-Golgi network (TGN) or from impaired retrograde trafficking. At the molecular level, this recruitment is proposed to involve a positively charged motif within NLRP3 that mediates its binding to phosphatidylinositol-4-phosphate (PI4P), a lipid enriched on these disrupted membrane compartments.

In this study, Husebye and colleagues report that RAB11-FIP2 (FIP2), a Rab11 effector protein, functions as a regulator of NLRP3 activation. FIP2 is known to coordinate vesicular trafficking and has previously been linked to the regulation of TLR signaling. The authors demonstrate that knockdown of FIP2 leads to reduced LDH and IL-1 β release following nigericin-induced NLRP3 activation, along with diminished processing of caspase-1 and gasdermin D in both THP-1 cells and monocyte-derived macrophages (MDMs). Notably, the knockdown of Rab11b produced a similar phenotype, suggesting that Rab11b and FIP2 cooperate to regulate NLRP3 inflammasome activity.

Imaging studies revealed that FIP2 and NLRP3 co-localize with the ASC speck during activation, and co-immunoprecipitation assays support a direct interaction between NLRP3 and FIP2. Further investigation into the activation mechanism showed that LPS priming promotes the recruitment of NLRP3 to the Golgi/TGN in a FIP2-dependent manner, and that this recruitment persists during inflammasome activation. Co-localization analyses also indicated that both NLRP3 and FIP2 associate with Rab5-positive endosomes, suggesting a role for early endosomes in coordinating inflammasome assembly.

This is an interesting and timely study that identifies a novel role for the FIP2-Rab11b axis in regulating NLRP3 inflammasome activation. Given recent findings highlighting the importance of TGN disruption and retrograde trafficking impairment in NLRP3 activation, this work is likely to be of broad interest to the innate immunity and cell biology communities. However, as NLRP3 activation remains a highly active and debated area of research, it is essential that the authors carefully contextualize their findings within the broader literature, particularly with regard to existing models of inflammasome regulation.

Furthermore, to convincingly establish the proposed mechanism and gain acceptance within the field, the study must meet a high standard of rigor. Therefore, I recommend the inclusion of additional controls and validation experiments to strengthen the conclusions.

Please find my detailed comments and suggestions below.

Address

NTNU, Faculty of
medicine, IKOM
Postbox 8905
NO-7491 Trondheim

Org.nr. 974 767 880

Email: kontakt@ikom.ntnu.no
<http://www.ntnu.edu/ikom>

Location

Laboratoriesenteret
5th floor
Erling Skjalgsons gate, 1

Phone

+47 97706661

Major points:

1) One major point that remains unclear is whether FIP2 regulates NLRP3 activation at the level of translational priming, post-translational priming, or activation itself. Given FIP2's known roles in TLR signaling, and the observation that FIP2 deficiency reduces pro-IL-1 β induction, it is plausible that the primary effect on NLRP3 activation results from a defect in priming-i.e., Signal 1 of the NLRP3 pathway. However, it is also possible that FIP2 functions at both Signal 1 and Signal 2 levels. This distinction requires further clarification and experimental dissection.

- *We have now carefully investigated FIP2's role in signal 1 (priming) and signal 2 (inflammasome activation). In the revised manuscript we have included the data **in a new Figure 2**. We have included data from THP-1 cells with FIP2 silencing or Flag-FIP2 co-expression as well as primary macrophages with FIP2 silencing (Fig EV2). These robust data show that FIP2 does not significantly affect neither NLRP3 nor IL-1 β mRNA induction during LPS priming (signal 1). Instead, FIP2 has a stabilizing effect on the NLRP3 and IL-1 β proteins during priming (Fig 2B, D, E, F). Thus, FIP2 has a major effect on NLRP3 and IL-1 β protein stabilization during LPS priming. The way FIP2 does this is to stabilize TAK1 that phosphorylates and activates IKK β it during LPS "priming". IKK β has been shown by several research groups to regulate NLRP3 translocation to the TGN. In the TGN, NLRP3 shuttles between the perinuclear TGN-ring and the early endosome in a FIP2-dependent manner. Upon inflammasome activation by nigericin, FIP2 allows inflammasome activation by regulating PI4P levels at the endosome and NLRP3 binding to PI4P.*
- *In the new Figure 3 (previously Figure 2), we have also included new data on the role of Rab11b in Signal 1 and Signal 2. This data shows that also Rab11b (and not Rab11a), controls NLRP3 and IL-1 β proteins levels during LPS priming while their respective mRNA levels are relatively unchanged.*

2) While the authors present a substantial amount of data, it remains difficult to reconcile these findings with existing literature. **Specifically, it is unclear what model the authors propose regarding NLRP3 activation.** Including a schematic summarizing their proposed mechanism would significantly enhance clarity and help situate their findings within the broader field.

- *Our model suggests that FIP2, with the help of Rab11b, is involved in regulating NLRP3 anterograde trafficking from the perinuclear TGN-ring to early endosomes and retrograde trafficking from the early endosomes and back to the perinuclear TGN during LPS priming. In other words, FIP2 together with Rab11b, controls recycling from the endosome to the TGN, similar to what has been observed for NLRP3, the trans-Golgi protein TGN46 and the cation-independent mannose-6-phosphate receptor (CI-MPR) (Zhang et al, 2023).*
- *Our data supports that NLRP3 and PI4P accumulate on early endosomes, positive for TGN46, as a result from a stop in retrograde trafficking caused by nigericin. We found that these NLRP3 positive endosomes increased in size due to a stop in retrograde trafficking, probably because anterograde trafficking is still operational after nigericin treatment. Endosome enlargement because of a stop of retrograde trafficking was first described in (Huotari & Helenius, 2011). Also, our results show that FIP2 silenced cells have reduced both anterograde and retrograde trafficking that leads to less TGN46 accumulation after nigericin. One effect of deficient retrograde trafficking is that TGN46 could be missorted to*

the lysosome during priming, as can be the case for NLRP3. Such observations have been done for the low-density lipoprotein receptor LDL receptor, mannose-6-phosphate receptor and for TGN38, in Rab11 deficient cells, (Hu et al, 2022; Zulkefli et al, 2019).

- ***We have also revised the Graphical abstract that now better clarifies the model how FIP2 controls NLRP3 activation in six major steps.***

3) It is commendable that the study investigates NLRP3 activation using antibodies and knockdowns of the endogenous protein, rather than relying on overexpression of fluorescently tagged constructs. However, the validity of these tools must be rigorously verified. ***All knockdowns should be confirmed by immunoblotting for the targeted proteins, and the specificity of all antibodies must be validated in knockout or knockdown cell lines.*** This is especially critical given the known cross-reactivity of many NLRP3 antibodies. ***Without such validation, there is a risk that some of the observed staining in the figures reflects off-target signals rather than genuine NLRP3.***

- *Just to clarify, no overexpression of Fluorescently tagged proteins has been used in this study! We have only used Flag-tagged pLVX vectors of Flag-FIP2 and the pLVX-empty vector as control for Flag-M2-antibody-staining for confocal imaging to visualize Flag-FIP2 on endosomes.*
- *To verify the specificity of the FIP2 antibody used for immunoblotting in THP-1 cells expressing a non-targeting RNA cell line deficient in FIP2 expression was made. Next, we made lysates and immunoblotted two of the FIP2 KO cell lines together with control cells encoding a non-targeting guide. No FIP2 signal could be detected in none of the knockout (KO) cell lines, FIP2KO1 and FIP2KO2 using the FIP2 antibody used for immunoblotting. This information has been included in Appendix Figure S1A.*
- *To verify the specificity of the NLRP3 antibody we used control- and NLRP3 KO THP-1 cells that were either unstimulated or stimulated with LPS for 2h, before lysis and immunoblotting with the NLRP3 antibody. In THP-1 control cells, but not in NLRP3 KO THP-1 cells, a unique NLRP3 band was markedly enhanced upon LPS stimulation, verifying the specificity on Western blotting of the NLRP3 antibody used. This information has been included in Appendix Figure S1B.*
- *To show the specificity of the NLRP3 antibody for immunostaining for confocal imaging, we have included images of control- and NLRP3 KO THP-1 cells before and after LPS priming and nigericin treatment. These images show enlarged NLRP3 positive endosomes in inflammasome activated control cells that were not present in the NLRP3 KO THP-1 cells (appendix Figure S2 A-D).*

4) Another point that requires clarification is whether the authors are investigating a disrupted trans-Golgi network (TGN) compartment or stalled endosomes resulting from impaired retrograde trafficking. ***While TGN46 is used as a marker, it is important to consider whether TGN46 truly reflects an intact TGN structure, or if it is itself affected by retrograde trafficking defects.*** This distinction is critical for the interpretation of the data. ***The authors should explain why they chose TGN46 as a marker and clarify which model-the disruption of the TGN versus the accumulation of stalled endosomal compartments-is better supported by their***

data. This would help the reader understand the mechanistic implications of their findings within the context of retrograde trafficking pathways.

- ***Yes, TGN46 is indeed affected by retrograde defects.***
Previously, TGN46 (TGN38 in mice) has been used a marker for dispersion of TGN or NLRP3 accumulation on dTGN/TGN46 positive early endosomes following nigericin treatment (Chen & Chen, 2018; Zhang et al., 2023). In LPS primed cells we found the TGN46 antibody to stain a perinuclear ring structure that after nigericin disappears and multiple endosomal structures appeared ((the endosomal structures were previously called dispersed TGN (dTGN) (Chen & Chen 2018)). To visualize our supportive data more clearly, we have combined the data previously shown in Figure 5 and Figure 6, in a new Figure 6. Our results strongly support that TGN46 and NLRP3 are cycling between the peri-nuclear ring and the early endosome during LPS priming and that these accumulate on endosomes when a stop in retrograde trafficking is caused by nigericin. That nigericin inhibits retrograde trafficking has been described previously shown by (Zhang et al., 2023). Also, we found a significant reduction of TGN46 positive endosomes following nigericin treatment of FIP2 silenced cells, supporting that FIP2 plays a role in that NLRP3 and TGN46 cycling between TGN and the early endosome. We have included a section in the discussion describing this.

Additional points:

Figure 1:

Fig. 1A: Immunoblots confirming *FIP2 knockdown are missing and should be included to validate the efficiency of the knockdown.*

- *Immunoblot confirming efficient FIP2 knockdown have been included in Figure 1C, 1F and 1I.*

Fig. 1C: There is a noticeable reduction in pro-IL-1 β levels. How much of the reduction seen in the ELISA can be attributed to this decrease in pro-IL-1 β ?

It may be more informative to assess IL-18 secretion by ELISA, as IL-18 does not require priming and could better reflect inflammasome activation independently of Signal 1.

- *How much of the reduction seen in the ELISA that can be attributed to pro-IL-1 β relative to pro-IL-1 β we cannot say. However, we have included a new Figure 2 showing that FIP2 silencing indeed affect pro-IL-1 β levels, both in THP-1 cells and when pro-IL-1 β is averaged between 3 human donors (new Figure 2B right panel and EV1H). (The new Figure 2B and Figure EV1H has one human donor in common, the one showed in the new Figure 1I).*

Figure IL-18 ELISA: (A) IL-18 secretion in supernatants from FIP2 silenced cells primed by LPS for 2h before treated with nigericin for 1h. (B) IL-18 secretion in cells treated with MCC950 or FIP2 siRNA as indicated and stimulated with 2.5 μM or 10 μM nigericin for 4 h.

- Using the Abcam IL-18 ELISA on supernatants from THP-1 cells. In LPS primed and nigericin treated cells and there was no effect of FIP2 silencing on THP-1 cells stimulated in the same way as those in Figure 1C. In cells stimulated with 2.5 μM or 10 μM nigericin for 4h we saw no effect of FIP2 silencing on IL-18 release while the NLRP34 inhibitor MCC950 reduced the IL-18 release by more than 75%.

Fig. 1F: Include blots for pro-IL-1β and pro-caspase-1 to provide a complete picture of inflammasome component levels.

- *First, we have changed the order of data, so the data from the THP-1 Flag-FIP2 expressing cells appears in Figure 1D-to 1F and primary macrophage data from Figure 1G-II. Also, all blots in Figure 1 are from new experiments that include the pro-forms of IL-1β and caspase-1 and degree of FIP2 silencing in these (Fig 1C, F, I)*

Fig. 1G-I: These experiments raise concerns. FIP2 overexpression appears to increase pro-caspase-1 levels, suggesting potential transcriptional regulation. **Do the authors propose that FIP2 regulates inflammasome components at the transcriptional level? This possibility should be explored.**

- *This possibility has now been investigated, and we found that this was a mistake made by us, inserting a blot where the full-length/pro-form was absent. Thus, we have added new data to clarify this point. As stated above, the original blot was missing the pro-caspase-1 p50 band and instead showing a band at 36 kDa with unknown function. We have replaced the blots in Figure II (Fig 1F in the revised figure) with blots from a new experiment. This blot shows that the pro-caspase 1 levels largely are unchanged, while Casp-1 p20 is increased.*

General Comment (Fig. 1): It is essential to demonstrate that NLRP3 and ASC protein levels remain unchanged upon FIP2 knockdown to ensure observed effects are not due to altered expression of these core inflammasome components.

- *We have now included data on NLRP3 and IL-1 β proteins in a New Figure 2. Our new results from THP-1 cells and primary macrophages show that NLRP3 protein is significantly reduced upon LPS priming in the FIP2 silenced cells while in unstimulated cells the NLRP3 levels are not significantly affected by FIP2 silencing (new Fig. 2A, B and Fig EV2). In Flag-FIP2 expressing THP-1 cells also included (Fig 2C, D), NLRP3 protein levels are generally increased. However, FIP2 silencing of THP-1 cells did not significantly change NLRP3 mRNA, suggesting that FIP2 may affect NLRP3 stability.*
- *We have not been able to verify the specificity of the ASC antibody used in Western blots on lysates from THP-1 ASC KO cells, and therefore not including ASC-blot in the revised Figure 1. That the ASC-antibody used for ASC-speck detection is working are shown by an indirect way: using this ASC-antibody we could easily detect ASC-specks in wild type THP-1 cells while neither in the THP-1 ASC KO cells nor in THP-1 NLRP3 KO cells, as expected if the antibody was specific.*
- *These results on ASC-specks were included in the original Figure EV3. Now results from THP-1 KO cells have been included in the revised Figure EV3E and EV3F together with the results from THP-1 NLRP3 KO cells (Fig EV3G-EV3I). However, the ASC antibody used works for immune-staining ASC for imaging, included as new Figure 4C-4E. The images of ASC and FIP2, and ASC and NLRP3 co-localization on ASC-specks (Figure 3A, 3B) have been moved to Figure EV3.*

Figure EV1C: Confirm FIP2 knockdown by immunoblotting to support the specificity and efficiency of the silencing.

- *Blots showing FIP2 silencing have been included 1C and 1I. In Figure 1F a blot with the level of FIP2 expression have been included. To give an overview of the action of FIP2 in THP-1 cells, we have moved the results from the Flag-FIP2 expressing THP-1 cells to 1D-F and the results from the primary human macrophages to Fig 1G-H.*

Figure 2 (New Figure 3): Rab11a/b KD: Confirm knockdowns of Rab11a and Rab11b by immunoblotting.

- *We have included immunoblots of Rab11a in both THP-1 cells and primary human macrophages (Fig. EV3C, D). However, we could not get any of the 4 antibodies towards Rab11b from different suppliers to work and has instead included q-PCR results for three independent experiments of each siRNA treated THP-1 cells and primary human macrophages (Fig EV3C and EV3D).*

Fig. 2C (new Figure 3C): Full-length blots should be shown for pro-IL-1 β , caspase-1, ASC, NLRP3, and GSDMD to allow proper interpretation of protein processing.

- We have included full-length blots for pro-IL-1 β , caspase-1, NLRP3, and GSDMD (new Figure 3). However, as stated above we could not find an ASC-antibody to work on Western. However, this works very well for immunostaining of ASC-specks for confocal.

Figs. 2C-D (new Figure 3C-D): To address the discrepancy between ELISA data and immunoblots, it would be *helpful* to assess pro-IL-1 β levels in the supernatant via immunoblotting.

Figure IL-1 β blots of lysates from THP-1 cells. (Left) IL IL-1 β blot using an IL-1 β antibody also recognizing pro- IL-1 β . (Rigth) IL-1 β blot using an IL-1 β antibody specific for cleaved IL-1 β p17

- We have inserted a western blot showing both IL-1 β p17 and pro- IL-1 β in Figure 3C and 3D (Figure 2C and 2D in original manuscript). However, the figure above shows how the IL1-beta antibodies show different affinity towards pro-IL-1 β and IL-1 β p17. The blot in the left panel been inserted as a new Figure 3C (original figure 2C).

Fig. 2D (new Figure 3D): The text states that caspase-1 p20 levels are increased, but the figure appears to show a reduction. Please clarify this inconsistency.

- This data is true for the THP-1 cell results presented in Figure 2C (3C of the revised manuscript), but not for the data from primary macrophages in 2D. We have no explanation for this.

New figure 4 (original Figure 3):

Validate the specificity of the NLRP3 and FIP2 immunostaining by using knockout or knockdown cells.

- We have included new data on FIP2 and NLRP3 levels on ASC specks in cells showing FIP2 knock-down by siRNA in primary macrophages. For clarity, we are now only showing images from primary macrophages. We have removed the images of FIP2 and NLRP3 on ASC -specks in THP-1 cells and included them in a **new Figure EV4**, together with data

from THP-1 ASC and NLRP3 KO cells. In the ASC- and NLRP3 knock out cells used there were absolutely no ASC-speck formation, neither in the ASC KO nor the NLRP3 KO cells.

Include additional quantification, such as the number of NLRP3-positive or FIP2-positive ASC specks, to strengthen the interpretation of speck formation and co-localization in new Figure 4 (original Figure 3).

- *Data showing the level of FIP2 and NLRP3 on ASC specks in primary human macrophages have been included in Figure 4D and 4E, respectively. In addition, results from THP-1 cells showing the effect of FIP2 silencing on ASC-speck NLRP3 level have been inserted (Figure EV3D). What is interesting about NLRP3 on the ASC-specks, is that the mean values do not change much between control and FIP2 silenced cells (Figs 4D and EV 4D) and (Fig EV3D). There was no significant effect of FIP2 silencing on ASC-speck NLRP3 although the number of ASC-specks was significantly reduced (new Figs. 4C and new EV3C).*

New Figure 5 (original Figure 4):

Include a schematic of NLRP3 domains and the constructs tested to help the reader understand the structure-function relationship being investigated.

- *We have included schematics for NLRP3 in Figure 5D, F and FIP2 in Figure 5H.*

Fig. 4D: Whole-cell lysates (WCLs) are missing and should be included to verify equal protein expression and loading.

- *We have included WCL blots showing EGFP-FIP2 and Flag-NLRP3 in Figure 5E (original Fig. 4D).*

A discrepancy exists: **Panel A shows that Rab11 is not required for the NLRP3-FIP2 interaction, whereas Panel E suggests it is. This contradiction should be addressed and explained.**

- *The explanation could be that the THP-1 cells were not silenced for endogenous Rab11 so endogenous Rab11GTPase could contribute to complex formation.*

The following text was added “However, the HEK293T cell have endogenous Rab11 that could contribute to interaction.”

Clarify whether the Rab11 used in pull-down experiments refers to Rab11a, Rab11b, or both.

- *The pulldown refers to Rab11a. However, it is known that ectopic expression of Rab11a or Rab11b gives similar results. This is evident in the new Figure EV5 (old Fig EV7), where both THP-1 cells expressing Flag-Rab11a and Flag-Rab11b show marked elevated GSDMD cleavage giving GSDMD p31, in contrast to Rab11a and Rab11b silencing that have largely the opposite effect.*

Figure 5:

- Indicate the percentage of Golgi structures that are NLRP3-positive to better quantify the observed localization.
- *A new Figure was included in Appendix Fig S4 and shows the average of NLRP3 positive TGN of the four independent experiments from the samples presented in the new Figure 6E. In one of these experiments, 98 % of TGN-structures was above the level of the highest value in the unstimulated NS RNA treated control. Also, in three out of four experiments there were no NLRP3 in the TGN of FIP2 silenced cells following LPS priming using the described thresholding. These results have been inserted as a new Appendix Figure S4.*

Figure: Percentage of NLRP3 positive TGN structures The percentage of NLRP3 positive TGN structures after LPS priming was calculated by first sorting all values in the NS RNA non-stimulated THP-1 cells in the respective experiment, in all n = 4 independent experiments, from high to low. All values larger than the highest value of the medium NS RNA control of the respective experiment were included in the calculation.

Consider whether the NLRP3 signal seen in unprimed cells is background or non-specific (the question is asked from old Fig 5).

- *When comparing LPS stimulating THP-1 wild type and NLRP3 KO we find one unspecific band of slightly higher molecular weight and one slightly lower band only present in unstimulated and LPS stimulated wild type cells Fig 2. This band becomes much stronger following LPS stimulation but is undisputedly present in unstimulated cells.*
- *In the confocal unstimulated cells investigated by confocal imaging cells show NLRP3 staining although at lower level than LPS primed cells. Using the NLRP3 KO cells to address the specificity of the NLRP3 antibody staining. In wild type THP-1 we found that the NLRP3 antibody showed both peri-nuclear and endosomal staining not found in the*

NLRP3 KO cells showing a much weaker staining proving the specificity of the NLRP3 antibody used (new Appendix Fig S3A-S3D).

Could the reduced expression of NLRP3 in FIP2-deficient cells explain the observed staining patterns?

- *Yes, in FIP2 silenced cells upon priming (new Figure 2A-B). In THP-1 cells the basal level of NLRP3 is not reduced by FIP2 silencing. This is also seen in data from primary human macrophages included in Figure EV2A-EV2C*

Figure 6: Is a direct interaction between FIP2 and NLRP3 required for NLRP3 translocation to the TGN, or is this an indirect effect? Additional experiments could help distinguish between these possibilities.

- *Much likely, but we have no direct proof. We have not yet mapped the exact motif in the N-terminal of FIP2 that binds NLRP3, other than it is in the first 192 amino acids that also contains the C2 domain (aa 1-141). Our results suggest that FIP2 resident in TGN helps NLRP3 to be transported to trans-Golgi membranes, this to enter the trans-Golgi to traffic to the early endosomes. A possibility is also that FIP2 binds the KMKK motif of NLRP3 to prevent binding to PI4P in the of TGN unstimulated and LPS primed cells, but only after accumulating at enlarged early endosomes with high PI4P. A shift in PI4P from the peri-nuclear TGN to endosomes was observed during LPS priming of THP-1 cells. These observations are summarized in a new extended version of Figure 6 (Fig. EV4A-B). These data also demonstrate that FIP2 only affects PI4P in the TGN of unstimulated cells, and not after LPS priming.*
- *However, after nigericin treatment of primed cells FIP2 again affects endosomal PI4P levels. This is probably a result of deficient vesicle trafficking from TGN to early endosomes in FIP2 silenced cells (Fig. 7).*
- *We are now planning to rescue NLRP3 expression in the NLRP3 KO cell with wild type or mutant NLRP3_{AAA} to investigate this. But this could not be done in the limited time of this revision.*

Figure 7: Why does FIP2 deficiency or MCC950 treatment reduce PI4P levels on vesicles?

This result implies that NLRP3 recruits PI4P, which contradicts some previously published findings. The authors should reconcile this discrepancy.

- *This is not necessarily a discrepancy. FIP2 is likely to control anterograde trafficking from TGN to the early endosome. Because of an impaired anterograde trafficking in FIP2 silenced cells, fewer vesicles will fuse with the early endosome resulting in less PI4P. Also, as nigericin treatment will force a stop in retrograde trafficking, the endosomes will accumulate less PI4P. This is also supported by the endosomal enlargement observed.*
- *What has been suggested is that inflammasome activation is seeded on PI4P containing endosomes (Zhang et al., 2023) or dispersed-TGN punctuate structures (Chen & Chen,*

2018), a process requiring NLRP3 oligomerization at PI4P containing membranes. MCC950 is inhibiting NLRP3's intrinsic ATPase-activity and this could prevent NLRP3 oligomerization at PI4P foci on endosomal membranes. This is supported by MCC950 treatment also results in less NLRP3 on the PI4P positive endosomes.

Figure 8:

- The current data do not establish a clear link to NLRP3 activation. It would strengthen the manuscript to demonstrate that IKK signalling is required for NLRP3 activation in this context, possibly via inhibition or genetic approaches:

- *We have used the TAK1 inhibitor 5Z-O to inhibit LPS stimulated IKK α /beta activation. Our results seem to “phenocopy” the results obtained for LPS stimulated phosphorylation of TAK1 and IKK α /beta. The translocation of NLRP3 to the TGN has been reported to be dependent on IKK β activation, and IKK β is required for the assembly and activation of the NLRP3 inflammasome (Nanda et al, 2021; Schmacke et al, 2022). In line with these results, TAK1 inhibition like FIP2 silencing markedly impaired LPS+Nigericin stimulated NLRP3 inflammasome activation which clearly demonstrate a link between FIP2, TAK1 and NLRP3 (new Figure 8).*

Referee #2:

In this MS the authors present data to support that Rab11-FIP2 and Rab11b, but not Rab11a, control caspase-1 mediated cleavage of pro-IL-1b and GSDMD, pyroptotic cell death and TAK1/IKK mediated NLRP3 translocation to the trans-Golgi network in human macrophages. They present further data to show that NLRP3 binds to Rab11-FIP2 via its KMKK motif, Rab11-FIP2 interacts with NLRP3 via its N-terminal C2-domain and that PI4P positive endosomes and ASC-speck formation were also controlled by Rab11-FIP2. This study builds on earlier work by the authors and further develops work from others presenting a mechanism to explain how endosomal regulation of PI4P is important for endosome-to-trans-Golgi network trafficking for recruitment of NLRP3 and subsequent inflammasome activation. The data broadly supports the authors conclusions although additional control data is required and further explanation for some of the observations are necessary.

Concerns to be addressed:

Fig 1A-F FIP2 knockdown or silencing only reduces inflammasome activity. *Knock down probably does not remove all the FIP2 although data is not presented which shows how successful the knock down is and this information needs to be presented in all cell types for all relevant experiments especially as the authors claim this technique is FIP2 silencing (apologies if this reviewer has missed this: so far a FIP2 blot is present in Fig 8, but not elsewhere and it needs to accompany every figure where knock down/silencing/depletion is present).*

- *We have included a FIP2 blots showing the level of FIP2 silencing in Figure 1C and 1I as well as in EV1H*

Given the knock down only partially reduces inflammasome activation can the authors explain why this is (presumably due to residual FIP2 expression after knocking down, but this needs clarifying)? The words FIP2 knock down, silencing and deletion are used interchangeably and this needs to be consistent supported by FIP2 blots to confirm the claims.

In the revised manuscript we are now only using FIP2, Rab11a and Rab11b silencing. In the case of the CRISPR/Cas9 genome edited THP-1 cell lines made deficient in ASC-, NLRP3- and FIP2-expression we are using the terms ASC KO, NLRP3 KO and FIP2 KO.

Also, in the revised manuscript we have included FIP2 blots in the revised Figure 1C, 1F and 1I (please note that the FIP2 silenced primary macrophages (now Fig. 1G-I), have changed place with the pLVX-Empty and pLVX-FIP2 expressing THP-1 cells (now Fig 1D-F).

Fig 1G-H: *why is the empty vector control here also affecting inflammasome activity?*

- *This is probably because the transduced Lentiviral vector DNA backbone may partly pre-activate the transduced cells.*
- *The FLAG-peptide might also affect cell homeostasis. In other word we have no good explanation for this.*

Fig 1I: *why would an elevation in pro-IL-1 β expression be expected in the presence of the lentiviral Flag-FIP2 (pLVX-FIP2) or Flag-Empty (pLVX-Empty) vectors as suggested by the authors "in line with these results" when the previously presented results concern changes in inflammasome activity rather than inflammasome priming?*

- *Somehow elevated FIP2 levels stabilize pro-IL-1 β , the increase is about 100% (Figure 1F, original Figure 1I). As seen in the new Figure 2C Flag-FIP2 co-expression is on average increasing pro- IL-1 β levels with 100% during LPS priming. Thus, increased FIP2 seems to stabilize pro-IL-1 β .*

Control blots showing increased or no change of FIP2 expression in test vs control transduced cells are required here.

- *A new blot showing the levels of FIP2 in the Flag-FIP2 is included in Figure 1F, part of a new series of blots replacing the original Figure 1 (previously Fig. 1I).*

Fig EV1 control blots showing the efficiency of silencing approaches need to be shown here

- *We have included blots of FIP2 and FIP1 showing the efficiency of silencing. For FIP5 we could not get any of the three commercially antibodies to work and therefore show the level of silencing by q-PCR.*

New Fig 3 (original Fig 2) again control blot along side the qPCRs showing the efficiency of knock down of the Rab proteins is required here.

- *In the old Figure 2, now new Figure 3, we have included blots of Rab11a silencing in both THP1 cells and primary human macrophages (Fig EV3) in addition to the q-PCR (Fig EV3). We could not get any of the 3 antibodies we used for Rab11b to work. Although the Rab11a blots show that Rab11b expression is intact upon Rab11a silencing.*

Rab11a seems to enhance casp 1 processing as well as proIL1 processing in the blots in 2C and D?

- *Yes, in THP-1 cells. In primary macrophages, Rab11a silencing increases pro-IL-1 β processing but not caspase-1 p20.*

Rab11a silenced THP-1 cells show enhanced Caspase-1 processing of pro-IL-1 β . In Rab11a silenced Primary macrophages there is a strong increase in IL1beta p17 that for some reason does not correlate with Caspase-1.

- *In THP-1 cell both IL-1 β p17 and the Casp1 p20 fragments are markedly increased, so is the GSDMD 31 fragment (new Figure 3C old Fig 2C). In primary macrophages, only the IL-1 β p17 is increased by Rab11a silencing while both the Casp1 p20 and GSDMD p31 fragments are as shown in the new Figure 3D (old Fig 2D). We don't have an explanation for this.*

Fig 3: at this resolution it is hard to say if FIP2 is at the core of the ASC speck. The proteins are certainly co-located, but the text needs to be modified to take this into account.

- *We have changed the heading of section to: “ FIP2 co-localizes with the ASC-specks and is needed for its formation” and the text in this section to: “Both FIP2 and NLRP3 were found to co-localize with ASC at the ASC-speck in THP-1 cells (Figs. 3A and 3B)”*

References

Chen J, Chen ZJ (2018) PtdIns4P on dispersed trans-Golgi network mediates NLRP3 inflammasome □ activation. *Nature* 564: 71-76

Hu J, Zhu Z, Chen Z, Yang Q, Liang W, Ding G (2022) Alteration in Rab11-mediated endocytic trafficking of LDL receptor contributes to angiotensin II-induced cholesterol accumulation and injury in podocytes. *Cell Prolif* 55: e13229

Huotari J, Helenius A (2011) Endosome maturation. *EMBO J* 30: 3481-3500

Nanda SK, Prescott AR, Figueras-Vadillo C, Cohen P (2021) IKKbeta is required for the formation of the NLRP3 inflammasome. *EMBO Rep* 22: e50743

Schmacke NA, O'Duill F, Gaidt MM, Szymanska I, Kamper JM, Schmid-Burgk JL, Madler SC, Mackens-Kiani T, Kozaki T, Chauhan D *et al* (2022) IKKbeta primes inflammasome formation by recruiting NLRP3 to the trans-Golgi network. *Immunity*

Zhang Z, Venditti R, Ran L, Liu Z, Vivot K, Schurmann A, Bonifacino JS, De Matteis MA, Ricci R (2023) Distinct changes in endosomal composition promote NLRP3 inflammasome activation. *Nat Immunol* 24: 30-41

□

Zulkefli KL, Houghton FJ, Gosavi P, Gleeson PA (2019) A role for Rab11 in the homeostasis of the endosome-lysosomal pathway. *Exp Cell Res* 380: 55-68

Dear Harald,

Thank you for the submission of your revised manuscript EMBOJ-2025-121050R for consideration by The EMBO Journal, and for your patience during peer review. Your manuscript has now been seen by the two original referees who had previously reviewed the first version of the work, and we have received their detailed comments, which you can find appended below.

As you will see, both experts recognize that all their previous comments and concerns have been addressed in a significantly strengthened revised version. They both find the results presented in the manuscript novel and significant (in their comments and also in the separate evaluation sheets that all referees are asked to return along with their reports). However, they both have a number of remaining conceptual and technical concerns, which must be fully addressed before we can proceed with acceptance of the manuscript for publication in The EMBO Journal.

In particular, referee #1 points out that the correct interpretation of the findings is that the observed effects occur primarily at the priming level through modulation of NLRP3 stability rather than through direct control of inflammasome activation. In addition, both referees identify a number of other remaining concerns. We agree with referee #2 that it is not sufficient to only address some of the previously raised points in the point-by-point response letter; instead, the manuscript would benefit from the addition of your responses to the main text, where most appropriate, including in those cases where you do not have an explanation (as acknowledgement of the study's limitations).

In light of the referees' comments, I would like to invite you to submit another revised version completely addressing all remaining comments. We will share the new version with the referees for their input on whether the major comments have been fully addressed. Please also submit, along with your revised manuscript, another detailed point-by-point response addressing all referees' comments.

From the editorial side, we also have a number of requests for changes and corrections in your revised manuscript. Please address them completely in your resubmission, and briefly explain how they are addressed in your point-by-point response letter.

- Three of your co-authors (i.e., Karoline Ruud Dahlen, Astrid Skjesol, and Caroline Gravastrand) could not be reached at their provided e-mail addresses. Please provide valid e-mail addresses for them.
- Please provide a list of up to 5 keywords (preferably broad terms to enhance the online search engine discoverability of your article) after the Abstract of your revised manuscript.
- Heading "Materials and methods" should be renamed to "Methods".
- The author contributions statement should be removed from the manuscript file. Instead, we use CRediT to specify the contributions of each author in the journal submission system. Please feel free to use the free text box to provide more detailed descriptions during submission. See also our guide to authors for more information:
<https://www.embopress.org/page/journal/14602075/authorguide#authorshippinguidelines>.
- We noticed that callouts for the following Figures are missing: Fig. 1D, 2D, 2F-I, 3E-I, 5G, 6G-H. Similarly, callouts for the panels of EV figures are also missing.
- Please fill in the general information section at the top of your Author Checklist.
- In the Author Checklist, please indicate in the last column only the sections of the main manuscript where the relevant information can be found; the information itself should be provided in the manuscript, not in the checklist. In addition, please note that this information in the last column should only be provided for the positively answered questions (you currently include information -in section "Cell materials"- for a question that you answered as "Not Applicable"). Please correct your Author Checklist and upload the revised version.
- There is a "Supplemental Figure 1-4" file, but it contains only one figure without legend; Appendix Supplementary Figure legends should be removed from the main manuscript file and placed below the respective Figures in an Appendix PDF file; please also note that there are two different legends for Appendix Supplementary Figure S4; the Appendix file needs to be in PDF format; its title page should contain "Appendix for:" followed by the manuscript's title and a brief Table of Contents including the page numbers for the listed items; the nomenclature for the Appendix items should be "Appendix Figure S#" and "Appendix Table S#" throughout the Appendix file and the main manuscript file (callouts).
- Please incorporate the primer sequences (currently provided in separate Tables 1, 2, and 3 in your Methods) to the main Reagents and Tools Table that is uploaded as a separate file.

- Thank you for providing some of the requested Source Data. Our team noticed that some Source Data are still missing, however: in particular, for Fig. 1A-B, 1D, 1E, 1G-H, 2B, 2D-F, 2H-I, 3a_B, 3F-J, 4C-E, 5D, 5F, 6E-H, 7E-F, 8B, 8D, 8F-G. Please also note that the Source Data for EV Figures should be zipped together in a single ZIP folder.

- Please note that EMBO press papers are accompanied online by:

A) a short (2 sentences) summary of the findings and their significance,

B) 2-5 short bullet points highlighting the key results, and

C) a synopsis image in .jpg or .png format that is exactly 550 pixels wide and 300-600 pixels high (the height is variable). Please note that all text needs to be legible at the final size.

Please upload this information along with your revised manuscript (the text for A and B should be provided in a separate Word file).

- During our standard Figure integrity checks, we noticed that Figures 3, 5, 8, and EV Figures 1, 2, 5 contain blots that are over-contrasted. Please check these blots carefully and clarify whether this is an issue from when the images were captured. Please explain what capture settings were used and if the supplied source data are pre-enhanced or if these are the raw files from the capturing system.

- During our routine data checks, our data editors have raised the following queries regarding data, Figures, and legends. Please make sure that all requests below are completely addressed in the final version of your manuscript (please highlight all changes in the revised manuscript):

1. Please note that Figure 6G is mislabeled as Figure 6F in the manuscript. This needs to be rectified.

2. Please note that the legends for Figures 2I; 3G, H, I; 8C-G; EV3 G-I are missing in the manuscript. This needs to be rectified.

3. Please provide the exact p-values in the legends of Figures 1A, B, D, E; 2B, 3B, G; 4C, 6E, G, H; 8F, G; EV2 A, B.

4. Please note that information related to "n" is missing in the legends of Figures EV1 D, F, G; EV2 A, B; EV4 C.

5. Please note that n=2 in Figures 3A, EV1 A, B. When n=2, no statistics can be calculated and shown; instead, the individual data points must be shown.

6. Please note that the error bars must be defined in the legends of Figures 2B, D, F, H, I; 3G-I; 4C-E; 6F, 7E, 8B, D; EV1 E, EV4 C.

7. Please note that the white arrows must be defined in the legends of Figures 6C, D; 7A-D; EV5 A-E.

- The order of manuscript sections and their headings must be corrected as follows: Title page - Abstract - Keywords - Introduction - Results - Discussion - Methods - Data Availability - Acknowledgements - Disclosure and Competing Interests Statement - References - Figure Legends - main Tables (if applicable) - Expanded View Figure Legends.

Please also note that as part of the EMBO publications' Transparent Editorial Process, The EMBO Journal publishes online a Peer Review File along with each accepted manuscript. This File will be published in conjunction with your paper and will include the referee reports, your point-by-point response and all pertinent correspondence relating to the manuscript. You can opt out of this by letting the editorial office know (contact@embojournal.org). If you do opt out, the Peer Review File link will point to the following statement: "No Peer Review File is available with this article, as the authors have chosen not to make the review process public in this case."

We look forward to seeing a final version of your manuscript as soon as possible. Please let us know if you have any questions and use this link to submit your revision: Link Unavailable.

Best regards,

Ioannis

Referee #1:

The authors have conducted a number of experiments and have diligently addressed my previous concerns. In particular, they

investigated whether FIP2 and Rab11a/b affect signal 1 or signal 2 of the NLRP3 inflammasome.

The new data clearly demonstrate that the absence of FIP2 (Figures 1 and 2) as well as the absence of Rab11b (Figure 3) impacts NLRP3 inflammasome activation at the level of protein stability. Specifically, in the absence of these factors, there is a marked reduction in pro-IL-1 β levels and, importantly, in NLRP3 protein itself.

Interestingly, this strong effect on "signal 1" of the inflammasome-i.e., the priming step-likely accounts for the majority of the observed changes in NLRP3-induced cell death and IL-1 β secretion upon stimulation with nigericin. However, this key finding is neither mentioned in the title nor in the abstract. While the authors discuss the effects on caspase-1 activation, IL-1 β release, cell death, and NLRP3 localization, they fail to indicate that these effects occur primarily at the priming level through modulation of NLRP3 stability. In my opinion, this omission misrepresents their actual findings and should be corrected prior to publication. Consistently, the title should be revised to reflect that the role of FIP2 and Rab11b is to regulate NLRP3 stability during LPS priming, rather than directly controlling inflammasome activation.

This bias in the interpretation of the data is consistent throughout the manuscript. The authors frequently describe their results in a way that suggests FIP2 controls NLRP3 activation, whereas the data indicate that the effects are more likely due to altered NLRP3 protein stability. I strongly recommend revising the text throughout to ensure that the descriptions are precise and accurately reflect the data.

I would like to underline, however, that this does not in any way diminish the importance or relevance of the findings for NLRP3 biology. The results remain novel, insightful, and in my view are worthy of publication in The EMBO Journal, once the above clarifications are made.

Specific Comments

Figure 1C:

The caspase-1 blots appear overexposed. If the authors wish to claim that there is no effect on pro-caspase-1 levels, they should provide shorter-exposure blots.

Figure 1F:

The same comment applies to the caspase-1 and IL-1 β blots-these exposures are too strong and may obscure differences.

Page 5, line 1:

The authors refer to Figure 1I, but this appears to be an error. Should this reference be to Figure 1F instead?

Figure 3:

I have significant concerns with the text describing this figure.

The section title, "Rab11b controls NLRP3-stimulated cell death and IL-1 β secretion," implies that Rab11b acts at the level of NLRP3 activation. However, the results clearly show that in the absence of Rab11b, there is almost no pro-IL-1 β and very little NLRP3 protein. The title should therefore be revised to more accurately reflect the findings-for example:

"Rab11b controls NLRP3-stimulated cell death and IL-1 β secretion by regulating NLRP3 and pro-IL-1 β protein levels."

In addition, the text does not adequately describe panels 3E-I, where it is evident that NLRP3 levels are strongly reduced in Rab11b-silenced cells. It is unclear whether this omission was an oversight or deliberate, but it must be addressed.

Page 7, line 14:

The section begins with the statement, "As FIP2 controlled NLRP3 inflammasome activation, ..." However, at this point, the data mainly demonstrate that FIP2 regulates NLRP3 levels, not its activation. This should be rephrased accordingly. Furthermore, the reduced ASC speck counts in FIP2 knockdown cells could simply result from the lower NLRP3 protein levels in the absence of FIP2, and this possibility should be explicitly mentioned.

Page 10, lines 20-21:

The authors state that FIP2 silencing reduces NLRP3 on the dTGN but fail to relate this to the overall reduction in total NLRP3 protein levels. Could the observed decrease in dTGN-localized NLRP3 simply reflect a general loss of NLRP3 in FIP2-deficient cells? This needs to be discussed.

Referee #2:

The authors have addressed all my comments. Note that for a few points the authors either responded to my questions, but added nothing to the text or could provide no explanation for my questions. These questions and responses should be added to the text because they address concerns with the MS and hopefully will improve it. Please see below

1. Fig 1G-H: why is the empty vector control here also affecting inflammasome activity?

- This is probably because the transduced Lentiviral vector DNA backbone may partly preactivate the transduced cells.
- The FLAG-peptide might also affect cell homeostasis. In other word we have no good explanation for this.

Fig 1I: why would an elevation in pro-IL-1 β expression be expected in the presence of the lentiviral Flag-FIP2 (pLVX-FIP2) or Flag-Empty (pLVX-Empty) vectors as suggested by the authors "in line with these results" when the previously presented results concern changes in inflammasome activity rather than inflammasome priming?

- Somehow elevated FIP2 levels stabilize pro-IL-1 β , the increase is about 100% (Figure 1F, original Figure 1I). As seen in the new Figure 2C Flag-FIP2 co-expression is on average increasing pro-IL-1 β levels with 100% during LPS priming. Thus, increased FIP2 seems to stabilize pro-IL-1 β .

2. Rab11a seems to enhance casp 1 processing as well as proIL1 processing in the blots in 2C and D?

- Yes, in THP-1 cells. In primary macrophages, Rab11a silencing increases pro- IL-1 β processing but not caspase-1 p20.

3. Rab11a silenced THP-1 cells show enhanced Caspase-1 processing of pro-IL-1 . In Rab11a silenced Primary macrophages there is a strong increase in IL1beta p17 that for some reason does not correlate with Caspase-1.

- In THP-1 cell both IL-1 β p17 and the Casp1 p20 fragments are markedly increased, so is the GSDMD 31 fragment (new Figure 3C old Fig 2C). In primary macrophages, only the IL-1 β p17 is increased by Rab11a silencing while both the Casp1 p20 and GSDMD p31 fragments are as shown in the new Figure 3D (old Fig 2D). We don't have an explanation for this.

Referee #1:

The authors have conducted a number of experiments and have diligently addressed my previous concerns. In particular, they investigated whether FIP2 and Rab11a/b affect signal 1 or signal 2 of the NLRP3 inflammasome.

The new data clearly demonstrate that the absence of FIP2 (Figures 1 and 2) as well as the absence of Rab11b (Figure 3) impacts NLRP3 inflammasome activation at the level of protein stability. Specifically, in the absence of these factors, there is a marked reduction in pro-IL-1 β levels and, importantly, in NLRP3 protein itself.

Interestingly, this strong effect on "signal 1" of the inflammasome-i.e., the priming step-likely accounts for the majority of the observed changes in NLRP3-induced cell death and IL-1 β secretion upon stimulation with nigericin.

However, this key finding is neither mentioned in the title nor in the abstract. While the authors discuss the effects on caspase-1 activation, IL-1 β release, cell death, and NLRP3 localization, they fail to indicate that these effects occur primarily at the priming level through modulation of NLRP3 stability. ***In my opinion, this omission misrepresents their actual findings and should be corrected prior to publication. Consistently, the title should be revised to reflect that the role of FIP2 and Rab11b is to regulate NLRP3 stability during LPS priming, rather than directly controlling inflammasome activation. This bias in the interpretation of the data is consistent throughout the manuscript.*** The authors frequently describe their results in a way that suggests FIP2 controls NLRP3 activation, whereas the data indicate that the effects are more likely due to altered NLRP3 protein stability. ***I strongly recommend revising the text throughout to ensure that the descriptions are precise and accurately reflect the data.***

I would like to underline, however, that this does not in any way diminish the importance or relevance of the findings for NLRP3 biology. The results remain novel, insightful, and in my view are worthy of publication in The EMBO Journal, once the above clarifications are made.

- *In this second revision we have now described in more detail that FIP2 controls NLRP3 stability. This is done throughout the paper. Also, the title has been changed accordingly.*

Specific Comments

Figure 1C:

The caspase-1 blots appear overexposed. If the authors wish to claim that there is no effect on pro-caspase-1 levels, they should provide shorter-exposure blots.

- *A blot with shorter exposure has also been included in the panel of Fig. 1C*

Figure 1F:

The same comment applies to the caspase-1 and IL-1 β blots-these exposures are too strong and may obscure differences.

Also, blots with shorter-exposure times have been included for pro-IL-1 β and pro-caspase-1 in the figure panel of Fig.

•

Page 5, line 1:

The authors refer to Figure 1I, but this appears to be an error. Should this reference be to Figure 1F instead?

- *Yes, this reference should be to Figure 1F instead. We have corrected this in the manuscript.*

Figure 3:

I have significant concerns with the text describing this figure.

The section title, "Rab11b controls NLRP3-stimulated cell death and IL-1 β secretion," implies that Rab11b acts at the level of NLRP3 activation. However, the results clearly show that in the absence of Rab11b, there is almost no pro-IL-1 β and very little NLRP3 protein. The title should therefore be revised to more accurately reflect the findings-for example:

"Rab11b controls NLRP3-stimulated cell death and IL-1 β secretion by regulating NLRP3 and pro-IL-1 β protein levels."

- *We have changed the text to: "Rab11FIP2 and Rab11b control the NLRP3 stability during the inflammasome priming step"*
- *We agree with reviewer on this point that FIP2 is important for NLRP3 and pro-IL-1 β protein stability and discuss this point more thoroughly in the revised manuscript. However, we suggest that FIP2 modulates NLRP3 stability by controlling its translocation to trans-Golgi network during LPS priming. This has been added in the Discussion section.*

In addition, the text does not adequately describe panels 3E-I, where it is evident that NLRP3 levels are strongly reduced in Rab11b-silenced cells. It is unclear whether this omission was an oversight or deliberate, but it must be addressed.

- *We have made the following change to address the concern raised by the reviewer (and ensure the reviewer that this was not done on purpose):*

“. Next, we investigated if Rab11a and Rab11b could control LPS-primed NLRP3 and pro-IL-1 β protein and mRNA stability. Interestingly, the Rab11b-, but not the Rab11a-silenced cells, showed significantly decreased NLRP3- and pro-IL-1 β protein levels (Fig. 3E-3G). Like observed for FIP2-silenced THP-1 cells, Rab11b silencing had no significant effect on neither LPS-primed NLRP3- nor pro-IL-1 β mRNAs (Figs. 3H and 3I). Together, these results show that Rab11b, but not Rab11a, mimics the effect of FIP2 on NLRP3 and pro-IL-1 β protein stability during LPS priming. It is therefore likely that Rab11b, together with FIP2, regulates NLRP3-stimulated cell death and IL-1 β secretion by stabilizing NLRP3 protein during the priming process”.

Page 7, line 14:

The section begins with the statement, "As FIP2 controlled NLRP3 inflammasome activation, ..." However, at this point, the data mainly demonstrate that FIP2 regulates NLRP3 levels, not its activation.

This should be rephrased accordingly. Furthermore, the reduced ASC speck counts in FIP2 knockdown cells could simply result from the lower NLRP3 protein levels in the absence of FIP2, and this possibility should be explicitly mentioned.

- *See answer below*

Page 10, lines 20-21:

The authors state that FIP2 silencing reduces NLRP3 on the dTGN but fail to relate this to the overall reduction in total NLRP3 protein levels. Could the observed decrease in dTGN-localized NLRP3 simply reflect a general loss of NLRP3 in FIP2-deficient cells? This needs to be discussed.

- *We agree with the reviewer and have inserted the following in the Discussion to address this point: "FIP2 silencing reduces the amount of NLRP3 on TGN during LPS priming. This could be due to a general loss of NLRP3 in FIP2-silenced cells. The possibility also exists that FIP2 controls translocation of NLRP3 to TGN and thereby contributes to its stabilization".*

Referee #2:

The authors have addressed all my comments. Note that for a few points the authors either responded to my questions but added nothing to the text or could provide no explanation for my questions. **These questions and responses should be added to the text because they address concerns with the MS and hopefully will improve it. Please see below**

1. New Fig 1D-E (old Fig 1G-H) why is the empty vector control here also affecting inflammasome activity?

- *This is probably because the cells encode the lentiviral vector DNA backbone. We have inserted the following (pg 5 first paragraph): ..."this is probably because the genomic insertion of Lentiviral vector DNA partly could preactivate the transduced cells. The encoded Flag-peptide could also play a role.*

Fig 1I: why would an elevation in pro-IL-1 β expression be expected in the presence of the lentiviral Flag-FIP2 (pLVX-FIP2) or Flag-Empty (pLVX-Empty) vectors as suggested by the authors "in line with these results" when the previously presented results concern changes in inflammasome activity rather than inflammasome priming?

- *We agree with the reviewer and have changed this to:" Accordingly, a marked increase in IL-1 β p17 and GSDMD p31 was observed in these cells that could result from high caspase-1 activity in these cells, as suggested by the observation of more caspase-1 p20 in the Flag-FIP2- than in Flag-Empty expressing cells as observed in Figure 1F. We did not observe an increase in pro-caspase-1 nor full-length GSDMD resulting from Flag-FIP2 expression that could account for this increase (Fig. 1F)".*
- Somehow elevated FIP2 s stabilize pro-IL-1 β , the increase is about 100% (Figure 1F, original Figure 1I).
- *As seen in the new Figure 2C Flag-FIP2 expression is on average increasing pro- IL-1 β levels with 100% during LPS priming. Thus, Flag-FIP2 co-expression seems to stabilize pro-IL-1 β during LPS priming.*

2. Rab11a seems to enhance casp 1 processing as well as proIL1 processing in the blots in (old 2C and D), new 3C and 3D?

- *Yes, Rab11a seems enhance caspase-1 processing in THP-1 cells. However, in primary macrophages, Rab11a silencing increases pro- IL-1 β processing but not caspase-1 p20. And see the answer below.*

3. Rab11a silenced THP-1 cells show enhanced Caspase-1 processing of pro-IL-1 β . In Rab11a silenced Primary macrophages there is a strong increase in IL1beta p17 that for some reason does not correlate with Caspase-1.

- *In primary macrophages we observe a huge variation in caspase-1 auto cleavage, that does not always with pro-IL-1 β (Figs. 1I. We have inserted the following (pg 7): "Similar results were observed in primary human macrophages where Rab11a silencing resulted in higher IL-1 β p17 (Fig. 3D). However, both caspase-1 p20 and GSDMD p31 were moderately reduced. In the Rab11b-silenced cells, IL-1 β p17 and caspase-1 p20 as well as GSDMD p31 were both reduced (Fig. 3D). Neither the pro-caspase-1 nor GSDMD full length protein were affected by Rab11a or Rab11b silencing in THP-1 cells or primary human macrophages (Figs. 3C and 3D). We cannot explain the difference in caspase-1 activity giving high IL-1 β p20, and low GSDMD p31 and caspase-1 autocleavage in the rab11a silenced cells (Fig. 3D)."*

Dear Harald,

I am very pleased to inform you that referee #1 is now fully satisfied with the new revised version and supports the publication of the manuscript in its current form (their comments are included below). Your manuscript has now been accepted for publication in The EMBO Journal - congratulations on an excellent work, and thanks for comprehensively addressing the initially raised referee concerns and our editorial requests for corrections and other changes.

You may qualify for financial assistance for your publication charges - either via a Springer Nature fully open access agreement or an EMBO initiative. Check your eligibility: <https://link.springer.com/journal/44318/how-to-publish-with-us>

If you have any questions, please do not hesitate to contact the Editorial Office. Thank you for your contribution to The EMBO Journal. Working with you has been a pleasure.

Best regards,

Ioannis

Referee #1:

I appreciate that the authors have carefully addressed my previous concerns regarding the roles of Rab11Fip2 and Rab11b in controlling NLRP3 protein stability. They have thoughtfully revised key sections of the manuscript and conducted additional experiments to strengthen their conclusions.

The revised paper represents a highly valuable contribution that advances our understanding of the NLRP3 inflammasome, and I fully support its publication in its current form.

Please note that it is The EMBO Journal policy for the transcript of the editorial process (containing referee reports and your response letters) to be published as an online supplement to each paper. If you should prefer removal of any referee-only figures included in the point-by-point response(s), e.g. because they may still be used for future publication or because they have been reproduced from published work by others, please do let us know immediately via response email.

More information is available here: <https://link.springer.com/partners/embo-press/editorial-policies#Peer%20review>